# Empirical single-cell tracking and cell-fate simulation reveal dual roles of p53 in tumor suppression

Ann Rancourt[1,2], Sachiko Sato[1,3], Masahiko S Satoh[2,3]*

[1]Glycobiology and Bioimaging Laboratory of Research Center for Infectious Diseases and Research Centre of CHU de Québec, Quebec, Canada; [2]Laboratory of DNA Damage Responses and Bioimaging, Research Centre of CHU de Québec, Quebec, Canada; [3]Faculty of Medicine, Laval University, Quebec, Canada

**Abstract** The tumor suppressor p53 regulates various stress responses via increasing its cellular levels. The lowest p53 levels occur in unstressed cells; however, the functions of these low levels remain unclear. To investigate the functions, we used empirical single-cell tracking of p53-expressing (Control) cells and cells in which p53 expression was silenced by RNA interference (p53 RNAi). Here, we show that p53 RNAi cells underwent more frequent cell death and cell fusion, which further induced multipolar cell division to generate aneuploid progeny. Those results suggest that the low levels of p53 in unstressed cells indeed have a role in suppressing the induction of cell death and the formation of aneuploid cells. We further investigated the impact of p53 silencing by developing an algorithm to simulate the fates of individual cells. Simulation of the fate of aneuploid cells revealed that these cells could propagate to create an aneuploid cell population. In addition, the simulation also revealed that more frequent induction of cell death in p53 RNAi cells under unstressed conditions conferred a disadvantage in terms of population expansion compared with Control cells, resulting in faster expansion of Control cells compared with p53 RNAi cells, leading to Control cells predominating in mixed cell populations. In contrast, the expansion of Control cells, but not p53 RNAi cells, was suppressed when the damage response was induced, allowing p53 RNAi cells to expand their population compared with the Control cells. These results suggest that, although p53 could suppress the formation of aneuploid cells, which could have a role in tumorigenesis, it could also allow the expansion of cells lacking p53 expression when the damage response is induced. p53 may thus play a role in both the suppression and the promotion of malignant cell formation during tumorigenesis.

*For correspondence:
masahiko.sato@crchudequebec.ulaval.ca

Competing interest: The authors declare that no competing interests exist.

## Editor's evaluation

This article examines the role of p53 in cell division by using a combination of live-cell imaging, cell tracking, and simulations. Overall, the results are extensively and transparently documented and are of interest to cell biologists studying cell division, cell death, and p53.

## Introduction

The p53 gene is mutated in >50% of cancers (*Kandoth et al., 2013*; *Hollstein et al., 1991*; *Petitjean et al., 2007*), and loss of p53 function is considered to be involved in tumor progression, thereby defining *TP53* as a tumor suppressor gene (*Boutelle and Attardi, 2021*; *Levine, 2019*). The cellular functions mediated by p53 are mainly related to its cellular levels, which increase in response to stress, thus activating various mechanisms, for example, cell cycle arrest, cell senescence, cell death, and

DNA damage responses (*Levine, 2019*; *Kastenhuber and Lowe, 2017*; *Lavin and Gueven, 2006*; *Vogelstein et al., 2000*; *Hafner et al., 2019*). Given that cell cycle arrest, cell senescence, and cell death could lead to the removal of the damaged cells, these processes could act to suppress tumor formation (*Kastenhuber and Lowe, 2017*; *Livingstone et al., 1992*; *Kastan et al., 1991*; *Clarke et al., 1993*; *Lowe et al., 1993*; *Campisi, 2005*; *Rufini et al., 2013*).

p53 levels are regulated via a mechanism mediated by Mdm2, which constantly degrades p53 by ubiquitination to maintain a low level. p53 accumulation occurs when Mdm2 activity is suppressed by stress-induced phosphorylation, which inhibits its ubiquitination activity, resulting in inhibition of p53 degradation (*Boutelle and Attardi, 2021*; *Kastenhuber and Lowe, 2017*; *Lavin and Gueven, 2006*; *Haupt et al., 1997*; *Honda et al., 1997*; *Kubbutat et al., 1997*). These observations suggest that increased stress levels are required to elevate p53 levels sufficiently to allow it to exert its tumor suppressor function. Maintaining a background low level of p53 may thus have evolved as a means of allowing cells to respond quickly to stress. Alternatively, given that removing p53 by gene silencing increases the efficiency of induced pluripotent stem cell formation (*Guo et al., 2014*), maintaining low levels of p53 in unstressed cells per se may have functional implications. In this regard, it has been suggested that low levels of p53 activity may be required to respond to a broad spectrum of stress (*Vousden and Lane, 2007*) through suppression of apoptosis and regulation of metabolisms (*Boutelle and Attardi, 2021*; *Kastenhuber and Lowe, 2017*; *Kruiswijk et al., 2015*; *Lassus et al., 1996*). If low levels of p53 indeed play a role in regulating cellular functions, it is plausible that loss of these functions could be involved in tumorigenesis. However, the effects of loss of low levels of p53 on cell characteristics remain unclear.

In this study, we silenced the expression of low levels of p53 using small interfering RNA (siRNA) and investigated the effects of this loss on the frequency of induction of cellular events using single-cell tracking to detect rare cellular events and creating a cell-lineage database revealing the fates of individual cells. Cell lineages can be generated by lineage-reconstruction approaches; however, these assume that all cell divisions are bipolar (*Chow et al., 2021*; *Chapal-Ilani et al., 2013*; *Zafar et al., 2020*), and thus cannot be used to analyze lineages, including cells that undergo cell death, multiple cell division, and cell fusion. To overcome this problem, we developed a single-cell-tracking method that can detect cell death, multipolar cell division, and cell fusion events in individual cells, thus allowing the generation of accurate cell-lineage data. In addition to silencing, we also used the alkylating agent, *N*-methyl -*N'*-nitroso-*N*-nitrosoguanidine (MNNG), which acts as a DNA-damaging agent by generating methylated bases (*Wood, 1996*; *Jacobs and Schär, 2012*; *Lindahl, 1990*; *Satoh and Lindahl, 1992*; *Lindahl et al., 1995*). Among MNNG-induced methylated bases, $O^6$-methylguanine, which acts as a premutagenic DNA lesion that induces a G:C to A:T transversion mutation, is repaired by $O^6$-methylguanine methyltransferase, and 7-methylguanine, which causes the induction of cell death through the formation of DNA breaks, is repaired by base excision repair (*Wood, 1996*; *Jacobs and Schär, 2012*; *Lindahl, 1990*; *Satoh and Lindahl, 1992*; *Lindahl et al., 1995*). It has been reported that MNNG can increase p53 levels in cells (*Lindahl et al., 1988*; *Kim et al., 2005*) and activate DNA repair processes that are not related to p53. We, therefore, studied how the fate trajectories of wild-type and p53-silenced cells were affected by p53-mediated and non-mediated processes at single-cell resolution. Furthermore, because single-cell tracking can reveal spatiotemporal information on individual cells, we used this information to simulate the fates of individual cells beyond the empirical limit of cell culture by developing a cell-fate simulation algorithm.

Our results obtained by single-cell tracking and cell-fate simulation suggest that the low levels of p53 in unstressed cells play a role in suppressing the induction of cell death and cell fusion, which may lead to multipolar cell division and the production of aneuploid progeny. Cell-fate simulation analysis revealed that some aneuploid progeny derived from p53-silenced (p53 RNAi) cells could propagate in an environment dominated by p53-expressing (Control) cells, while p53 RNAi cells per se were unable to gain an advantage over Control cells in terms of population expansion. In contrast, the balance of expansion between Control and p53 RNAi cells was altered by induction of the damage response by MNNG, resulting in the relative expansion of p53 RNAi cells. Thus, although low levels of p53 could act as a tumor suppressor by inhibiting the formation of aneuploid cells, this role could largely depend on the status of the cell population harboring the cells with impaired p53 function, and the stress-damaged environment.

## Results

### System to investigate the functional implications of maintaining low levels of p53 in unstressed cells

We developed a system to determine whether the low levels of p53 retained in unstressed cells had a functional role in suppressing changes in cell characteristics. We, therefore, performed concurrent video recordings of A549 p53 proficient lung carcinoma cells treated with scrambled siRNA or p53 siRNA (Control and p53 RNAi cells, respectively) and analyzed cells using single-cell tracking. Distinct from other single-cell analyses, for example, single-cell transcriptomics, which reveal the characteristics of individual cells at a certain moment in time, data obtained by single-cell tracking include information on spatiotemporal changes for individual cells, thus revealing the changes in cell population characteristics over time. Furthermore, because tracking of individual cells allows cellular events to be detected regardless of the frequency of their occurrence, the response of a cell population can be delineated at single-cell resolution with higher accuracy compared with analyses based on the average responses of a cell population.

The first step in single-cell-tracking analyses is the generation of a long-term live-cell imaging video recording continuous cell movement. Images, typically 2300 × 2300 pixel area (1200 × 1200 μm), were acquired every 10 min (*Figure 1*). Phototoxicity, which can damage cells during long-term live-cell imaging, was minimized by using differential interference contrast (DIC) imaging with near-infrared light. The single-cell-tracking videos were then started by selecting cells referred to as progenitors in the video images at a tracking time of 0 min, and single-cell tracking of these progenitors and their progeny was performed using our previous computerized single-cell-tracking analysis system (*Sato et al., 2018*). This system created fully automated live-cell imaging videos (up to 16 videos concurrently) and carried out image segmentation, automated single-cell tracking, database creation, data analysis, and archiving. Because we excluded any data related to progenitors or progeny that moved out of the field of view, the resulting single-cell-tracking data contained complete information for progenies derived from progenitors produced during the single-cell-tracking period. The resulting single-cell-tracking data thus contained the position of each cell at every time point in an image, the type of events that occurred in a cell (bipolar cell division, tripolar cell division, tetrapolar cell division [tripolar and tetrapolar cell divisions were defined as multipolar cell division], cell death, and cell fusion [see *Figure 1—figure supplement 1* for still images of bipolar cell division, multipolar cell division, cell death, and cell fusion; corresponding videos have been deposited in Dryad]), and the relationships of a cell to its parent and offspring cells. To organize single-cell-tracking data, each progenitor and its progeny was assigned a unique ID, that is, a cell-lineage number. In this study, a group of cells (a progenitor and its progeny cells) with the same ID is referred as to a cell lineage. A unique cell number was also assigned to each cell constituting a cell lineage, and a database composed of multiple cell-lineage data is referred as to a cell-lineage database. We then used the cell-lineage database to perform quantitative bioinformatics analysis (*Figure 1*) at the single-cell and cell-lineage levels. Cell population-level analysis could also be performed by collecting data for cells that belonged to different cell lineages.

Some information in the cell-lineage database could be visually represented as a cell-lineage map (*Figure 1*), containing multidimensional information, for example, number of progeny produced from a progenitor, type, and time of events that occurred in a cell or in all cells comprising the lineage, and the length of time between events. Quantitative bioinformatics analyses were performed to analyze specific aspects of the information. Furthermore, given that the information stored in a cell-lineage database can reveal a pattern of events that occurred in the cell, cell lineage, and cell population, we used this information to simulate the fates of cells by developing a cell-fate simulation algorithm.

### Quantitation of cell numbers, areas, and densities

First, we prepared uniform cell cultures for siRNA treatment using plated cells occupying ~90% of the culture surface, where >99% of cells were attached to other cells. As shown in *Figure 1—figure supplement 2a*, the number of cells/μm$^2$ increased linearly while the average cell surface area was reduced, confirming that the number of cells could increase continuously in a linear manner during video monitoring. To visualize this increase, density maps were also created at each imaging time point (*Figure 1—figure supplement 2b* for still images and *Figure 1—video 1*) by assigning a value of 1 to pixels within a 20-pixel diameter (10.43 μm) of the position of the cell (*Figure 1—figure*

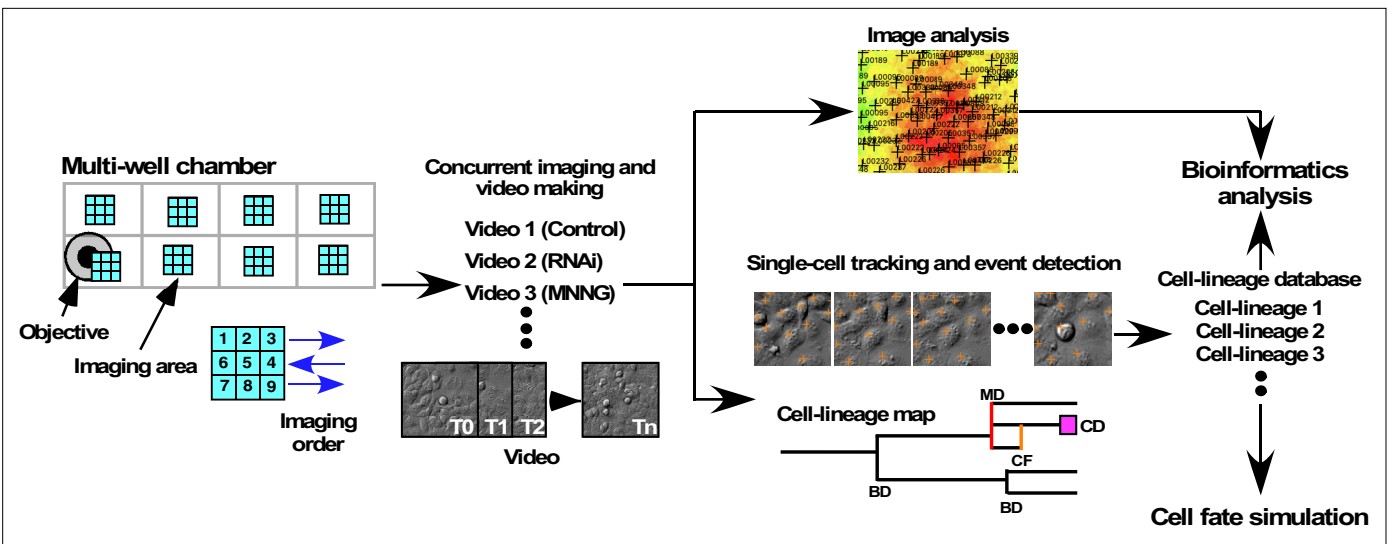

**Figure 1.** System for investigating the functional role of maintaining low cell levels of p53. A549 cells were cultured on multi-well chamber slides and the areas of interest were scanned. Cells were treated with scrambled siRNA or p53 siRNA in the presence or absence of various doses of MNNG, and the cell responses were monitored to generate live-cell imaging videos. The videos were used for imaging and single-cell tracking, starting immediately after MNNG treatment. Progenitors were selected from the image designated as tracking time 0 min (T0), and the progenitors and their progeny were tracked empirically. Cell-lineage database comprised tracking results for one progenitor and their progeny, and any events that occurred in the cells. A cell-lineage map was generated from the database to represent the proliferation profile of a progenitor and its progeny through bipolar cell division (BD), multipolar cell division (MD), cell fusion (CF), and cell death (CD). If any progeny moved out from the areas of interest, data related to the lineage of that was not included in the cell-lineage database. Quantitative bioinformatics analysis was performed using the database, which was also used to develop a cell-fate simulation algorithm.

The online version of this article includes the following video, source data, and figure supplement(s) for figure 1:

**Figure supplement 1.** Events occurring in the cells.

**Figure supplement 2.** Quantitation of cell numbers, areas, and densities.

**Figure supplement 2—source data 1.** Quantitation of cell numbers, areas, and densities.

**Figure supplement 3.** Determination of minimum number of cell-lineages required to build a cell-lineage database.

**Figure supplement 3—source data 1.** Determination of minimum number of cell lineages required to build a cell-lineage database.

**Figure supplement 4.** Accumulation of particles following siRNA transfection and analysis of p53 expression following transfection and/or MNNG exposure.

**Figure supplement 4—source data 1.** Accumulation of particles following siRNA transfection and analysis of p53 expression following transfection and/or MNNG exposure.

**Figure 1—video 1.** Visualization of cell density.

https://elifesciences.org/articles/72498/figures#fig1video1

**Figure 1—video 2.** Visualization of progress of transfection.

https://elifesciences.org/articles/72498/figures#fig1video2

---

supplement 2c). Cell positions were determined using data recorded in the cell-lineage database, and the pixel was assigned the sum of the number of overlapped areas. When a cell was close to other cells, the number of overlaps (i.e., the sum) increases, representing the cell density in an area. These sums were converted into a heat map, which showed an increase in density during video monitoring (*Figure 1—figure supplement 2b* and *Figure 1—video 1*).

## Minimum number of cell lineages required to build a cell-lineage database

We then determined the minimum number of cell lineages (i.e., number of progenitors to be tracked) needed per experiment to build a cell-lineage database. For this purpose, we first constructed 485 cell lineages by performing single-cell tracking using 485 progenitor Control cells and then randomly selected 50, 80, 120, 140, and 240 cell lineages from the 485 cell lineages to generate cell-lineage

databases containing the selected number of lineages. This selection was repeated three times for triplicate analysis for each cell-lineage database. Variation in the rate of cell population expansion was then evaluated by calculating the total number of cells at each time point using the cell-lineage databases. The results for the 50, 80, 120, 140, and 240 cell lineages are shown in *Figure 1—figure supplement 3a–e*. For example, the variation in the number of cells at the 4000 min-determined cell-lineage database containing 50 cell lineages (*Figure 1—figure supplement 3a*, black bar) was 51 cells, and the variations became constant (7.5–6.0 cells for 150–240 cell lineages) when >240 cell lineages were used (*Figure 1—figure supplement 3f*). This suggests that 240 cell lineages are suffi-cient for reproducible single-cell-tracking analysis, and we therefore used 335 cell lineages to create a cell-lineage database.

## Determination of length of time for single-cell tracking following siRNA and/or MNNG treatment

We determined the length of time required for single-cell tracking following siRNA transfection. Because transfection can be monitored by observing the accumulation of particles (transfection reagents) in cells (*Figure 1—figure supplement 4a*, DIC), we quantitated the number of particles in an area of the cell culture by extracting particles with a grayscale value >150/255 from the video image and counted the number of particles with a size of 5–30 pixels (*Figure 1—figure supplement 4a*, Particles, and *Figure 1—video 2*). The number of particles started to increase at about 1200 min (20 hr) after transfection and peaked at 3750 min (62.5 hr) (*Figure 1—figure supplement 4b*). Western blotting analysis performed 48 hr after transfection confirmed silencing of p53 (*Figure 1—figure supplement 4c*; Scrambled siRNA vs. p53 siRNA, ~70% reduction), the expression of low levels of p53 (Scrambled siRNA, 0 µM MNNG), and the occurrence of p53 stabilization by exposure of cells to 7 µM MNNG (Scrambled siRNA, 5.8-fold increase). Based on these parameters, we applied a 30 min pulse of MNNG treatment after the start of particle accumulation (*Figure 1—figure supplement 4b*; 1740 min [29 hr] after transfection). Single-cell tracking was started at the end of MNNG treatment and continued for 4000 min (66.7 hr), by which time the number of particles was reduced to 70% of that at the peak.

Notably, the degree of p53 silencing may vary among cells, and it is possible that only certain groups of cells may respond to silencing. This technique cannot reveal the level of silencing within the individual cells, but the data for the population as a whole can reveal differences, providing a clue to the function of the low levels of p53.

## Analysis of rates of cell population expansion of Control and p53 RNAi cells, and cells treated with MNNG using cell-counting and single-cell-tracking approaches

The lowest levels of p53 could occur in unstressed cells (*Boutelle and Attardi, 2021*; *Kastenhuber and Lowe, 2017*; *Lavin and Gueven, 2006*; *Haupt et al., 1997*; *Honda et al., 1997*; *Kubbutat et al., 1997*). To gain insights into the functions of such low levels of p53, we analyzed the expansion rate of a p53 RNAi cell population and their response to MNNG exposure. The rate of cell popula-tion expansion, often referred to as the cell growth curve, has generally been measured by counting cells at certain time intervals. We therefore initially determined the rate of expansion using a clas-sical, cell-counting approach (count cell numbers every 1000 min) and compared the results with those obtained by single-cell-tracking analysis (every 10 min) to evaluate the accuracy of the analysis (*Figure 2*). For the cell-counting approach, cell population expansion curves were determined manu-ally by counting cells present in images of live-cell imaging videos every 100 time points (1000 min). To this end, images corresponding to every 100 time points were divided into 25 (5 × 5) squares (~512 × 512 pixels per square, 266.24 × 266.24 µm per square), and the number of cells within a square was counted (*Figure 2—figure supplement 1*). We used three independently generated videos and randomly selected 15 squares. The results of counting of Control and p53 RNAi cells are summarized in *Figure 2a and b*, respectively. The rates of expansion of the Control and p53 RNAi cells and their responses to MNNG exposure were also compared (*Figure 2c–f*). There was no significant difference in the expansion rates of Control and p53 RNAi cells (*Figure 2c*). There was also no significant differ-ence in the expansion rates of Control cells and Control cells exposed to 1 µM MNNG (MNNG1) (*Figure 2a*, Control vs. MNNG1), while the expansion rate of p53 RNAi cells exposed to 1 µM MNNG

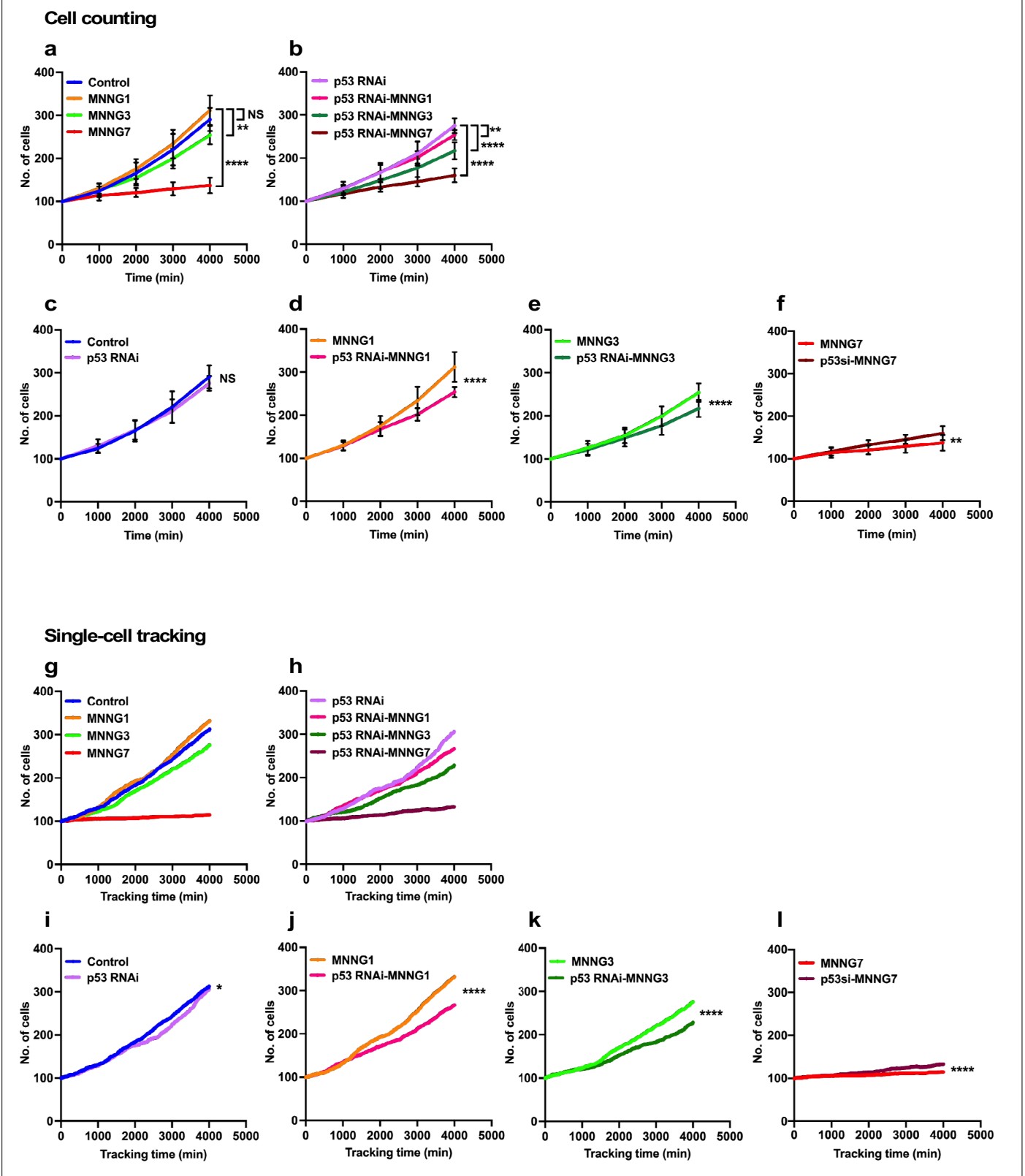

**Figure 2.** Cell population expansion rates determined by cell-counting and single-cell-tracking analysis. Control (**a**) and p53 RNAi (**b**) cells were exposed to 1, 3, and 7 μM MNNG (Control and p53 RNAi cells exposed to those doses of MNNG are referred to as MNNG1, MNNG3, MNNG7, p53 RNAi-MNNG1, p53 RNAi-MNNG3, and p53 RNAi-MNNG7, respectively). (**a–f**) Images were divided into a 5 × 5 squares and cell population expansion curves were determined by counting the number of cells in a square (~512 × 512 pixels) at 1000 min intervals (see *Figure 2—figure supplement 1*). Fifteen

*Figure 2 continued on next page*

Figure 2 continued

squares were selected from three independently generated videos. Expansion curves were compared as follows: Control vs. p53 RNAi (**c**), MNNG1 vs. p53 RNAi-MNNG1 (**d**), MNNG3 vs. p53 RNAi-MNNG3 (**e**), and MNNG7 vs. p53 RNAi-MNNG7 (**f**) cells. Student's *t*-test (**c–f**) and ordinary one-way ANOVA (Tukey's) (**a, b**) were performed (n = 15), and standard errors are shown. Cell population expansion curves of Control (**g**) and p53 RNAi (**h**) cells were determined using a cell-lineage database. Single-cell tracking was performed for 4000 min with videos acquired at 10 min intervals (n = 399). The number of cells every 10 min was extracted and plotted. Cell population expansion curves were compared as follows: Control vs. p53 RNAi (**i**), MNNG1 vs. p53 RNAi-MNNG1 (**j**), MNNG3 vs. p53 RNAi-MNNG3 (**k**), and MNNG7 vs. p53 RNAi-MNNG7 (**l**) cells. (**i–l**) The data were analyzed by Student's *t*-tests in the following approach. The difference in cell numbers between n and n + 10 min was calculated (e.g., if the numbers of cells at 10 and 20 min were 100 and 103, the difference was 3). Because the total number of imaging points was 400, the 399 differences in time points can be calculated. Those differences for Control and p53 RNAi cells and those exposed to MNNG were used for the statistical analyses. (**a–f, i–l**) NS, nonsignificant, *p<0.05, **p<0.01, ***p<0.001, ****p<0.0001.

The online version of this article includes the following source data and figure supplement(s) for figure 2:

**Source data 1.** Cell population expansion rates determined by cell-counting and single-cell-tracking analysis.

**Figure supplement 1.** Determination of cell population expansion rate using video images.

(p53 RNAi-MNNG1) was reduced relative to Control cells exposed to MNNG1 (MNNG1) and p53 RNAi cells (**Figure 2b and d**). When the dose of MNNG was increased to 3 μM (MNNG3), the expansion rate of MNNG3 cells was reduced relative to Control cells (**Figure 2a**) but was higher than that of p53 RNAi-MNNG3 cells (**Figure 2e**, MNNG3, and p53 RNAi-MNNG3). At the highest MNNG dose used in this study (7 μM, MNNG7), at which significant accumulation of p53 was observed (**Figure 1—figure supplement 4c**), the rates of expansion of both the MNNG7 and p53 RNAi-MNNG7 cell populations were significantly reduced compared with non-exposed Control and p53 RNAi cells, respectively, but p53 RNAi cells were significantly less sensitive to the MNNG dose than Control cells (**Figure 2f**). These results suggest that cells with different sensitivities to MNNG compared with Control cells were generated by silencing the low levels of p53. Furthermore, the relative sensitivities of Control and p53 RNAi cells to different doses of MNNG differed, further suggesting that these p53 RNAi cells did not respond linearly to the different doses of MNNG.

These results revealed how cell population size was influenced by p53 silencing and/or MNNG treatment. However, this approach could not provide information on how each cell responded to the treatment or how the response affected the size of the cell population. We then analyzed cell-lineage databases obtained by single-cell tracking, given that the database contains information on events such as cell death, fusion, and division, which influence the cell population expansion, at single-cell resolution. Because the cell-lineage database generated by tracking contained information on individual cells at each time point in the images (every 10 min), the total number of cells every 10 min was plotted in **Figure 2g–l** (videos of the single-cell-tracking processes, cell-lineage maps, and cell-lineage data have been deposited in Dryad). Single-cell-tracking analysis and cell counting yielded similar results (**Figure 2g and h**); the relative differences in the rates of cell expansion between the Control and p53 RNAi cells, and those treated with MNNG, were consistent between the counting and single-cell-tracking results (**Figure 2j–l**). Regarding the comparison between Control and p53 RNAi cells, although counting every 1000 min showed no significant differences, single-cell tracking, which can survey the cell number every 10 min, revealed that the population expansion rate of p53 RNAi cells was significantly lower than that of Control cells, confirming that the single-cell-tracking analysis approach produced equivalent results to the cell-counting approach (**Figure 2c and i**). However, in contrast to the classical counting approach, which only represents the number of cells at a certain time point, cell-lineage databases contain additional spatiotemporal information at single-cell resolution, which could allow a more detailed analysis of how the cell population is expanded or reduced, what type of cellular events were induced, and how they impacted the phenotype of a cell population with reduced p53 expression or DNA damage stress. We therefore conducted further analyses to gain insights into the Control and p53 RNAi cell populations and those exposed to MNNG using computer-assisted single-cell analysis.

## Analysis of reproductive ability of cells using cell-lineage database

Cell populations are known to be composed of cells with different reproductive abilities (**Puck et al., 1956**), and the overall rate of cell population expansion (**Figure 2**) is thus often influenced by the relative reproductive abilities of the cells within the population. Cell population expansion curves per se

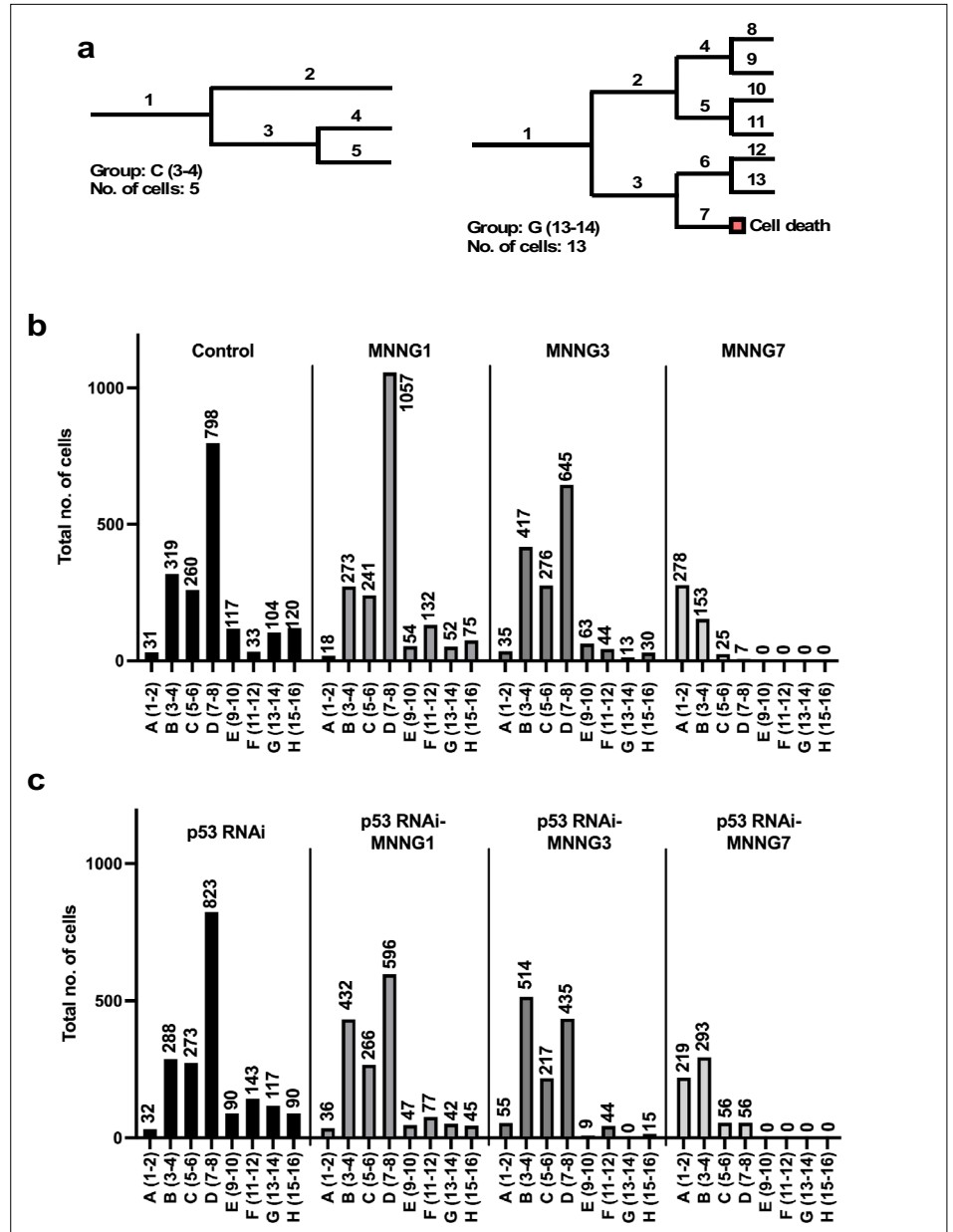

**Figure 3.** Number of progeny produced from a progenitor. (**a**) Example cell lineages are shown. If a cell lineage was composed of one progenitor (1) and four progenies (2–5) (**a**: left), the cell lineage was grouped as C (5–6). Similarly, if a cell lineage was composed of one progenitor (1) and 12 progenies (2–13) (**b**: right), the lineage was grouped as G (13–14). Cells that underwent multipolar cell division, cell death, and cell fusion were also included in this analysis. (**b, c**) The total number of cells belonging to each group was calculated. For example, if five cell lineages were composed of three cells (group B), the number of cells in group B was 5 × 3 = 15, which could represent the number of cells with the reproductive ability to produce three cells within 4000 min of culture in a cell population. (**b**) Control cells and cells exposed to MNNG. (**c**) p53 RNAi cells and cells exposed to MNNG.

The online version of this article includes the following source data for figure 3:

**Source data 1.** Number of progeny produced from a progenitor.

do not provide any insights regarding the reproductive abilities of individual cells to allow the impact of such variations in the cell population to be assessed. We therefore further revealed the characteristics of Control, p53 RNAi, and MNNG-exposed cell populations by analyzing the reproductive abilities of the different cells comprising each cell population.

To this end, we sorted each cell lineage into groups (*Figure 3a*). Given that the reproductive ability of a cell could be represented by the number of cells (progenitor and its progeny) comprising a cell lineage, we calculated the total number of cells comprising each cell lineage and sorted them according to the number (the example cell lineages shown in *Figure 3a* are composed of 5 and 13 cells). We grouped cell lineages that produced 1–2, 3–4, 5–6, 7–8, 9–10, 11–12, 13–14, and 15–16 cells as groups A, B, C, D, E, F, G, and H, respectively. The lineages in *Figure 3a*, left and right, were therefore sorted as groups C and G, respectively.

The total number of cells in each group was then calculated (*Figure 3b and c*). For example, if 20 cell lineages were composed of 11 cells (group G), the total number shown in the group G column would be 220 cells. The columns in *Figure 3b and c* thus represent the composition of cells that could have different reproductive abilities, categorized as A–H. In the case of Control cells, most cell lineages belonged to group D (7–8 cells), which accounted for 45% of the total number of Control cells (*Figure 3b*). Most p53 RNAi cell lineages also belonged to group D, but the number of cells with higher reproductive ability , that is, group F (11–12 cells), was increased relative to Control cells (Control: 33 cells vs. p53 RNAi: 143 cells), which might represent a change induced by silencing the low levels of p53 (*Figure 3c*). The number of cell lineages in group D was increased in the MNNG1 compared with the Control population (Control: 798 cells vs. MNNG1: 1057 cells), while the number of corresponding lineages in p53 RNAi-MNNG1 cells was reduced to 596 cells compared with p53 RNAi 823 cells (*Figure 3b*). This difference may reflect the difference in cell population expansion rates between MNNG1 and p53 RNAi-MNNG1 cells (*Figure 2j*). Because MNNG is a cytotoxic agent, it could reduce the number of cells with a higher reproductive ability and increase cells with a lower reproductive ability. Indeed, the numbers of group B cell lineages were increased in MNNG3 and p53 RNAi-MNNG3 cells, but MNNG3 cells could maintain a higher ability to expand relative to p53 RNAi-MMNG3 cells, because of the larger content of group D cell lineages (645 cells vs. 435 cells) (*Figure 3b and c*). Most lineages in MNNG7 and p53 RNAi-MNNG7 cells were groups A and B. However, p53 RNAi-MNNG7 cells still contained group C and D lineages, which could allow the cell population to recover from the impact of 7 µM MNNG treatment. These results suggest that the Control cells did not respond to different doses of MNNG in a simple dose–response manner and that silencing the low levels of p53 also differentially affected the response patterns. Multipolar cell division, cell death, and cell fusion, which could have a negative impact on the cell proliferation rate, could likely be involved in determining the response patterns, and we therefore further analyzed the impact of p53 silencing and MNNG exposure on these events.

## Impact of p53 silencing and MNNG treatment on cell death, multipolar cell division, and cell fusion

It has been reported that multipolar cell division occurs at a frequency of 1–10% of all cell divisions in established cultured cell lines (*Ganem et al., 2009*; *Sato et al., 2016*). Cell death could be a less-frequent event in a growing cell population, except in the presence of a cytotoxic agent. Although cell fusion is often induced during differentiation, for example, myogenesis, it is rarely induced in a growing cell population. These events would thus be expected to occur less frequently than bipolar cell division. As noted above (*Figure 1—figure supplement 3f*), more than 240 cell lineages were sufficient for reproducible single-cell-tracking-based analysis of a cell population expansion curve, which mainly represents the number of cell divisions, that is, bipolar cell divisions. Although we started with 335 progenitors to perform single-cell-tracking analysis, it was not clear if tracking this number of progenitors and their progeny would produce enough multipolar cell division, cell death, and cell fusion events for statistical analysis. To this end, we first generated reference data by visually examining the videos and counting the numbers of multipolar cell division and cell death events in three videos to gain an overview of the frequencies of these events in the entire fields of view, and compared the results with those obtained using single-cell-tracking analysis (*Figure 4*). Notably, cell fusion was not included in the counting analysis because it was difficult to detect by visual examination.

For counting, each time point of the video image was divided into four squares (*Figure 4—figure supplement 1a*), and cell death and multipolar cell division events were marked. p53 silencing significantly increased multipolar cell division and cell death events relative to Control cells (*Figure 4a and b*, *Figure 4—figure supplement 1b c*). This revealed that, even though Control and p53 RNAi cells showed similar cell population expansion curves (*Figure 2i*), the cell population sizes (total cell

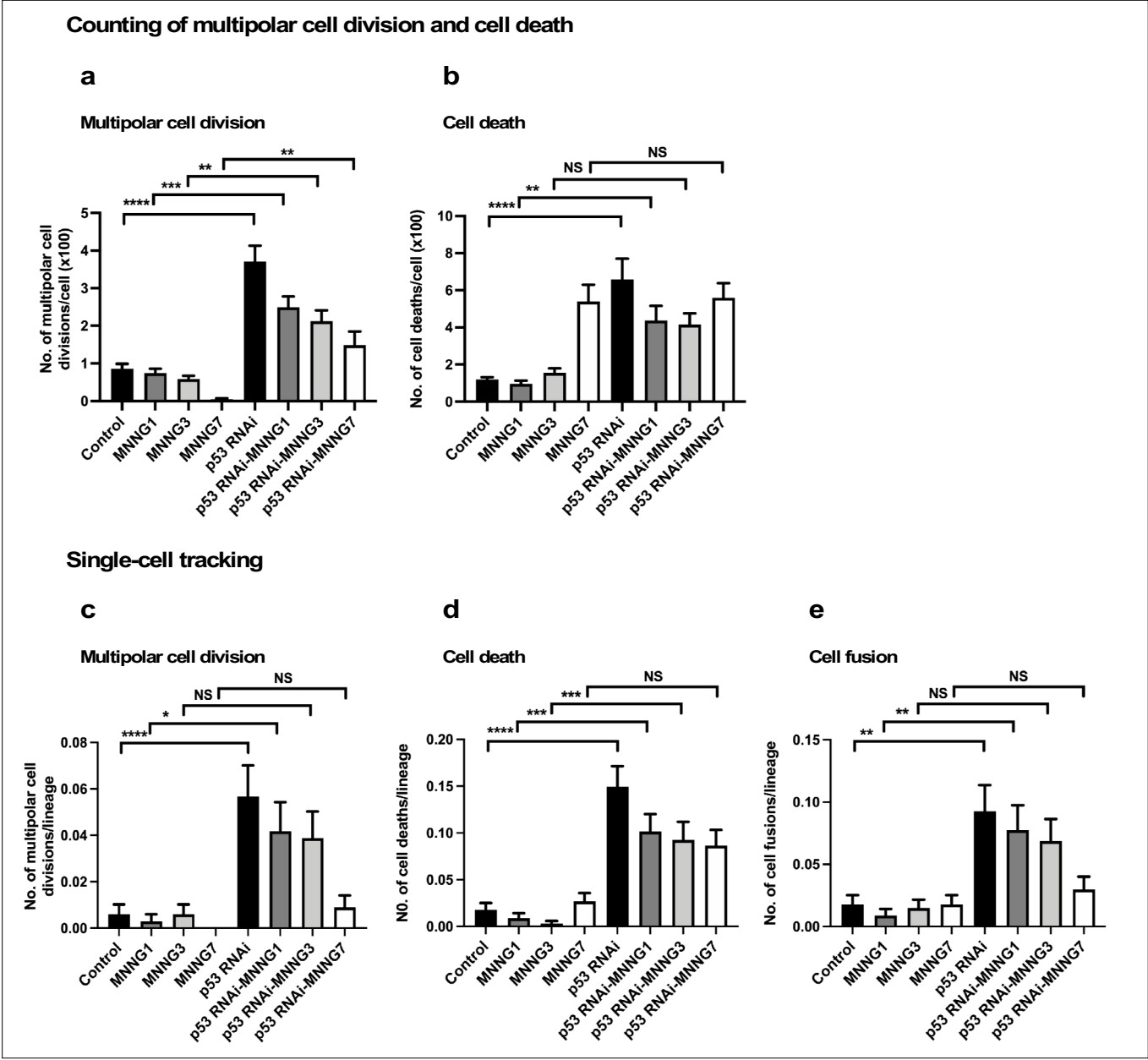

**Figure 4.** Analysis of multipolar cell division, cell death, and cell fusion using counting and single-cell-tracking analysis. Control and p53 RNAi cells were exposed to 1, 3, and 7 µM MNNG. The numbers of multipolar cell division (**a**) and cell death (**b**) events that occurred during the tracking time (4000 min) were determined by counting those events recorded in a video. Each time point of image was divided into four areas (*Figure 4—figure supplement 1*), and three independently generated videos were used. Thus, counting was performed in 12 squares (n = 12). The numbers of multipolar cell division (**c**), cell death (**d**), and cell fusion (**e**) events that occurred during the tracking time (4000 min) were determined using single-cell-tracking and cell-lineage database (n = 335). (**a–e**) Statistical analysis was performed by ordinary one-way ANOVA (Tukey's). NS, non-significant, *p<0.05, **p<0.01, ***p<0.001, ****p<0.0001. Standard errors are shown.

The online version of this article includes the following source data and figure supplement(s) for figure 4:

**Source data 1.** Analysis of multipolar cell division, cell death, and cell fusion using counting and single-cell-tracking analysis.

**Figure supplement 1.** Counting multipolar cell division and cell death using live-cell imaging videos.

numbers) of the p53 RNAi and Control cell populations resulted from different processes. Increased multipolar cell division and cell death events following p53 silencing were reduced by MNNG treatment (*Figure 4a and b*, *Figure 4—figure supplement 1b c*), with the degree of suppression of multipolar cell division being more significant than that of cell death. Multipolar cell division and cell death

occurred less frequently in Control cells, even after exposure to MNNG, except for cells exposed to 7 µM MNNG (*Figure 4b*). These results suggest that silencing of the low levels of p53 led to the induction of multipolar cell division and cell death.

We then determined the numbers of multipolar cell division and cell death events using single-cell-tracking-based cell-lineage analysis. Given that the cell-lineage database contains information on not only the events occurring in individual cells, but also their timing, it was possible to perform in-depth analysis of other events that occurred before and after the events in question, as well as sibling relationships. The areas selected for the analysis are indicated by black boxes in *Figure 4—figure supplement 1b c*. The results of analyses of multipolar cell division and cell death by single-cell-tracking analysis were consistent with the counting approach (*Figure 4a and b* [counting] vs. *Figure 4c d* [single-cell tracking]), suggesting that the number of events determined by tracking the 335 progenitors and their progeny was sufficient to represent the occurrence of these events in an entire field of view in the videos. In one case, that is, treatment of cells with 7 µM MNNG, the number of cell death events in Control cells determined by counting was higher than that determined by single-cell-tracking analysis. Given that such events tend to occur in a nonuniform manner within the field of view (*Figure 4—figure supplement 1b c*), there may be some variation between the counting and single-cell-tracking approaches. Although such variations should be taken into account when interpreting the single-cell-tracking results, this approach still has an advantage over the counting approach by detecting the cellular events encountered by a cell and their trajectories. Cell fusion, which required the fused cells to be followed for at least 30 time points (~5 hr), can thus be detected by single-cell-tracking analysis. Similar to multipolar cell division and cell death, p53 silencing increased the frequency of cell fusion, and this was counteracted by MNNG (*Figure 4e*). Taken together, these results suggest that silencing of the low levels of p53 induced multipolar cell division, cell death, and cell fusion, and these were reduced by exposure of the cells to MNNG.

## Impact of cell death on cell population expansion of p53 RNAi cells using in silico generation of a cell-lineage database

The rate of cell population expansion was similar in Control and p53 RNAi cells (*Figure 2c and i*). However, detailed analysis revealed that silencing the low levels of p53 increased multipolar cell division, cell death, and cell fusion (*Figure 4*). The probabilities of detecting bipolar cell division, multipolar cell division, cell death, and cell fusion at a certain time point in an image were calculated as 0.54, 0.001, 0.004, and 0.004/100 lineages, respectively, for Control, and 0.54, 0.014, 0.037, and 0.023/100 lineages, respectively, for p53 RNAi cells. This implies that the more frequent occurrence of these events in p53 RNAi cells relative to Control cells did not reflect the overall rate of cell population expansion, even though the events, especially cell death, could counteract the expansion of the cell population. In this regard, given that the p53 RNAi cell population contained an increased number of cells with higher reproductive ability (e.g., group F cell lineages; *Figure 3c*) than Control cells, silencing p53 may have increased cells with higher reproductive ability than Control cells, but this increase may have been counteracted by the induction of cell death (*Figure 4b and c*), resulting in similar population expansion curves for p53 RNAi cells and Control cells. In this case, if no cell death occurred in p53 RNAi cells, they would show a faster rate of cell population expansion than Control cells.

To test the effects of cell death on the rate of population expansion, we generated a cell-lineage database in silico (*Figure 5a*) by assuming that cells that undergo cell death continue to proliferate.

To this end, we searched the data for siblings of cells that underwent cell death in a cell-lineage database. Given that their proliferation pattern was likely to be similar to their siblings, we replaced the data for cells that underwent cell death with those for their siblings (*Figure 5a*). The cell population expansion rate of the in silico-generated p53 RNAi cells without cell death (p53 RNAi-Silico(-)cell death) (*Figure 5b*) showed that the cell population size was 7.4% larger than that of the p53 RNAi cells at a tracking time of 4000 min (*Figure 5b*; p53 RNAi vs. p53 RNAi-Silico(-)cell death). The number of cell deaths that occurred in p53 RNAi cells is indicated in *Figure 5b*, and the p53 RNAi cell population expansion rate was reduced around 2000 min following cell death at 1000–2000 min of tracking. This reduction was less evident in p53 RNAi-Silico(-)cell death cells, suggesting that cell death in p53 RNAi cells did indeed counteract the expansion of this cell population. The cell population size of the p53 RNAi-Silico(-)cell death cells was also larger than those of the Control and in silico-generated

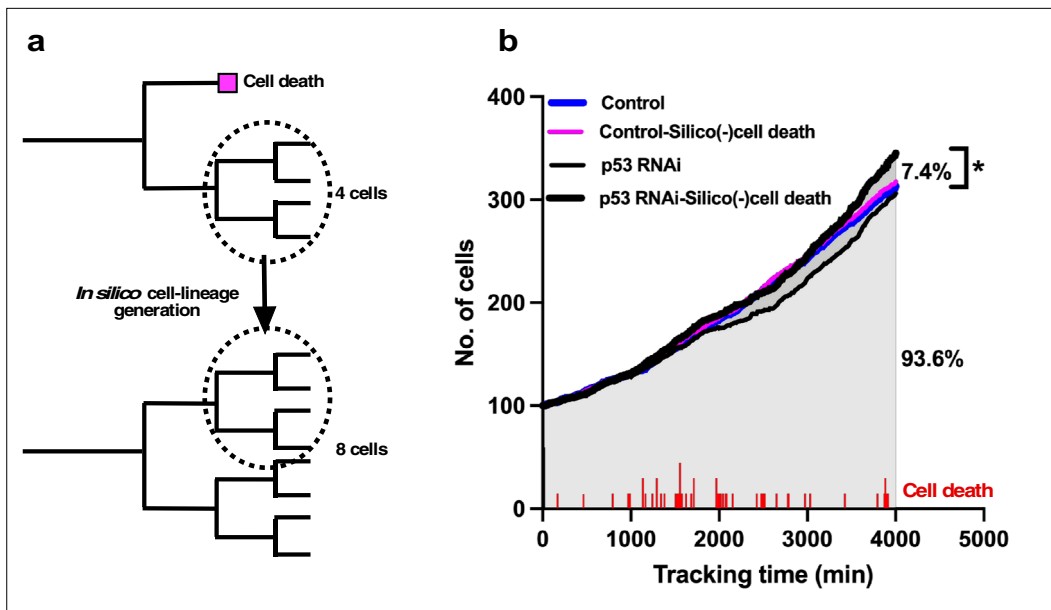

**Figure 5.** In silico generation of cell-lineage database to evaluate the impact of cell death on the cell population expansion. (**a**) To evaluate the impact of cell death on population expansion, we created an in silico cell-lineage database, in which cells do not undergo cell death, to address the impact of cell death on the cell population expansion. In the example cell lineages, the number of survived cells was reduced by half as a result of cell death. If the siblings of the dead cell grew, the dead cell was assumed to continue to grow similar to its sibling, and a cell-lineage database was created in silico based on this assumption. (**b**) A cell population expansion curve was generated using the in silico-created cell-lineage database for Control and p53 RNAi cells (Control-Silico(-)cell death and p53 RNAi-Silico(-)cell death, respectively). At a tracking time of 4000 min, cell death resulted in a 7.4% reduction in the population size of p53 RNAi cells. The frequency of cell death (red lines) occurring during tracking is also shown. Statistical analysis was performed by ordinary one-way ANOVA (Tukey's). *$p < 0.05$.

The online version of this article includes the following source data for figure 5:

**Source data 1.** In silico generation of cell-lineage database to evaluate the impact of cell death on cell population expansion.

Control cells (Control-Silico(-)cell death), where Control cells that underwent cell death were similarly replaced with their surviving siblings. These results suggest that p53 silencing promoted the reproductive ability of p53 RNAi cells, but this was counteracted by the induction of cell death, resulting in the formation of p53 RNAi cell populations that were similar to or smaller than Control cell populations (*Figure 2i*). This analysis also demonstrated that single-cell tracking provided more nuanced and detailed information about the effects of silencing p53 and the processes involved in determining the cell population size by taking account of the occurrences of multipolar cell division, cell death, and cell fusion, which are often difficult to put into spatiotemporal context in terms of the individual fate of cells when using methods that only reveal the status of cells at a certain moment in time.

## Single-cell tracking revealed the most common event preceding multipolar cell division

While cell death counteracted the expansion of the cell population, multipolar cell division and cell fusion could have greater impacts than cell death in terms of creating diversity in the cell population by causing the formation of aneuploid cells (*Boveri, 2008*; *Keryer et al., 1984*; *Holland and Cleveland, 2009*; *Gisselsson et al., 2008*; *Saunders, 2005*; *Emdad et al., 2005*) if the progenies of these processes survive. We further examined how silencing the low levels of p53 altered the characteristics of cells by focusing on multipolar cell division and cell fusion and the relationships between these events. To this end, we used cell-lineage maps (*Figure 1* and cell lineage maps deposited in Dryad) to identify cells that underwent multipolar cell division and traced back along the map to find an event that occurred prior to the division (*Figure 6a*). We identified two patterns: multipolar cell division occurring after cell fusion (Pattern 1a: cell fusion ➤ multipolar cell division) and multipolar cell division

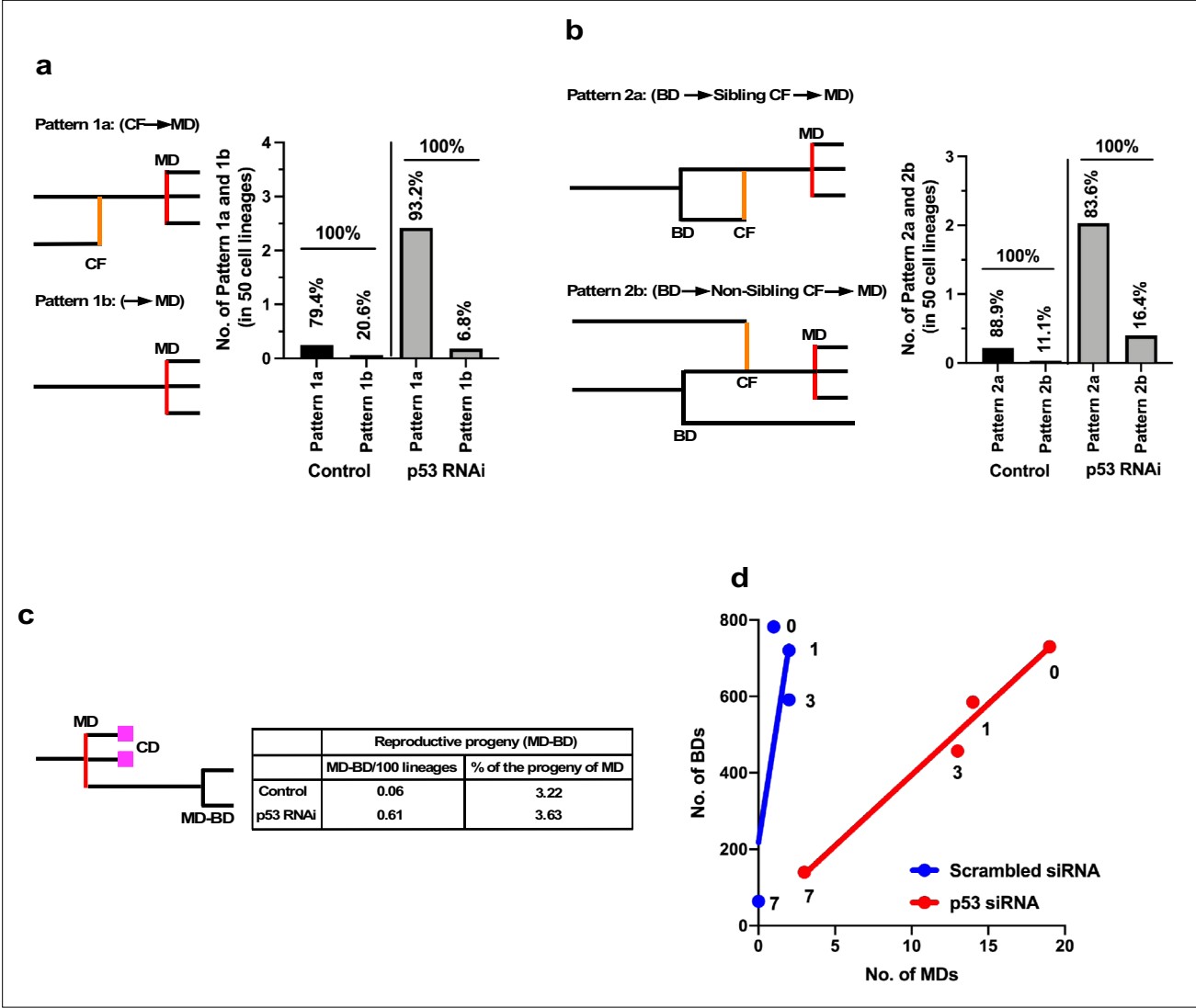

**Figure 6.** Events leading to the induction of multipolar cell division. (**a**) Cellular events that occurred prior to multipolar cell division were analyzed by tracing the cell lineage backward. Pattern 1a (cell fusion [CF] ➤ multipolar cell division [MD]): cell fusion occurred prior to multipolar cell division. Pattern 1b (➤ MD): no cell fusion occurred. The numbers of each pattern occurring in 50 cell lineages are shown. Percentages of Patterns 1a and 1b in Control and p53 RNAi cells are also shown. (**b**) Cells that underwent multipolar cell division were identified. If the cells were generated by cell fusion, the origin of the fused cells was searched by tracing the cell lineage backward. Pattern 2a (BD ➤ sibling CF ➤ MD): bipolar cell division occurred, cells produced by division were fused, followed by multipolar cell division. Pattern 2b (BD ➤ non-sibling CF ➤ MD): one sibling produced by bipolar cell division was fused with another non-sibling cell, followed by multipolar cell division. The numbers of each pattern in 50 cell lineages are shown. Percentages of Patterns 2a and 2b in Control and p53 RNAi cells are also shown. (**c**) Progeny produced by multipolar cell division was tracked and the number of reproductive progeny that underwent bipolar cell division was determined. The numbers of reproductive progeny found in 100 cell lineages and the percentages of such progeny among the total number of progeny produced by multipolar cell divisions are shown. (**d**) The number of multipolar cell division events was plotted against the number of bipolar cell division events. Numbers in (**d**) indicate doses of MNNG. (**a–d**) Because the number of multipolar cell divisions in Control cells was low, we searched three videos to find multipolar cell divisions and then traced the cells back to their progenitors to determine the events preceding multipolar cell division. To normalize the values, we assumed that multipolar cell division occurred at a similar frequency to that recorded in the cell-lineage database. BD: bipolar cell division; MD: multipolar cell division, CD: cell death; and CF: cell fusion.

The online version of this article includes the following source data for figure 6:

**Source data 1.** Events leading to the induction of multipolar cell division.

without cell fusion (Pattern 1b: multipolar cell division). Control and p53 RNAi cells demonstrated that 79.4 and 93.2% multipolar cell divisions, respectively, occurred following cell fusion (Pattern 1a). It is likely that cells with increased ploidy could have a higher chance of undergoing multipolar cell division and silencing of p53 increased the incidence of Pattern 1a (*Figure 6a*, Pattern 1a, Control

vs. p53 RNAi). We then analyzed cells that fused by tracing further back along the cell-lineage maps (*Figure 6b*). Interestingly, cell fusion was more frequent between sibling cells (Pattern 2a). In this regard, a previous study reported that some persistent links remained after bipolar cell division leading to cell fusion (*Shi and King, 2005*). However, given that cell fusion also occurred between non-siblings (Pattern 2b), the cell fusion observed in this study was mediated by other processes. In summary, p53 silencing mainly promoted the sequence of events: bipolar cell division ➤ cell fusion between siblings ➤ multipolar cell division (*Figure 6b*, Pattern 2a).

The progeny of multipolar cell division have been reported to be fragile, although some are capable of entering a proliferation cycle (*Holland and Cleveland, 2009*; *Gisselsson et al., 2008*). We, therefore, analyzed the survival of the progeny using cell-lineage maps. In terms of percentage survival, 3.2–3.6% of the total number of progeny produced by multipolar cell division survived in both Control and p53 RNAi cells (*Figure 6c*). On the other hand, given that silencing the low levels of p53 per se increased the number of multipolar cell divisions, the number of surviving progenies produced by division in the p53 RNAi cell population was increased 10.1-fold (0.61 progeny of multipolar cell division/100 cell lineages of p53 RNAi cells vs. 0.06 progeny of multipolar cell division/100 cell lineages of Control cells). These results suggest that p53 silencing mainly promoted the number of events with the sequence: bipolar cell division ➤ cell fusion between siblings ➤ multipolar cell division ➤ survival of aneuploid progeny, leading to the creation of genetic diversity in the p53 RNAi cell population.

We plotted the number of multipolar cell divisions against the number of bipolar cell divisions in Control and p53 RNAi cells and those treated with MNNG (*Figure 6d*). Although fewer multipolar cell divisions occurred in Control cells relative to p53 RNAi cells, MNNG exposure resulted in a dose-dependent linear reduction of bipolar vs. multipolar cell division, suggesting that the probability of inducing multipolar cell division was proportional to the number of bipolar cell divisions. These results also suggest that MNNG indirectly reduced the number of events in the sequence: bipolar cell division ➤ cell fusion between siblings ➤ multipolar cell division ➤ survival of aneuploid progeny. Although MNNG is a mutagen that induces G-T to A-T transversion mutations through the formation of $O^6$-methylguanine (*Lindahl et al., 1988*), it could act to suppress the formation of aneuploid cells in some contexts, with possible relevance to the development of cancer (*Rajagopalan and Lengauer, 2004*).

## Cell-fate simulation algorithm

The above data suggest that the fate of cell progeny following silencing of the low levels of p53 could alter the characteristics of the cells and their sensitivity to MNNG. However, characterization of the cells using a direct single-cell-tracking approach was limited by various factors. For example, there is a practical limit to the number of cells that can be analyzed by tracking and the duration that a cell culture can be maintained without cell passage, which makes it difficult to follow the fate of a cell beyond a certain duration. We, therefore, developed a cell-fate simulation algorithm to overcome this limitation and further analyzed the effects of p53 silencing on the fate of the cell population. For the simulation, we utilized the spatiotemporal information for each cell in the cell-lineage database records, which revealed the cell's relationships with its parents, siblings, and offspring and events that occurred in that cell. In addition to overcoming the limitation, cell-fate simulations can provide powerful and flexible tools to model conditions that are not readily accessible by direct imaging, such as simulating the fate of a cell in a mixed culture population, evaluating the response of cells to drug treatment virtually, and testing existing models in silico.

The underlying concept of the cell-fate simulation algorithm is shown in *Figure 7* (*Figure 7—source code 1*: overall scheme; and *Figure 7—figure supplements 1–6*: algorithms). We first decomposed a cell lineage into units sandwiched between two events, such as bipolar cell division, multipolar cell division, cell fusion, and cell death (*Figure 7a*). We referred to events that initiated and ended the unit as Start and End events, respectively, and each unit was thus defined by the nature of its Start and End events, and the length of time between the two events. An algorithm combining such units can thus be used to generate a virtual cell lineage with a similar pattern to an empirically determined lineage. Because each cell lineage in a cell population shows a variety of patterns, and the combination of these variations defines the characteristics of the cell population, the simulation algorithm thus needs to reflect this variation. To this end, we first produced a series of histograms of the distribution of the end events for each Start event (*Figure 7b*). For example, if the Start event was bipolar cell division, we created a histogram of the frequency of all possible End events; that is, bipolar cell division,

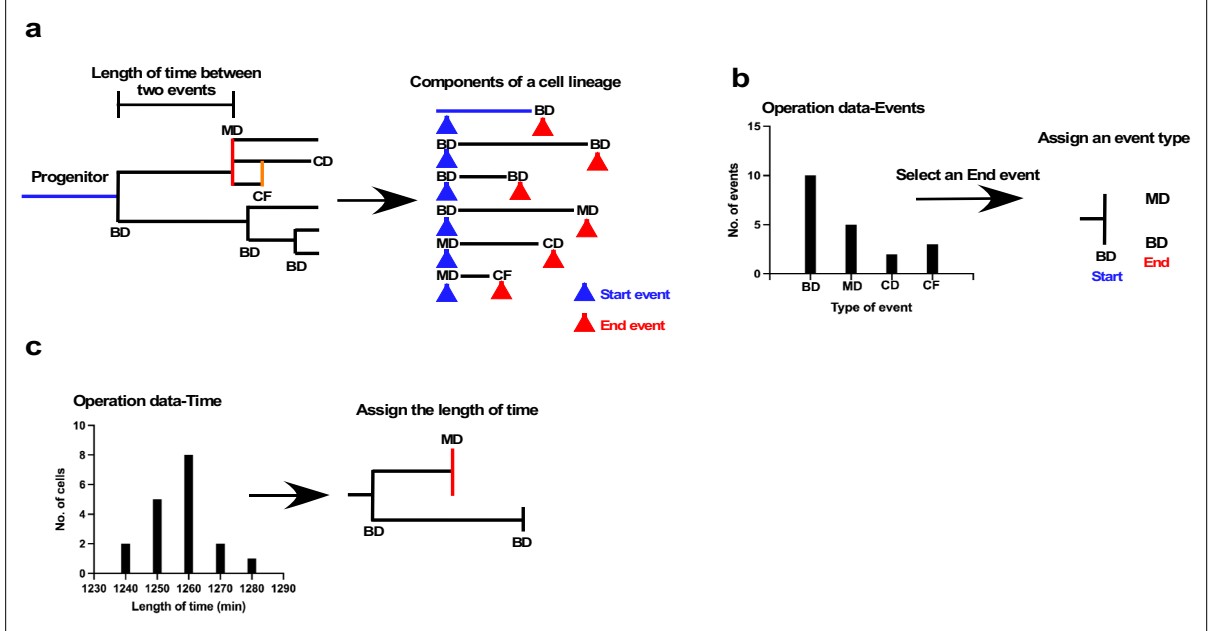

**Figure 7.** Schematic illustration of cell-fate simulation process. (**a**) A cell lineage is composed of a progenitor and its progeny. Each of those cells is generated by a Start event (blue arrowhead), followed after a certain length of time by an End event (red arrowhead), such as bipolar cell division (BD), multipolar cell division (MD), cell death (CD), and cell fusion (CF). Cell-fate simulation was performed by combining these components, types of event, and length of time between the two events. To perform the simulation, cells were classified based on the combination of the Start and the End events; for example, BD-BD, and BD-MD (see *Figure 7—figure supplement 2* for full list of classifications), and two sets of histogram data, referred to as Operation data-Events and Operation data-Time, were created. (**b**) The Operation data-Events holds the frequency of events that occurred following a Start event, such as BD, MD, or CF. In the case of the example shown in (**b**), if the Start event was BD, the histogram data indicated that the chances of the occurrence of each End event type, that is, BD, MD, CD, and CF, were 10, 5, 2, and 3, respectively. In the example, MD and BD were selected from the histogram to assign an End event type to daughter cells produced by BD. Selection was performed by generating a random value. (**c**) Operation data-Time was then used to assign the length of time between the Start and End events using histogram data (Operation data-Time) for BD-BD and BD-MD, which included the choices of length of time. The processes were repeated to generate cell-lineage data for virtual cells.

The online version of this article includes the following source code and figure supplement(s) for figure 7:

**Source code 1.** Single-cell tracking and cell-fate simulation processes.

**Source code 2.** Cell-fate simulation.

**Source code 3.** Cell-fate simulation.

**Source code 4.** Cell-fate simulation.

**Source code 5.** Cell-fate simulation.

**Figure supplement 1.** Cell-fate simulation algorithm.

**Figure supplement 2.** Cell-fate simulation algorithm.

**Figure supplement 3.** Cell-fate simulation algorithm.

**Figure supplement 4.** Cell-fate simulation algorithm.

**Figure supplement 5.** Cell-fate simulation algorithm.

**Figure supplement 6.** Cell-fate simulation algorithm.

**Figure supplement 7.** Cell-lineage maps generated by single-cell -tracking and a cell-fate simulation algorithm.

multipolar cell division, cell death, and cell fusion. The histogram indicated the unique tendency of a cell population. For example, the *Figure 7b* histogram indicates that the most frequent End event in the cell population was bipolar division followed by multipolar division. The algorithm for the simulation thus utilized the frequency to generate virtual cell lineages (see *Figure 7—figure supplement 2*: 2 for the list of Start events). We referred to the histogram data as Operation data-Events, which is composed of a series of histograms corresponding to each Start event. The algorithm then reflected this distribution of the histograms to choose the End event. The selection of the End event was performed by converting the histogram data into an array by taking account of the frequency of the

occurrence of each event to generate a random number to select an event type from the array. In this manner, an event type that occurred more frequently could have a higher chance of being selected, reflecting the unique characteristics of a cell population. Once the Start and End events were chosen, the algorithm referred to another set of histograms (Operation data-Time) that showed the distribution of the length of time for a given unit (*Figure 7c*). For example, for a unit with bipolar cell division as the Start event and cell fusion as the End event, the algorithm referred to the histogram data of time length distributions of the corresponding event combination (see the list of histograms for *Figure 7—figure supplement 2*: 1). The length of time was selected in a similar manner to the End event selection; that is, converting histogram data into an array and selecting an assigned length of time by generating a random number. These processes were repeated until the cell lineage reached the desired time point. In addition to the process of assigning event type and time length, the cell-doubling time of the parent cell was also taken into account. If both the Start and End events of a parent cell were bipolar cell division, our experimental results indicated that the length of time of its offspring was limited to ±10% of that of the parent cell. This restraint was determined by repeated stimulation with single-cell-tracking data generated using A549 cells. If a parent cell had a shorter length of time than others, its progeny could also have a shorter doubling time, creating an actively growing cell lineage. In this manner, the algorithm could generate lineages with different reproductive abilities, reflecting the characteristics of a cell population obtained by empirical live-cell imaging. This restraint for the duration between divisions was only applied to cells produced by bipolar cell division that underwent another bipolar division, given that cells produced by other types of events; for example, multipolar cell division and cell fusion, have less contribution to forming a cell population than those that undergo bipolar cell division.

We summarized the overall scheme of analysis with single-cell tracking and cell-fate simulation (*Figure 7—source code 1a*). After the generation of live-cell imaging videos, images were segmented and the segmented data was used for automatic single-cell tracking. Then, the database was used for cell-lineage map creation, data analysis, and generation of Operation data. Operation data was then used for the cell-fate simulation (*Figure 7—source code 1b*). After loading the cell-lineage data, Operation data-Time and -Events were created. The length of time for progenitors and a primary event type were assigned to each progenitor, followed by the generation of cell-lineage data by assigning an event type and length of time to a cell. Example cell-lineage maps generated by empirical single-cell tracking analysis and the algorithm are shown in *Figure 7—figure supplement 2a and b*, respectively (cell-lineage maps generated by the algorithm have been deposited in Dryad). Furthermore, because the simulation per se can be carried out using Operation data, various simulation options can be created by modifying the Operation data content.

## Cell-fate simulation options with Operation data

We created five different modes of simulations (*Figure 8*): Standard, Dose simulation, Mix culture, Switch, and Mix culture-Switch (results shown in *Figure 9*, *Figure 10*, *Figure 11*, *Figure 12* and *Figure 13* were generated by Standard, Dose simulation, Standard, Mix culture, and Mix culture-Switch modes, respectively). The Standard mode simulates the fate of one cell type using one Operation data (*Figure 8a*). We applied this mode to simulate the expansion of a cell population for an extended period of time, which cannot be achieved by in vitro cell culture (*Figure 9* and *Figure 11*). The Dose simulation mode can be carried out by taking advantage of the fact that Operation data-Events and Operation data-Time are histogram data, in which values can be modified by applying a certain rule, for example, a linear relationship, to create new Operation data. For example, if histogram arrays of Operation data-Events A and B (in *Figure 8*) obtained from the single-cell analysis of cells treated with 2 and 5 µM of a drug, respectively, recorded 10 and 1 bipolar cell divisions following a Start event, the number of bipolar cell divisions of a cell population exposed to 3 µM was calculated to be 7. In the example (*Figure 8b*), we also calculated multipolar cell division, cell death, and cell fusion data stored in the histogram array. Similar calculations were made for other histogram data stored in Operation data-Events and Operation data-Time (*Figure 7—figure supplement 2*), generating new Operation data-Events and Operation data-Time. We applied this mode to simulate cellular responses to different doses of MNNG (*Figure 10*). For example, the fate of cells exposed to 5 µM MNNG can be simulated by creating Operation data for 5 µM MNNG from Operation data for 3 µM and 7 µM MNNG. The Mixed culture mode (*Figure 8c*) simulates a fate of a cell population

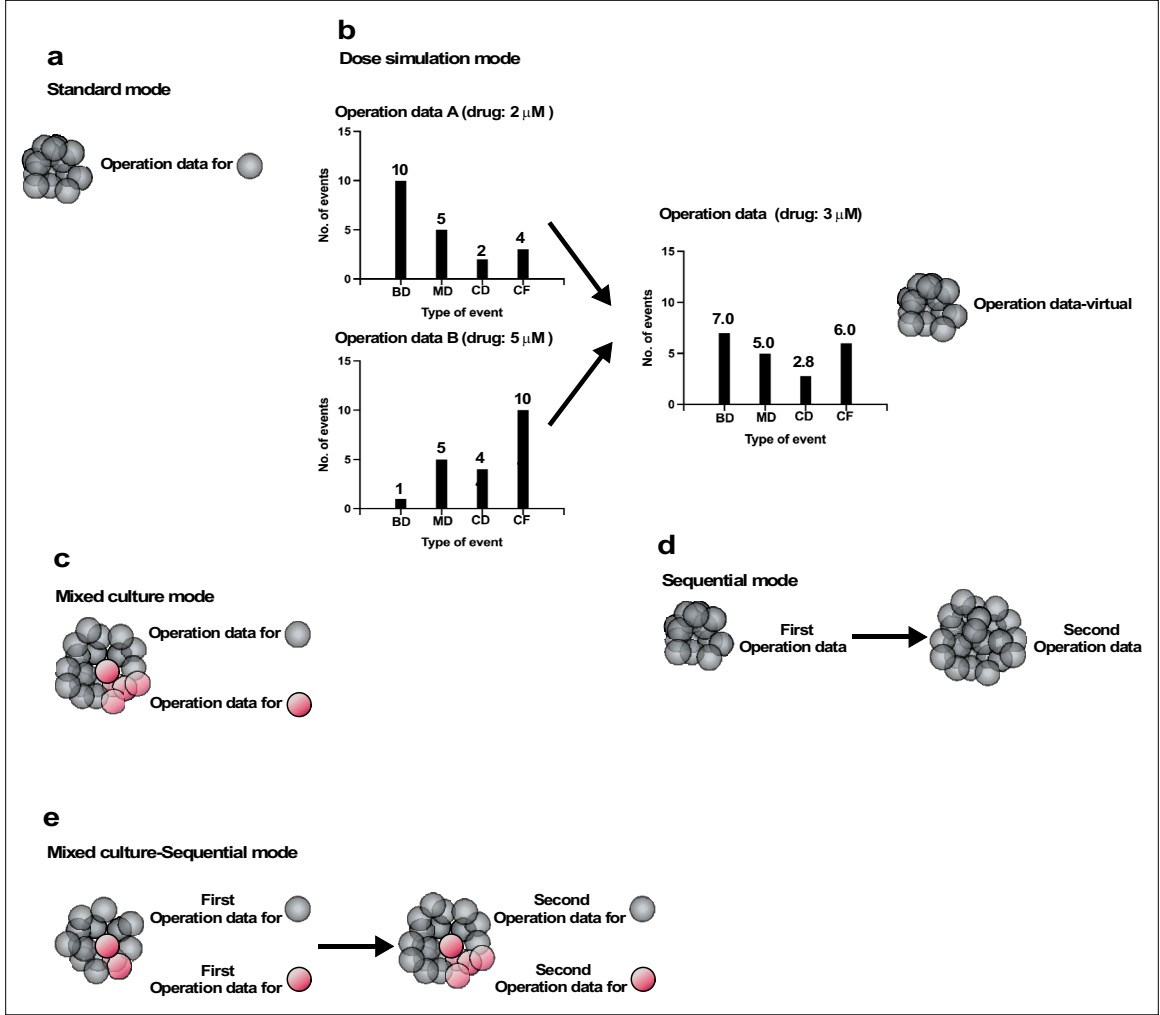

**Figure 8.** Operation modes. (**a**) Standard mode: cell-fate simulation with one type of cell population performed with Operation data. (**b**) Dose simulation mode: Operation data created from other Operation data. The example in (**b**) represents an Operation data-Events, which holds information on the chance of the occurrence of subsequent events following an event, as outlined in *Figure 7c*. In the case of Operation data A, generated from a cell-lineage database created by tracking cells exposed to a dose of a drug (e.g., 2 μM MNNG), 10, 5, 2, and 4 bipolar cell division (BD), multipolar cell division (MD), cell death (CD), and cell fusion (CF) events, respectively, occurred following the event. On the other hand, Operation data B, generated from a cell-lineage database created by tracking cells in another cell population exposed to a different dose (e.g., 5 μM) of the drug, assumed that 1, 5, 4, and 10 BD, MD, CD, and CF events would occur following the same type of events referred to in Operation A. If a cell population was assumed to be exposed to the drug at 3 μM, Operation data for the exposed cells could be created by calculating; for example, BD (7) = BD for Operation data A (10) + [(Operation data B (1) − Operation data A (10)/(Dose 2 (×5) − Dose 1 (×2))) × (Intermediate dose (×3) − Dose 1 (×2))]. A similar calculation can be performed for all data stored in Operation data-Time and -Events to create new Operation data (Operation data-virtual). Using the Operation data, the cell fate could be simulated without the need for empirical analysis. (**c**) Mixed culture mode: assuming that multiple cell types coexist in a culture, simulation was performed using corresponding Operation data for each cell type. (**d**) Switch mode: cell-fate simulation is performed with Operation data and then switched to another Operation data. If the second Operation data is for cells treated with a drug, this simulation allow the effect of the drug treatment to be evaluated virtually. (**e**) Mixed culture-Switch mode: this mode is a combined Mixed culture (**c**) and Switch (**d**) mode.

The online version of this article includes the following source data and figure supplement(s) for figure 8:

**Figure supplement 1.** Determination of number of progenitors required to perform a reproducible simulation.

**Figure supplement 1—source data 1.** Determination of number of progenitors required to perform a reproducible simulation.

composed of two or more types of cell populations using Operation data corresponding to each type of cell population. We applied this mode to simulate the expansion of cells under conditions in which 96% of cells were initially Control cells and 4% were p53 RNAi cells (*Figure 12*). The Switch mode per se was not used in this work, but this mode (*Figure 8d*) was used to perform virtual drug treatments. For example, simulation using the Operation data for Control cells was then switched to the second

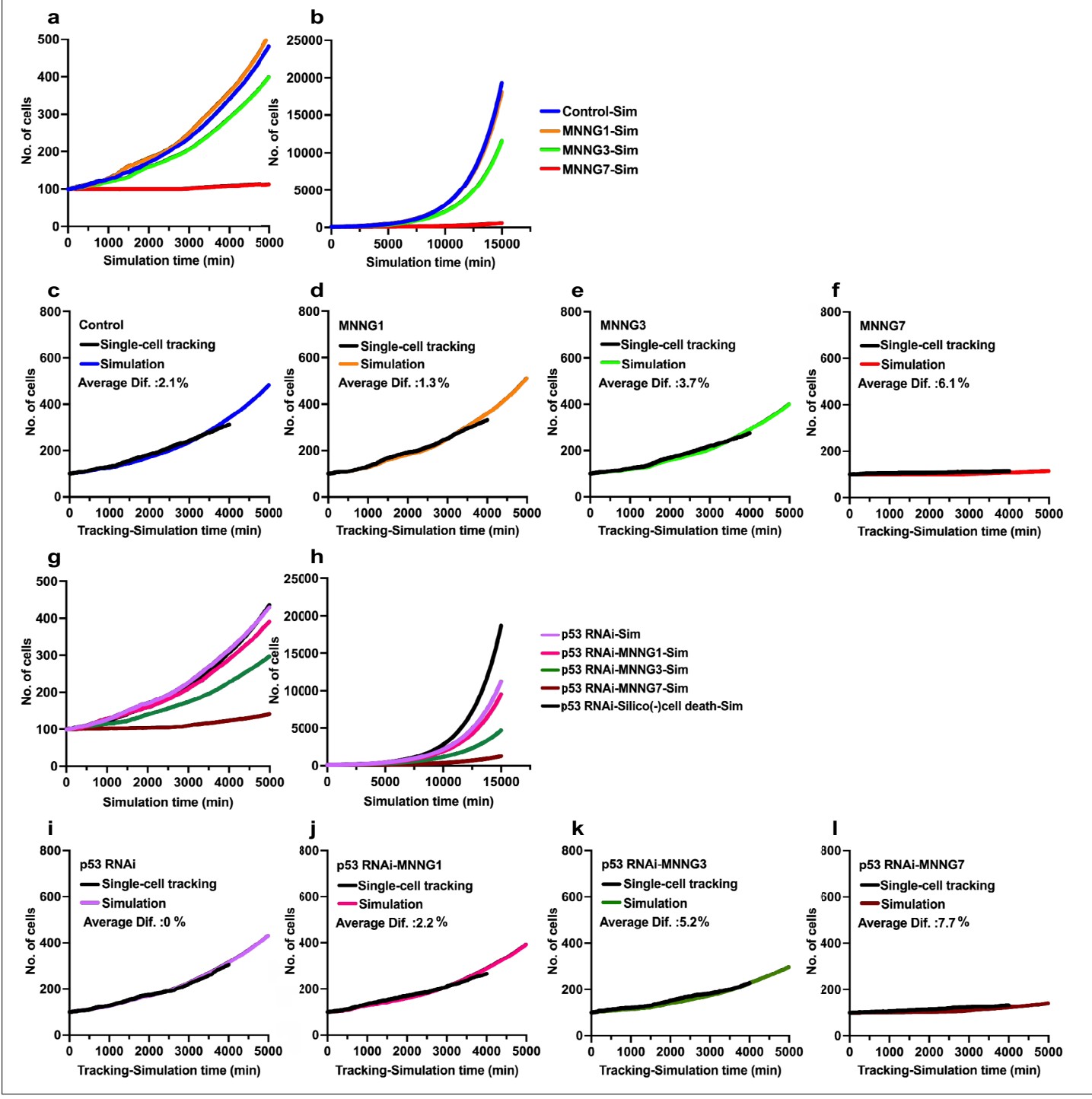

**Figure 9.** Simulations of effects of p53 silencing and MNNG exposure on cell population expansion. Simulations were performed using Operation data generated from cell-lineage data with a total of 5000 virtual progenitors for a simulation time of 15,000 min. (**a, b**) Virtual cell population expansion curves of Control-Sim cells and cells exposed to various doses of MNNG (**a**) 0–4000 min and (**b**) whole scale (0–15,000 min). (**c–f**) Data obtained from the simulation were compared with cell population expansion curves determined by single-cell-tracking analysis; (**c**) Control, (**d**) MNNG1, (**e**) MNNG3, and (**f**) MNNG7. Difference (%) was determined by obtaining the sum of the number of cells (single-cell tracking)/the number of Simulated cells determined at each simulation time point divided by 1500 simulation time ×100. For example, a difference of 2.7% implied that the simulation generated 2.7% more cells compared with the single-cell-tracking data. (**g, h**) Virtual cell population expansion curves of p53 RNAi-Sim cells and cells exposed to various doses of MNNG (**g**) 0–4000 min and (**h**) whole scale (0–15,000 min). (**i–l**) These simulation data were compared with the cell population expansion curves

*Figure 9 continued on next page*

*Figure 9 continued*

determined by single-cell-tracking analysis; (**i**) p53 RNAi, (**j**) p53 RNAi-MNNG1, (**k**) p53 RNAi-MNNG3, and (**l**) p53 RNAi-MNNG7. Results of simulation obtained using Operation data-p53 RNAi-Silico(-)cell death are also included in (**g**) and (**h**).

The online version of this article includes the following source data and figure supplement(s) for figure 9:

**Source data 1.** Simulations of effects of p53 silencing and MNNG exposure on cell population expansion.

**Figure supplement 1.** Cell-doubling time distribution.

**Figure supplement 1—source data 1.** Cell-doubling time distribution.

**Figure supplement 2.** Simulation results for numbers of multipolar cell division, cell death, and cell fusion events.

**Figure supplement 2—source data 1.** Simulations of multipolar cell division, cell death, and cell fusion.

Operation data for cells treated with, for example, 3 µM MNNG, allowing simulation of the effect of 3 µM MNNG on Control cells. The Mixed culture with Switch mode (*Figure 8e*) allowed simulations to be started with a virtual cell population composed of two or more different types of cell populations with virtual drug treatment. Thus, if two types of cell population fates are to be simulated, two Operation data are used initially and then switched to Operation data for drug-treated cells of each type (*Figure 13*). If the cell types have different treatment sensitivities, the effects of this difference can be simulated by the Mixed culture with Switch mode. We applied this mode to simulate the impact of repeated MNNG treatments on the expansion of a cell population initially comprising 96% Control and 4% p53 RNAi cells at the start of the simulation (*Figure 13h*). In summary, cell-fate simulation using Operation data allows the creation of various simulation options and provides flexibility for designing virtual experiments that would otherwise be difficult to perform empirically.

## Verification of number of progenitors used for simulation

We characterized the algorithm by performing a series of test simulations. To distinguish data produced by simulation from data produced by single-cell tracking, we referred to the cells as, for example, Control-Sim and MNNG7-Sim cells for the results produced by simulation (*Figure 8—figure supplement 1*). If the corresponding Operation data needed to be indicated, the cell populations were referred to as, for example, Operation data-Control and Operation data-MNNG7. We first performed simulations with 300, 500, 1000, 1500, and 2000 progenitors with a simulation time of 15,000 min, repeated five times to detect variations in the simulation. Cell population expansion curves of virtually created Control cells (Control-Sim cells) are shown in *Figure 8—figure supplement 1a*. Under these conditions, the variation was <2% of the average number of cells when the simulation was carried out with >1000 progenitors. The simulation performed using the Operation data-MNNG7 (*Figure 8—figure supplement 1b*) produced a larger variation, but this converged to about 6% when the simulation was carried out with >1000 progenitors. We therefore performed the simulation with 1000–2000 progenitors and repeated the simulation 5–10 times (total of 5000–10,000 progenitors).

## Comparison between cell population expansion curves determined by simulation and single-cell tracking

We then evaluated the accuracy of the simulation by referencing the cell population expansion curves determined by single-cell tracking analysis (*Figure 9*). In the simulations, we used Standard mode with Operation data generated from the cell-lineage database and performed simulation with 5000 progenitors for 15,000 min. The number of progenitors was normalized to 100 to compare the results obtained by simulation with those obtained by single-cell-tracking analysis. During the simulation, a Control-Sim cell population size increased 193.5-fold (5000–967,281 cells), and about $2 \times 10^6$ virtual cells were created in the simulation period of 15,000 min. The Control-Sim, MNNG1-Sim, MNNG3-Sim, and MNNG7-Sim cell population expansion curves (*Figure 9a*, 0–4000 min) were consistent with the curves determined using single-cell-tracking analysis (*Figure 2g*), although the Control-Sim cell curve intersected with the MNNG1-Sim cell curve at a simulation time of about 10,000 min (*Figure 9b*, 0–15,000 min). To evaluate the accuracy of the simulation, we calculated the average percent difference between the cell numbers determined by the simulation and by single-cell-tracking analysis (*Figure 9c–f*). When the number of cells determined by single-cell tracking was calculated as 100%, the average difference was 1–6%. We then performed a similar analysis with p53 RNAi cells and found that cell population expansion curves generated by the simulation showed consistent patterns

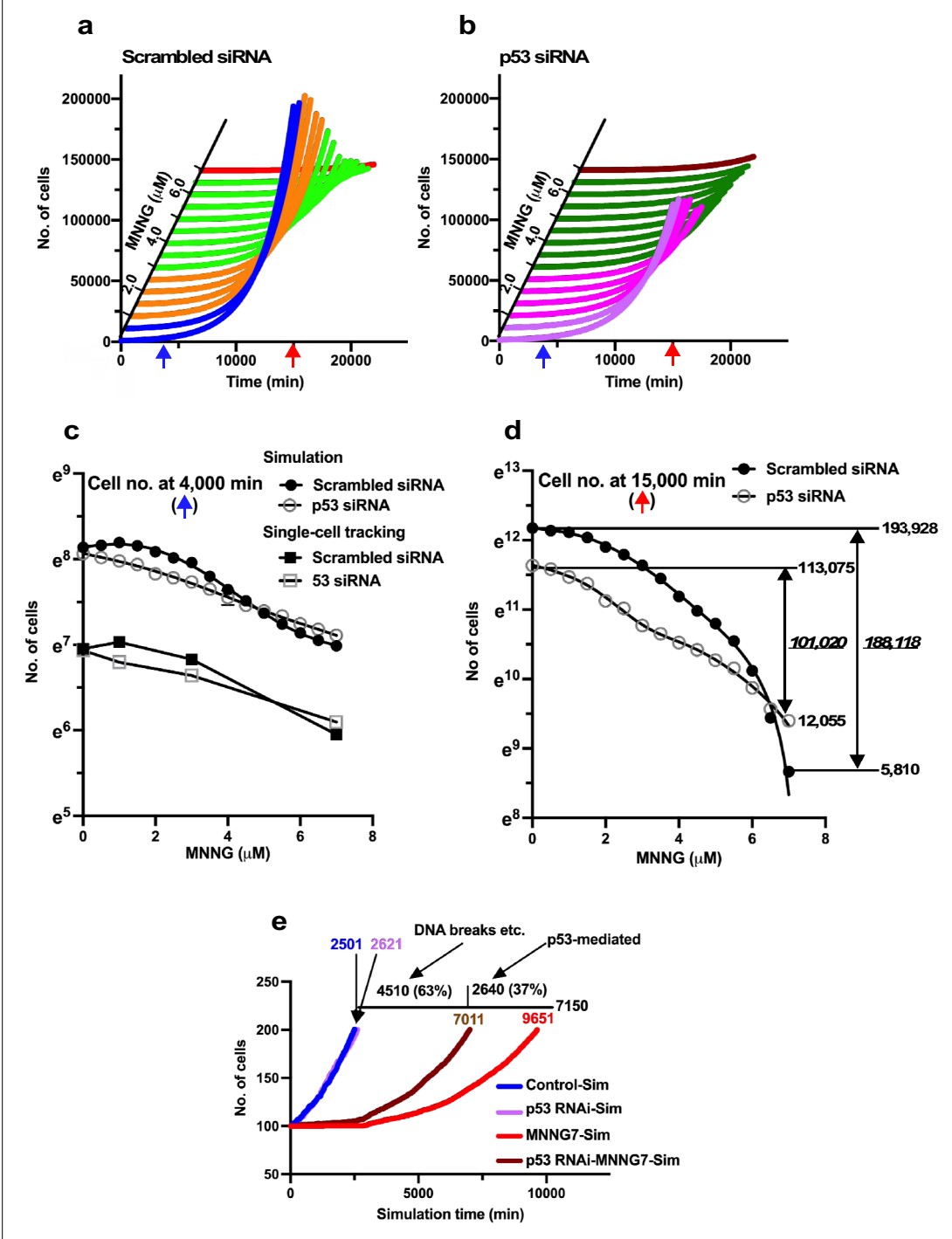

**Figure 10.** Virtual dose–response analysis. Virtual Operation data corresponding to each 0.5 µM dose of MNNG was generated in Dose simulation mode. (**a, b**) Virtual MNNG-dose–response cell population expansion curves of Control and p53 RNAi cells generated by the simulation using Operation data-Control to MNNG 1.5 (purple), 2–3 (pink), 3.5–6.5 (dark green), and 7 (maroon) are shown. (**c**) The numbers of cells at simulation times of 4000 min (indicated by a blue arrow in **a** and **b**) are plotted. Results of single-cell-tracking analysis are also included. (**d**) The numbers of cells at simulation times of 15,000 min (indicated by a red arrow in **a** and **b**) are plotted. The difference of cell number between Control cells and the cells exposed to 7 µM MNNG, and p53 RNAi cells and the cells to the dose of MNNG is shown (italic). The number of cells at 0 and 15,000 min is also shown. (**c, d**) Standard deviations are shown. A natural log scale was used. (**e**) Time necessary to reach a cell population to be doubled (100–200 cells) was determined by using Operation data-Control, MNNG7, p53 RNAi, and p53 RNAi-MNNG7. The difference in time between MNNG7-Sim (9651 min) and Control-Sim (2501 min) was 7150 min. Similarly, the difference in doubling time between p53 RNAi-MNNG7-Sim and p53 RNAi-Sim was 4510 min. Assuming that the difference in doubling time between MNNG7-Sim and Control-Sim represents the effect of MNNG induced by, for example, DNA

*Figure 10 continued on next page*

*Figure 10 continued*

breaks and p53-mediated responses, while that between p53 RNAi-MNNG7-Sim and p53 RNAi-Sim only represents the effect of MNNG induced by, for example, DNA breaks, 63 and 37% of suppression of cell population expansion in MNNG7-Sim were likely to be due to the effect of MNNG caused by, for example, DNA breaks and responses induced by the stabilization of p53, respectively.

The online version of this article includes the following source data for figure 10:

**Source data 1.** Virtual dose–response analysis.

(*Figure 9g*, 0–4000 min, and *Figure 9h*, 0–15,000 min) with those determined by single-cell-tracking analysis (*Figure 2h*). The percent differences were within 0–8% (*Figure 9i–l*). We thus concluded that the simulation could be performed within a variation of maximal 8% compared with the results of single-cell-tracking analysis.

## Doubling time of cells generated by the simulation

We also analyzed the doubling time of individual cells generated by single-cell-tracking analysis and compared them with ones generated by simulation. The average cell doubling time was prolonged following the increased dose of MNNG (31.23–41.85 hr, *Figure 9—figure supplement 1a–d*). The doubling time of individual cells generated by the simulation (*Figure 9—figure supplement 1e–h*) was similarly prolonged following the increased dose of MNNG (29.36–39.43 hr), suggesting that the algorithm could simulate the effect of MNNG on cell doubling. Similarly, the effect of MNNG treatment on the cell doubling time of p53 RNAi cells (*Figure 9—figure supplement 1i–l*) could also be simulated, showing the consistent results with the analysis of single-cell-tracking analysis (*Figure 9—figure supplement 1m–p* compared with *Figure 9—figure supplement 1i–l*). On the other hand, the simulation tended to yield an average cell doubling time about 2 hr shorter than that determined by single-cell-tracking analysis. Given that the algorithm assigned a cell-doubling time to each cell by generating a random number with Operation data-Time (*Figure 7*), a long cell-doubling time, for example, 3,000 min, which occurred less frequently, may have a lower chance of being assigned, resulting in the generation of a simulated cell population with a cell-doubling time about 2 hr shorter than that of cells analyzed by single-cell tracking. Alternatively, the cell-doubling time distribution generated by the simulation would represent the actual cell-doubling time of a cell population, given that the quantity of data generated by the simulation was ~200 times larger than that by single-cell tracking. This shorter average cell-doubling time could translate into an increased rate of cell population expansion, and the maximal 8% variation in cell population expansion curves (*Figure 9*) could reflect the cell-doubling time of the simulated cell population.

## Simulation of occurrence of multipolar cell division, cell death, and cell fusion

We examined how the algorithm simulated the occurrence of multipolar cell division, cell death, and cell fusion. Because these events were less frequent than bipolar cell division, simulations were performed 10 times with 1000 progenitors (total 10,000 progenitors). We compared the numbers of these events generated within 4000 min by the simulation with those obtained by single-cell-tracking analysis. These data were then shown as the percent of a total number of cells. The simulated results for multipolar cell division (*Figure 9—figure supplement 2a*), cell death (*Figure 9—figure supplement 2b*), and cell fusion (*Figure 9—figure supplement 2c*) showed similar tendencies to the single-cell-tracking results, and the differences between the tracking analysis and simulate results were within 1%. We, therefore, concluded that the cell-fate simulation algorithm could perform virtual cell experiments with similar accuracy to single-cell-tracking analysis.

## Virtual dose–response analysis and damage response

To begin the study with the fate simulation, we reanalyzed the responses of cells transfected with Scrambled siRNA or p53 siRNA to MNNG. To this end, Operation data were created for every 0.5 µM of MNNG from 0 to 7 µM using Dose simulation mode to determine a more detailed dose–response curve. The virtual dose–response curves determined by the simulation are shown in *Figure 10a and b*, and the numbers of cells at simulation times of 4000 and 15,000 min (indicated by blue and red arrows, respectively, in *Figure 10a and b*) are shown as natural logarithm in *Figure 10c and d*,

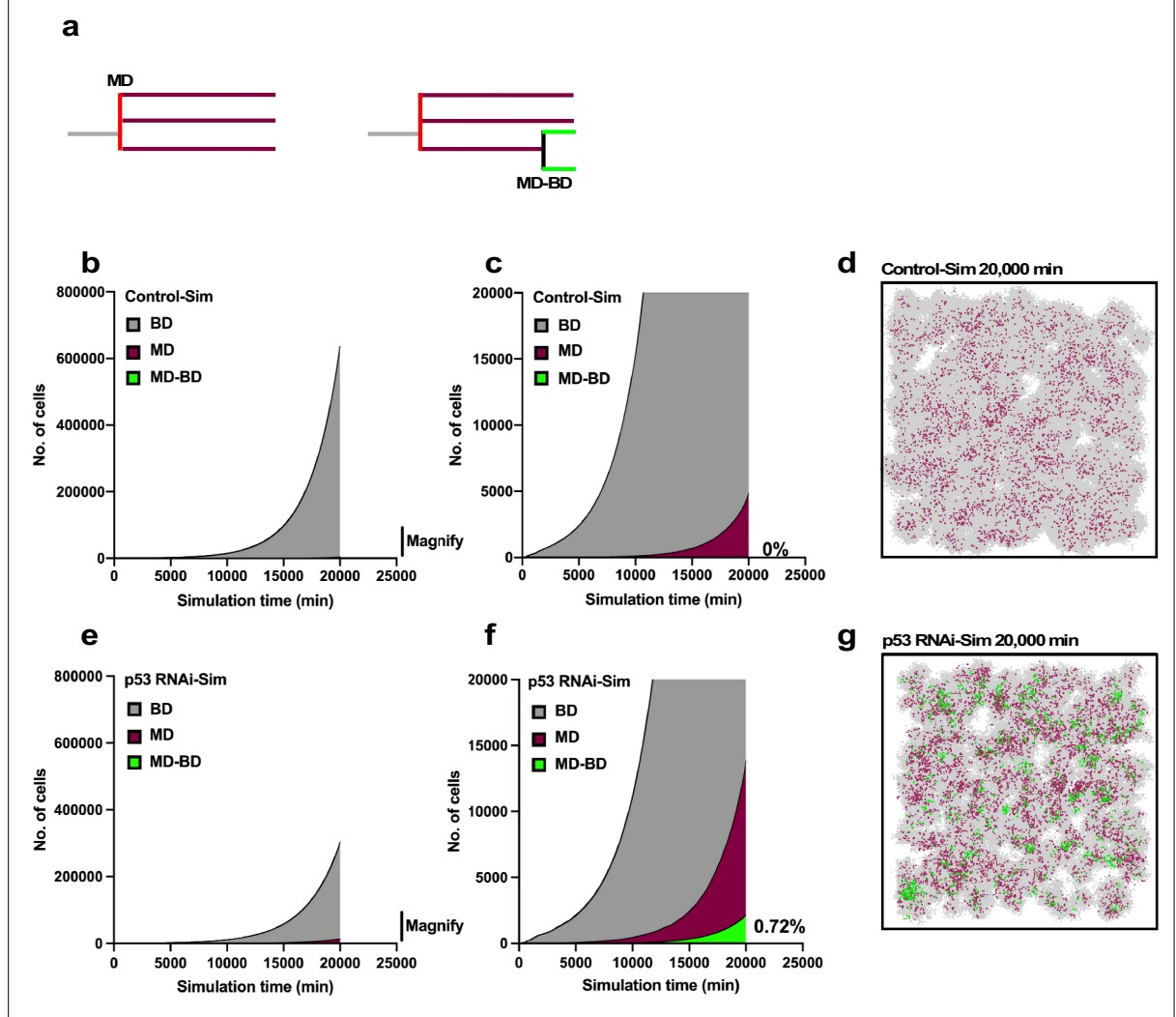

**Figure 11.** Simulation of proliferation of progeny of multipolar cell division. The simulation was performed using Operation data-Control and p53 RNAi in Standard mode. (**a**) The numbers of progeny of bipolar cell division (gray), multipolar cell division (burgundy), and progeny cells that first underwent multipolar cell division followed by bipolar cell division (green) are plotted. The simulation was carried out for 2500 progeny, which generated 6.6 × 10⁶ cells during the 20,000 min of simulation. (**b, c**) Whole-scale and magnified images generated by the simulation using the Operation data-Control are shown in (**b**) and (**c**), respectively. (**d**) An animation was created to visualize the proliferation of Control-Sim cells (*Figure 11—video 1*), and an image at simulation time 20,000 min is shown. Gray, burgundy, and green dots represent cells that underwent bipolar division, multipolar division, and multipolar division followed by bipolar division, respectively. (**e, f**) Whole-scale and magnified images generated by the simulation using Operation data-p53 RNAi are shown in (**e**) and (**f**), respectively. (**g**) An animation was created to visualize the proliferation of p53 RNAi-Sim cells (*Figure 11—video 2*), and an image at simulation time 20,000 min is shown. Bipolar cell division (BD), multipolar cell division (MD), and multipolar cell division followed by bipolar cell division (MD-BD).

The online version of this article includes the following video and source data for figure 11:

**Source data 1.** Simulation of growth of the progeny of multipolar cell division.

**Figure 11—video 1.** Cell-fate simulation of progeny of multipolar cell division in Control cells.
https://elifesciences.org/articles/72498/figures#fig11video1

**Figure 11—video 2.** Cell-fate simulation of progeny of multipolar cell division in p53 RNAi cells.
https://elifesciences.org/articles/72498/figures#fig11video2

respectively. The dose–response curves of cells containing low levels of p53 (Control cells) and p53 RNAi cells determined at a simulation time of 4000 min showed similar patterns to those determined by single-cell tracking (*Figure 10c*), suggesting that the fate of cells exposed to 0.5 µM increments of MNNG can be simulated by the Dose simulation mode. We then analyzed the effects of MNNG

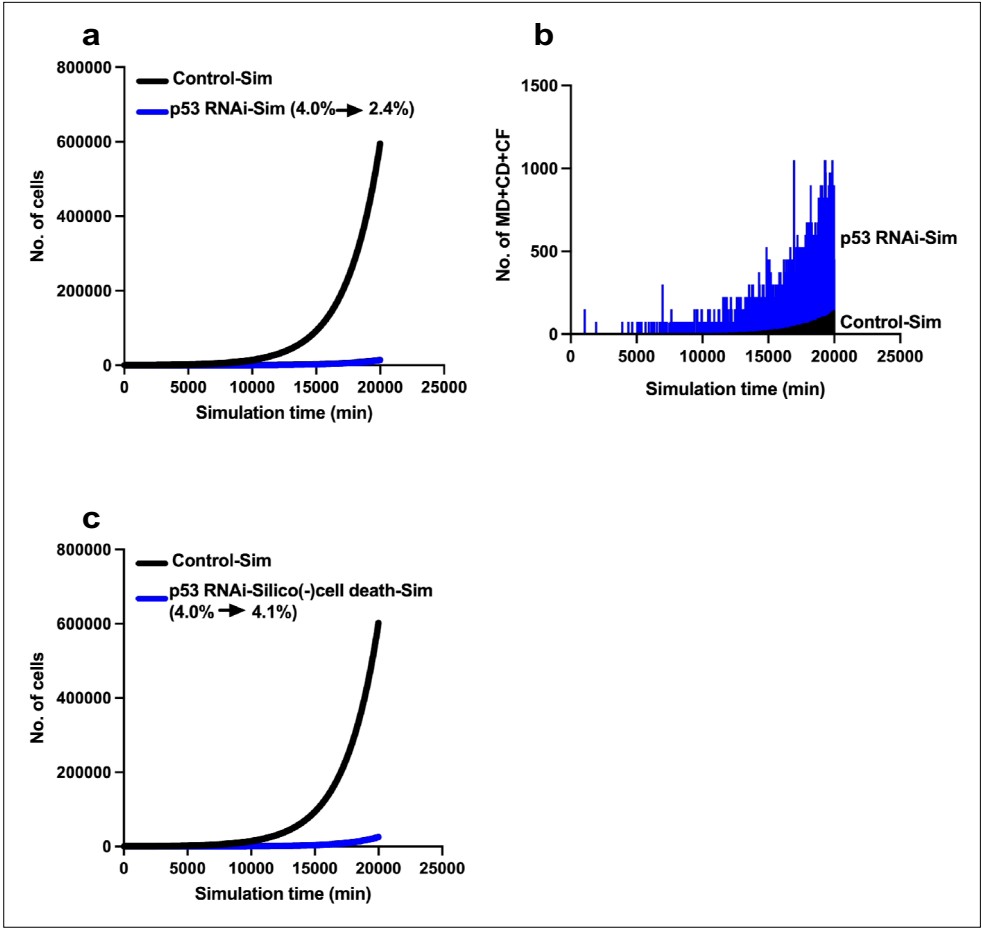

**Figure 12.** Limited expansion of p53 RNAi cells in a Control cell population. We simulated the expansion of the p53 RNAi cell population (4% at the initial time point) in a Control cell population (96% at the initial time point) using Mixed culture mode with Operation data-Control and p53 RNAi for a simulation time of 20,000 min with 500 progenitors and repeating 10 simulations (total 5000 progenitors). (**a**) Simulation of 480 Control progenitors was carried out using Operation data-Control and 20 p53 RNAi progenitors with Operation data-p53 RNAi. The initial percentage of p53 RNAi cells was 4%, which was reduced to 2.4% at 20,000 min. (**b**) The sums of multipolar cell division, cell death, and cell fusion events (MD + CD + CF) are shown. (**c**) Simulation was performed using Operation data-p53 RNAi-Silico(-)cell death.

The online version of this article includes the following video and source data for figure 12:

**Source data 1.** Limiting expansion of p53 RNAi cells in a Control cell population.

**Figure 12—video 1.** Cell-fate simulation of p53 RNAi cells in a Control cell population.
https://elifesciences.org/articles/72498/figures#fig12video1

exposure on Control and p53 RNAi cells at 15,000 min to determine the long-term impact of MNNG exposure. MNNG 7 µM reduced the cell population size of Control cells at 15,000 min from 193,928 to 5810 cells (*Figure 10d*, difference: 188,118 cells, 97%), while the number of p53 RNAi cells was reduced from 113,075 to 12,055 cells (difference: 101,020 cells, 89%), suggesting that Control cells showed a stronger inhibitory response to MNNG than p53 RNAi cells in terms of cell population expansion, possibly due to p53-induced responses (*Figure 10d*). These results suggest that the p53 RNAi cells were less sensitive to MNNG due to the removal of the low levels of p53, resulting in a low or absent p53-induced response triggered by MNNG.

MNNG induces its cytotoxic effects through the formation of DNA breaks and activation of the response caused by the accumulation of p53 (*Wood, 1996*; *Jacobs and Schär, 2012*; *Lindahl, 1990*; *Satoh and Lindahl, 1992*; *Lindahl et al., 1995*; *Lindahl et al., 1988*; *Kim et al., 2005*). It is therefore conceivable that responses induced in Control cells may be related to both DNA-break formation

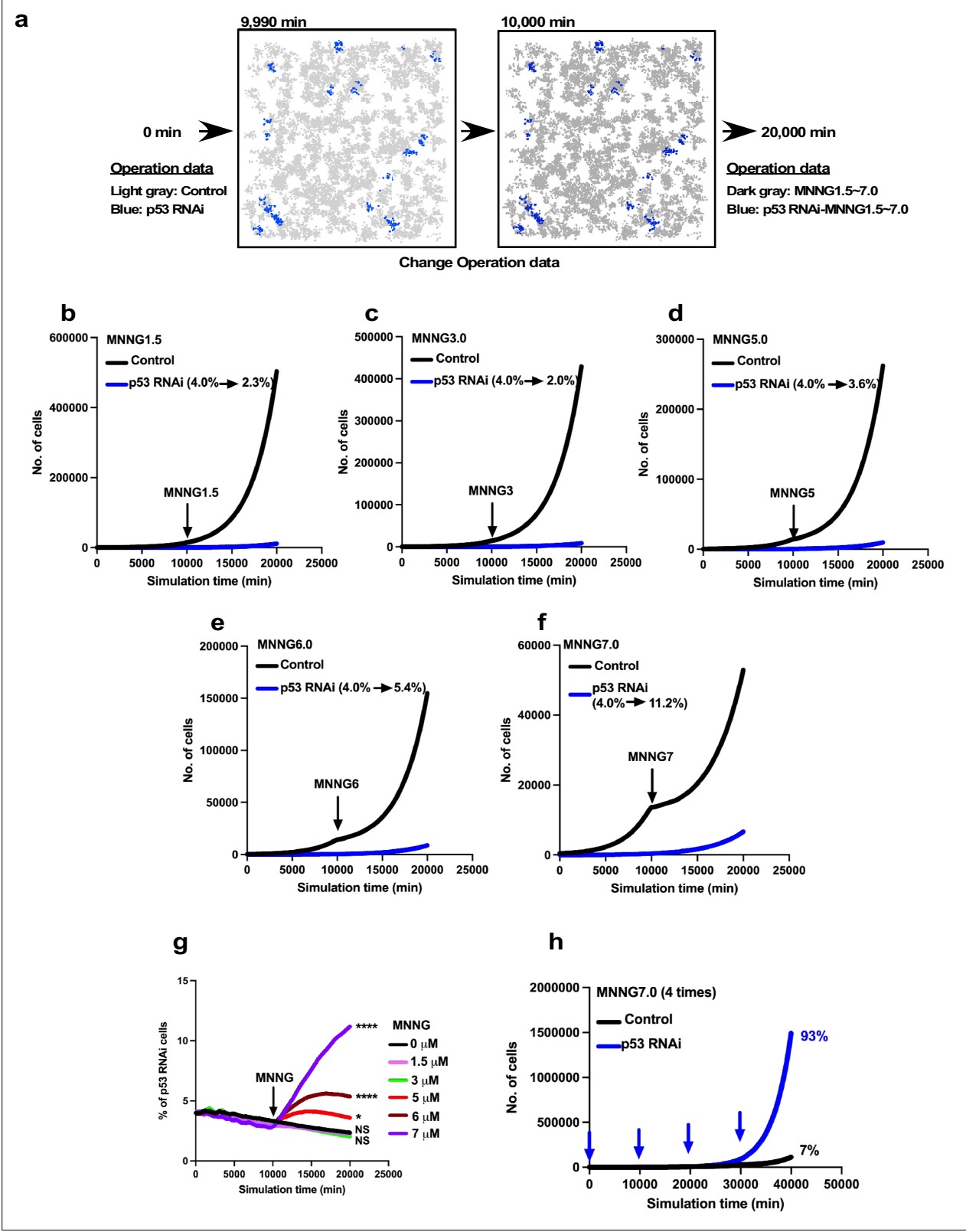

**Figure 13.** Expansion of p53 RNAi cells in a Control cell-dominant cell population subjected to virtual MNNG exposure. We simulated the response of p53 RNAi cells in a Control cell-dominant cell population to MNNG exposure using Mixed culture- Switch mode. (**a**) The simulation was performed with 96% Control and 4% p53 RNAi cells, using Operation data-Control and p53 RNAi for 9900 min. The Operation data was then switched to, for example, MNNG7 and p53 RNAi-MNNG7 for virtual MNNG treatment, and the simulation was continued up to 20,000 min. In the animation (**Figure 13—video**

*Figure 13 continued on next page*

Figure 13 continued

*1*), the pixel value of the image was reduced to indicate the virtual MNNG treatment. (**b–f**) In the simulation, Operation data corresponding to (**b**) 1.5, (**c**) 3, (**d**) 5, (**e**) 6, and (**f**) 7 µM MNNG were used to perform a virtual MNNG treatment. The numbers in the images indicate the percent of p53 RNAi cells at simulation times 0 and 20,000 min. (**g**) The percentages of p53 RNAi cells in the population are plotted. Statistical analysis (n = 10) was performed using data at simulation time 20,000 min by ordinary one-way ANOVA. NS, nonsignificant, *p<0.05, ****p<0.0001. (**h**) Based on the simulation data for 7 µM MNNG, percentage of p53 RNAi cells following repeated 7 µM MNNG exposure every 10,000 min (~7 days) was calculated. Arrows indicate the time that the virtual exposure was performed.

The online version of this article includes the following video and source data for figure 13:

**Source data 1.** Expansion of p53 RNAi cells in a Control cell population subjected to virtual MNNG exposure.

**Figure 13—video 1.** Cell-fate simulation of p53 RNAi cells in a Control cell population subjected to virtual MNNG treatment.
https://elifesciences.org/articles/72498/figures#fig13video1

and the response induced by the DNA damage-initiated accumulation of p53, while responses in p53 RNAi cells are only related to DNA-break formation. To estimate the relative contributions of these two effects to the inhibition of cell population expansion, we plotted the population expansion curves of Control-Sim, MNNG7-Sim, p53 RNAi-Sim, and p53 RNAi-MNNG7-Sim cells and determined the population doubling time (*Figure 10e*). We hypothesized that the difference in times between Control-Sim and MNNG7-Sim (7150 min) was due to both effects, while the difference between p53 RNAi-Sim and p53 RNAi-MNNG7-Sim (4510 min) represented the effect caused by the break formation. Based on these data, we estimated that about 63 and 37% of inhibition of cell population expansion caused by 7 µM MNNG exposure were due to effects related to DNA-break formation and the accumulation of p53, respectively. Although these percentages could be affected by other factors, dissecting the cellular response based on the possible mechanism of cytotoxicity could provide a deeper understanding of how cells respond to cytotoxic drugs.

## Survival of the progeny of multipolar cell division

Next, we asked how frequently the progeny of multipolar cell division can survive and grow by performing a simulation in Standard mode with 2500 progenitors for a simulation time of 20,000 min (13.8 days). Multipolar cell division-derived progeny, which lack reproductive ability (*Figure 11a*, left), and progeny that underwent multipolar cell division followed by bipolar cell division, thereby maintaining reproductive ability (*Figure 11a*, right), are shown in burgundy and green, respectively. When the simulation was performed using the Operation data-Control (*Figure 11b and c*), no reproductive progeny produced by multipolar cell division were found. To visually show such cells, we created a computer animation (*Figure 11—video 1* and *Figure 11d* for still image at 20,000 min), which visualizes the population expansion of the virtual cells, showing only burgundy cells that eventually underwent cell death or remained as nongrowing cells. However, when the simulation was performed using Operation data-p53 RNAi (*Figure 11e and f*), some progeny of multipolar cell division underwent bipolar cell division, accounting for 0.72% of the cell population at a simulation time of 20,000 min. We showed the population expansion of the progeny of multipolar cell division visually by creating a computer animation (*Figure 11—video 2* and *Figure 11g* for still image at 20,000 min), which showed the expansion of green cell populations (cells survived and expanded after multipolar cell division) in accordance with the suggestion (*Holland and Cleveland, 2009*; *Gisselsson et al., 2008*) that an aneuploid cell population, which could have distinct characteristics, could be generated by the survival of the progeny of multipolar cell division promoted by p53 silencing.

## Limiting expansion of p53 RNAi cells in the presence of Control cells

We then investigated how the presence of Control cells influenced the population expansion of p53 RNAi cells, given that cells that lack p53 likely arise within a Control cell population during tumorigenesis. We performed a simulation based on the assumption that Control and p53 RNAi cells comprised 96 and 4% of the cell population, respectively, using Mixed culture mode with Operation data-Control and p53 RNAi. A total of $1.2 \times 10^7$ virtual cells were generated after 20,000 min of simulation, and the percentage of p53 RNAi cell population was reduced from 4% to 2.4% (*Figure 12a*) because of the frequent occurrence of multipolar cell division, cell death, and cell fusion (*Figure 12b*, and *Figure 12—video 1* for the animation; p53 RNAi cells shown as blue cells). These data suggested that, although p53 RNAi cells could generate characteristic diversity in their cell population through

increased frequencies of multipolar cell division, cell death, and cell fusion compared with Control cells, this conferred a disadvantage in terms of p53 RNAi cell population expansion when both cell types co-existed in a population. We suggest that the occurrence of cell death in p53 RNAi cells may be the main factor limiting the expansion of these cells (*Figure 5b*). Indeed, a simulation using Operation data generated from p53 RNAi-Silico(-)cell death cells showed that the cell population expanded at a similar rate to Control-Sim cells (*Figure 12c*), again suggesting that cell death acted as a major factor limiting the population size of p53 RNAi cells.

## Induction of damage response in Control cells and its impact on the expansion of p53 RNAi cell population

The relative population expansion rates of Control and p53 RNAi cells were altered by exposure to MNNG (*Figure 10*: Dose simulation mode). We, therefore, simulated the responses of a cell population comprising Control (96%) and p53 RNAi cells (4%) to virtual MNNG treatment using the Mixed culture-Switch mode. We used Operation data-Control and p53 RNAi up to a simulation time of 9900 min, and the Operation data was then switched to the one for relevant doses of MNNG and the simulation was continued for another 10,000 min (total 20,000 min simulation) (*Figure 13a*). We simulated the responses of cells to 1.5, 3, 5, 6, and 7 µM MNNG. p53 RNAi cells were still unable to expand their population relative to Control cells following exposure to 1.5 and 3 µM MNNG (*Figure 13b and c*). However, p53 RNAi cells started to gain an advantage in terms of population expansion over Control cells after exposure to 5 µM MNNG (*Figure 13d*) and started to expand their population at doses of 6 and 7 µM MNNG (*Figure 13e and f*, and *Figure 13—video 1*). These results suggested that the damage responses induced in Control cells reduced its expansion speed while the expansion of the p53 RNAi cells continued (*Figure 13g*). Indeed, when the exposure of cells to 7 µM MNNG was repeated four times, 93% of the cell population was replaced with p53 RNAi cells (*Figure 13h*). These results suggest that, although p53 could suppress the formation of aneuploid cells, which could have a role in tumorigenesis, it could also allow the expansion of cells lacking p53 expression under damage-response conditions. p53 may thus play a dual role in the suppression and promotion of malignant cell formation during tumorigenesis.

## Discussion

Cell populations are known to be composed of cells with diverse phenotypic characteristics. Such diversity could be generated by the formation of a cell with distinct characteristics and the subsequent expansion of its progeny in the cell population. In this study, we revealed that silencing of the low levels of p53 generated cells with distinct characteristics from Control cells, and also affected the responses of cells to MNNG. These results suggest that a novel approach using single-cell tracking and cell-fate simulation can provide unique insights into the function of p53, thus deepening our understanding of its role in tumorigenesis.

### Low levels of p53 and possible functions

p53 is required for cells to respond to stress, through regulation of its cellular content (*Boutelle and Attardi, 2021*; *Kastenhuber and Lowe, 2017*; *Lavin and Gueven, 2006*; *Livingstone et al., 1992*; *Kastan et al., 1991*; *Clarke et al., 1993*; *Lowe et al., 1993*; *Campisi, 2005*; *Rufini et al., 2013*; *Haupt et al., 1997*; *Honda et al., 1997*; *Kubbutat et al., 1997*). The equilibrium between p53 degradation (through ubiquitination mediated by Mdm2) and its synthesis can be changed, depending on the degree of stress (*Boutelle and Attardi, 2021*; *Kastenhuber and Lowe, 2017*; *Lavin and Gueven, 2006*; *Haupt et al., 1997*; *Honda et al., 1997*; *Kubbutat et al., 1997*). Stronger stress tends to shift the equilibrium towards increased p53 levels, leading to various responses, for example, metabolic regulation, DNA damage responses, autophagy, cell cycle regulation, and cell death (*Kastenhuber and Lowe, 2017*; *Livingstone et al., 1992*; *Kastan et al., 1991*; *Clarke et al., 1993*; *Lowe et al., 1993*; *Campisi, 2005*; *Rufini et al., 2013*). On the other hand, the functions of the low levels of p53 present in unstressed cells remain unclear, but they may have a housekeeping function. Indeed, several lines of evidence, including the promotion of induced pluripotent stem cells by silencing p53 (*Guo et al., 2014*), and the spontaneous formation of tetraploid cells, as proposed cancer precursor cells, in p53 knockout mice (*Livingstone et al., 1992*; *Harvey et al., 1993*), suggest that low levels of

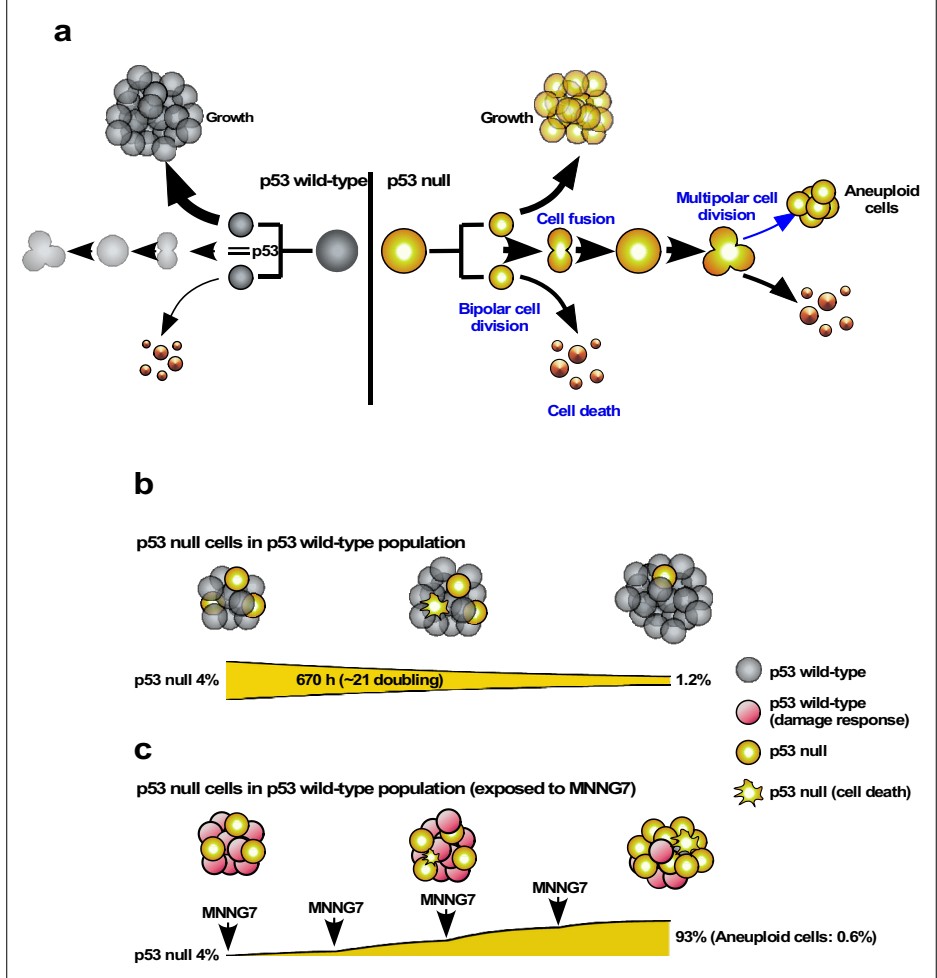

**Figure 14.** Role of the low levels of p53 in suppression of multipolar cell division, cell death, cell fusion, and damage response. (**a**) Characteristics of Control and p53 RNAi cells. (**b**) The simulation of the fate of p53 RNAi cells in the Control cell population. (**c**) The simulation of the fate of p53 RNAi cells in the Control cell population, which were virtually exposed to 7 µM MNNG. The percentage of aneuploid cells is also shown.

p53 are required for housekeeping and homeostasis functions in cells. In addition, Vousden and Lane pointed out that the low levels of p53 may play a role in responding to 'daily levels of stress' (*Vousden and Lane, 2007*), and Valente et al. recently demonstrated that p53 plays a role in responding to stress induced by physiological levels of oxygen tension (*Valente et al., 2020*). Furthermore, this study showed that low levels of p53 inhibited the induction of cell death and cell fusion, which could lead to the induction of multipolar cell division, suggesting that the low levels of p53 are involved in processes that ensure the accuracy of cell division.

## Silencing of low levels of p53 and induction of cell death, cell fusion, and multipolar cell division

The different characteristics of Control and p53 RNAi cells are summarized in *Figure 14a*. Most Control cells underwent bipolar cell division, resulting in expansion of the cell population. Although cell death, cell fusion, and multipolar cell division occurred in Control cells, they had relatively little effect on the expansion of the cell population. In contrast, although p53 RNAi cells could expand their cell population through bipolar cell division, p53 silencing led to more frequent induction of cell death and cell fusion.

The cell death events in p53 RNAi cells appeared to be induced by a p53-independent mechanism. Although the mechanism underlying the induction of cell death remains to be investigated, it may involve a natural defense mechanism to remove cells lacking p53 from a cell population. Indeed, we

demonstrated that cell death caused a 7.4% reduction in the rate of p53 RNAi cell population expansion. Cell death thus conferred a disadvantage to p53 RNAi cells to expand their population relative to Control cells. Simulation of the fates of cells in a population containing 4% p53 RNAi and 96% Control cells showed that the proportion of p53 RNAi cells was reduced to 1.2% after 670 hr (28 days) of culture (*Figure 14b*). When such a simulation was initiated with 5000 cells, the cell number reached ~1 × 10$^9$, with a mass equivalent to ~1 cm of cancer tissue, after 21 cell divisions. Within a cell tissue mass, p53 RNAi cells may occur as dormant cells, given that the percentage of p53 RNAi cells in the tissue would be reduced following the increase in tissue size. With regard to dormancy, many metastatic cells are known to become dormant (*Hüsemann et al., 2008*; *Riethmüller and Klein, 2001*), suggesting that their metastatic nature per se could confer a survival disadvantage on these cells relative to normal or non-metastatic cells, as indicated by the difference in expansion rates between p53 RNAi cells in a Control cell population.

Cell fusion was also increased following silencing of p53 (*Figure 14a*). Although the role of p53 in the suppression of cell fusion is unknown, previous reports have suggested a link between the loss of p53 and the induction of cell fusion. For example, fibroblasts derived from p53$^{-/-}$ mice rapidly became tetraploid (*Livingstone et al., 1992*; *Harvey et al., 1993*). Increased cell fusion of p53 RNAi cells, mainly between sibling cells produced by bipolar cell division, could account for the formation of tetraploid cells (*Figure 14a*). Such fused cells may be formed by failure of cytokinesis, mitotic slippage, or the formation of a link between siblings (*Shi and King, 2005*; *Eischen, 2016*; *Ganem et al., 2007*; *Storchova and Kuffer, 2008*). However, low levels of p53 may block the process of fusion itself, given that we also observed cell fusion following abscission and between non-sibling cells. During development, such as myotube formation (*Hernández and Podbilewicz, 2017*; *Shinn-Thomas and Mohler, 2011*), a cell fusion mechanism is activated, implying that cells can acquire a fusion-prone status. In addition to the physiological process of cell fusion, if cells; for example, siblings produced by bipolar cell division, became fusion prone, it could lead to the formation of polyploid cells, which have been suggested to be precursors of cancer cells (*Eischen, 2016*; *Deangelis et al., 1993*; *Dutertre et al., 2005*; *Galipeau et al., 1996*; *Ramel et al., 1995*). The low levels of p53 may thus function to prevent cells from becoming fusion-prone, thereby reducing the risk of cells becoming cancer cells.

Loss of the low levels of p53 increased the induction of multipolar cell division following cell fusion (*Figure 14a*). The division of a cell into three and more cells generates progeny with different numbers of chromosomes. Although such aneuploid cells are generally fragile (*Holland and Cleveland, 2009*; *Gisselsson et al., 2008*), loss of p53 function has been suggested to contribute to the survival of aneuploid cells (*Fujiwara et al., 2005*; *Thompson and Compton, 2008*; *Vitale et al., 2010*). Indeed, we found that some such aneuploid cells underwent bipolar cell division and the progeny could propagate, creating an aneuploid cell population (*Figure 14a*). Even though gene mutation has been proposed as the main cause of cancer, aneuploidy is a near-universal feature of human cancers (*Li and Zhu, 2022*) and is also involved in the process of tumorigenesis (*Boveri, 2008*; *Ganem et al., 2007*). Furthermore, impaired p53 function is known to cause frequent alterations in cell ploidy (*Eischen, 2016*; *Deangelis et al., 1993*; *Dutertre et al., 2005*; *Galipeau et al., 1996*; *Ramel et al., 1995*). The proposed process could thus be involved in the process of tumorigenesis. In summary, the low levels of p53 found in unstressed cells could suppress cell fusion, which leads to multipolar cell division, and loss of this function could thus create conditions favoring the generation of cancer cells.

## Response of p53-silenced cells to MNNG

The above-mentioned context reveals the potential roles of low levels of p53 under unstressed conditions. One of the best-established roles of p53 is related to cellular responses induced following its accumulation under stressed conditions, including its capacity to prevent the survival of damaged cells. This capacity is closely associated with cell growth arrest and cell death. On the other hand, induction of the damage response in Control cells may oppose the suppression of tumor formation. Exposure of a cell population composed of 4% p53 RNAi cells and 96% Control cells to 7 μM MNNG, which could lead to the accumulation of p53, was associated with growth suppression of the Control cell population due to induction of the damage response, conferring an opportunity for expansion of the p53 RNAi cell population, which lacks the p53-mediated damage response function. Indeed, our

simulation results suggested that 4% of p53 RNAi cells present at the beginning expanded to 93% of the cell population (containing 0.6% aneuploid cells) after 40,000 min (27.78 days) if the population was exposed to 7 μM of MNNG every 10,000 min (*Figure 14c*). These results suggest that exposure of tumor tissue composed of p53-proficient and -deficient cells to a reagent that induces a damage response may lead to preferential expansion of the p53-deficient cell population that contains aneuploid cells with reproductive activity. If the deficient cells have a malignant nature, p53 could indirectly promote, rather than suppress the formation of a malignant tumor by suppressing the expansion of the p53-proficient cell population. Understanding the role of p53 in tumorigenesis thus requires the delineation of its role at both the cellular and population levels. Furthermore, because most cancer cells carrying p53 gene mutations express mutated p53 in the cytoplasm instead of the nucleus, cytoplasmic mutated p53 has been suggested to confer a cellular function, referred to as gain of function (*Acin et al., 2011*; *Amelio and Melino, 2020*; *Bargonetti and Prives, 2019*; *Oren and Rotter, 2010*; *van Oijen and Slootweg, 2000*; *Yue et al., 2017*). Such cells may have a combined phenotype of both loss of wild-type p53 and gain of function, or mutated p53 may alter the phenotype caused by the loss of the wild-type p53. Nevertheless, these need to be considered in the context of p53-related tumorigenesis, and the current approach may be used to gain clues to reveal the context.

## Cell-fate simulation and its applications

Compared with other analytical methods, single-cell-tracking analysis has the ability to accurately detect various cellular events, regardless of the frequency of their occurrence, by creating spatio-temporal data for individual cells. However, because single-cell-tracking analysis is a morphological observation-based technique, we were unable to determine the silencing levels of p53 in individual cells. The availability of such information would allow deeper analysis, for example, to examine the relationship between p53-silencing levels and the chance of cell fusion. Although it is difficult to monitor expression levels of p53, or other proteins or genes, in individual live cells using existing methods, new technologies combining single-cell-tracking analysis with spatial transcriptomics (*Hou et al., 2021*; *Joglekar et al., 2021*), for example, could be developed to expand the applicability of this type of analysis. Such expansion could also provide additional flexibility for cell-fate simulations. We created a cell-fate simulation algorithm to allow the flexible design of various types of virtual experiments by creating Operation data, containing data generated by the categorization of event patterns. Such patterns could be further enriched by the addition of information on the expression levels of proteins or genes in individual cells, allowing the development of another new type of cell-fate simulation algorithm. Furthermore, given that each type of cell population tends to show a specific cell population expansion profile and response to treatment, the accumulation of public Operation data resources could be made accessible to various research projects to simulate the fate of cells of interest, predict the responses of cells to treatments, build models based on the simulations, and develop a theory-based model for cell behaviors. Because single-cell-tracking data contains information on the motility of each cell, it may also be possible to simulate cell behaviors that could be influenced by cell-to-cell contact. This simulation-based biology could thus help to overcome some of the limitations associated with empirical cell biological studies and provide greater flexibility in research with mammalian cells.

## Conclusions

p53 may affect cell population dynamics, and loss of p53 could lead to the formation of an aneuploid cell population. In addition, the dynamics of cell populations containing both p53-null cells and wild-type cells may be affected by the status of the wild-type cells. Our results suggest that wild-type cells could gain a growth advantage over p53-null cells in low-stress environments, while the growth balance between the p53-null and wild-type cells may change under stress conditions, allowing p53-null cells to gain a growth advantage. The role of p53 in the process of tumorigenesis may thus depend on the environment surrounding the cell population, as well as the p53 status of individual

cells within the population. Revealing the status of individual cells by single-cell tracking, cell-lineage analysis, and cell-fate simulation can thus further our understanding of the process of tumorigenesis.

# Materials and methods

## Key resources table

| Reagent type (species) or resource | Designation | Source or reference | Identifiers | Additional information |
|---|---|---|---|---|
| Antibody | Mouse anti-p53 (DO-1, mouse monoclonal) | Invitrogen | AHO0152 | 1 µg/mL |
| Antibody | Horseradish peroxidase conjugated goat anti-mouse antibody (goat polyclonal) | Abcam | Ab205719 | 1:25,000 |
| Chemical compound, drug | *N*-methyl-*N′*-nitroso-*N*-nitrosoguanidine | Sigma-Aldrich | 129941 | |
| Cell line (human) | A549-luc-C8 | Xenogen Corporation | N/A | |
| Sequence-based reagent | p53 siRNA I | Cell Signaling Technology | 6231 | |
| Sequence-based reagent | Control siRNA | Cell Signaling Technology | 6568 | |
| Software, algorithm | MetaMorph (Quorum WaveFX, v7.8.12.0) | Quorum Technology | N/A | |
| Software, algorithm | ImageJ | *Schneider et al., 2012* https://imagej.nih.ggov/ij/ | | |
| Software, algorithm | Computerized single-cell lineage tracking analysis system software | Rancourt, Sato, and Satoh | BioRxiv. doi: https://doi.org/10.1101/508705 | |
| Software, algorithm | Cell fate simulation algorithm | This paper | | Results: 'Cell-fate simulation algorithm'; Materials and methods: 'Overall flow of cell-fate simulation algorithm'; *Figure 7—source code 1* |
| Other | Cell lineage data | This paper | | Deposited in Dryad |

## Cells and cell culture

A549-luc-C8 cells were purchased from Xenogen Corporation and cultured in RPMI containing 10% fetal bovine serum (RPMI medium) in a humidified atmosphere with 5% $CO_2$. Cells were confirmed to be mycoplasma negative. Cells were plated in the center of each well of a cover glass Lab-Tek II 8-well chamber in 50 µL of cell suspension containing 3500 cells and left to attach to the cover glass surface. Culture was continued for 24 hr to allow more than 99% of cells attached to other cells, and culture medium (0.75 mL) was then added to each well and the chamber was viewed under a microscope (Olympus IX81) after 24 hr of plating.

## Concurrent long-term live-cell imaging, siRNA transfection, and MNNG treatment

Concurrent long-term live-cell imaging was performed as described previously (*Sato et al., 2018*; *Sato et al., 2016*). Briefly, images were acquired under a microscope using near infrared DIC imaging with a ×20 dry objective (UPlanSApo, ×20/0.75 NA, α/0.17/FN2G.5) and a ×1.5 coupler (Quorum Technologies) to generate a ×30-equivalent image. Images were acquired by area scanning (5 × 5 dimension, 512 × 512 pixels each, 1.77 mm$^2$) using multidimensional acquisition mode in MetaMorph (Quorum Technology Inc, WaveFX, v7.8.12.0) with 34 ms exposure and XY piezo stage. The selected area was close to the center of the cell population, given that the behavior of cells at the periphery may differ from those at the center, mainly due to differences in cell density and the chance of intercellular attachments. Typically, 30 z-planes were acquired every 1 µm, and 512 × 512 pixel multilayered .tiff files were created. Cells were maintained on the microscope stage in an environmental chamber (Live Cell Instrument, Korea) in a humidified atmosphere with 7.5% $CO_2$ and images were acquired every 10 min. Lipofection with scrambled siRNA and p53si RNA (Cell Signaling Technology) was performed approximately 24 hr after cell plating using Effectene Transfection Reagent (QIAGEN) with 1 µg siRNA

in 742 μL transfection mixture, according to the supplier's instructions. Image acquisition was started immediately after siRNA transfection. At 24 hr after siRNA treatment, cells were exposed to various doses of MNNG (Sigma-Aldrich) for 30 min in serum-free RPMI, and the medium was then replaced with fresh RPMI medium. The evaporation rate of the medium was about 10 μL per 24 hr, and each well of a cover glass Lab-Tek II 8-well chamber thus contained 800 μL of RPMI medium to minimize the impact of evaporation on cell growth. The RPMI medium contained phenol red to monitor the pH of the medium, and 7.5% $CO_2$ was typically required to maintain an optimal pH.

Image files corresponding to each field of view were saved using MetaMorph on the hard drive of a Windows computer. The files were then transferred automatically to a Macintosh computer (OS15) using in-house software (*Sato et al., 2018*). Unique file names were assigned to each file for archiving, followed by the creation of an all-in-focus image using in-house software (*Sato et al., 2018*). The focused images were positioned and the contrast was adjusted to create stitched images (25 images covering 5 × 5 dimension, approximately 2500 × 2500 pixels). The images were then saved in chronological order for display. File transferring, focused image creation, image positioning, and contrast adjustment were performed automatically, and eight stitched images that corresponding to each Lab-Tek II 8-well chamber well were typically created concurrently in an automatic manner (*Sato et al., 2018*). The files were displayed using an in-house movie player to monitor the progress of live-cell imaging (*Sato et al., 2018*).

## DIC image segmentation

Image segmentation was performed using our image segmentation software (for details, see *Sato et al., 2018*). Image segmentation can be performed by various methods, such as the region growing, by setting an optimal parameter for an image. However, the optimal parameter for a time point of the image may be difficult to use for other time points, given that the optimal parameter is affected by the cell density and size, the relative location of a cell to other cells, the formation of cell debris, and image quality variations caused by the microscope system. We typically processed >8000 images, and our segmentation software was therefore developed to perform DIC image segmentation automatically, regardless of image quality (*Sato et al., 2018*). Briefly, in a DIC image, cells appear as illuminated objects associated with a shadow, and this rule can be applied to any cells in an image. For example, if a flat cell is located behind a bulky cell, the flat cell appears as a darker cell, but the cell itself is brighter than its shadow. Our segmentation software used these characteristics of the DIC images. The mean pixel value of a DIC image (256 grayscale images) was first adjusted to a grayscale value of 100, and four images were then created by applying four different pixel threshold values: for example, >200, >180, >160, and >150. The image with the highest threshold value was used first, and connected pixels, representing an area, were identified by connectivity analysis. The connected pixels were then overlaid on an original DIC image and the area of connected pixels was extended. The connected pixels were likely to represent the illuminated cells in the image, and expansion was carried out to identify the possible borders of the cells. The connected areas were then overlaid on the image created with the second highest grayscale threshold value. Connectivity analysis was performed, except for the areas where a connected area had already been set in the previous step. This process allowed us to identify darker cells in a DIC image than those identified previously. Connected areas identified in the second step were overlaid on the original DIC image and the area was expanded to find a border of darker cells. The same process was repeated for the images created by the third and fourth thresholds to identify cells in an image. By this approach, DIC images of various qualities could be segmented.

## Single-cell tracking

Single-cell-tracking analysis was performed using in-house software that tracked cells automatically, detected cellular events, and allowed verification of the tracking results (*Sato et al., 2018*). Single-cell tracking was started by selecting a connected area (Area) that corresponded to a progenitor. Because the segmentation pattern of the next image was often changed, automatic cell tracking was performed by identifying the best-matched segmented area in the next image. Briefly, Area was overlaid on the connected areas in the next time point image. If Area overlapped with one area in the next image and the size of the area was within the twofold of Area, the area in the next image was determined as the area that corresponded to the cell that was being tracked. If Area overlapped with one

area in the next image, but the size of the area was larger than twice Area , the area was likely divided into multiple areas and one area that overlapped with Area was determined as the area corresponding to the tracked cell. If multiple areas in the next image overlapped with Area , the multiple areas were likely merged and the merged area was determined as the area corresponding to the tracked cell. This process was carried out for all selected progenitors and repeated until the predetermined time point. When the tracking reached the target time point, manual data verification was performed to correct tracking errors. Detected errors were corrected manually, and automatic single-cell tracking was restarted. Typically, verification was performed every 100 time points to create error-free cell-lineage data. The required time for verification was ~48 hr for the tracking of 400 Control progenitors and their progeny for 400 time points. Concerning event detection, different approaches were used for cell division, cell fusion, and cell death. Bipolar and multipolar cell division were detected by first identifying a mitotic round cell. Typically, such cells appeared as bright round objects in DIC images. The in-house cell-tracking software thus built a data library of the shapes of mitotic cells and determined if a tracked cell underwent mitosis. During the manual verification step, if an error, such as the detection of a non-mitotic cell as a mitotic one or failure to detect a mitotic cell, was found, the failed or missed pattern was incorporated into the software to improve the subsequent detection accuracy. Following the detection of mitotic cells, if a connected area overlapped with two, three, and four connected areas in the next image, the software determined that bipolar, tripolar, and tetrapolar cell division, respectively, had occurred. Concerning cell fusion, if two connected areas corresponding to two different cells overlapped with the same connected area in the next image, and the area did not divide into multiple areas in the next 10–20-time points, the software determined that two cells were fused. In the case of cell death, dead cells typically formed remnants that appeared as bright, irregularly shaped objects. If a connected area overlapped with an area corresponding to such an object, and the object was also found in the next 10–20 time point images, the software determined that cell death had occurred.

Automated cell tracking and manual verification can be used to generate an error-free cell-lineage database. Furthermore, to ensure complete cell-lineage data, if any progeny of a cell lineage moved outside of the field of view, all the data corresponding to that lineage were excluded from the analyses. During single-cell tracking, information on cell-lineage number, cell number, the coordinates of the cells, the types of an events occurring in the cell, the relationship of a cell to its parent cell, and information related to fused cells were recorded in the cell-lineage database (*Sato et al., 2018*).

## Western blotting

Cells were transfected with scrambled siRNA and p53 siRNA and treated with various doses of MNNG after 24 hr. At 48 hr after treatment, the cells were harvested and proteins (15 μg) were separated by 10% sodium dodecyl sulfate-polyacrylamide gel electrophoresis. Western blotting was performed using anti-p53 antibody (1 μg/mL, DO-1; Invitrogen) and horseradish peroxidase-conjugated goat anti-mouse antibody (25,000-fold dilution; Abcam). Proteins were visualized using an ECL reagent (Pierce). Quantitation of p53 was performed using ImageJ.

## Generation of density maps

To create a cell density map, we assigned a value of 1 to a pixel within the 20-pixel diameter area for cell density and 100-pixel diameter area for multipolar cell division and cell death density from the position of the cell or position where multipolar cell division or cell death occurred. If an area overlapped with other areas, a pixel was assigned the sum of the number of overlapped areas. These values were used to generate heat maps.

## Overall flow of the cell-fate simulation algorithm

The flow of the cell-fate simulation algorithm is shown in *Figure 7—figure supplement 1*. The simulation mode was selected from the available five modes; Standard, Dose simulation, Mix culture, Switch, and Mix culture-Switch (*Figure 7—figure supplement 1*: 1 and *Figure 8*). Standard mode performed a simulation by setting a simulation time (Sim. Time end) and an initial number of cells (Init. No. of cells; progenitors) followed by Operation data creation (*Figure 7—figure supplement 1*: 2, and *Figure 7—figure supplement 2*) or uploading, and generation of simulation arrays (*Figure 7—figure supplement 1*: 3, and *Figure 7—figure supplement 2*). Operation data held the categorized information of

events, and Simulation arrays (*Figure 7—figure supplement 1*: 3) were used to assign the length of time between two events (Ltime) and event type. Dose–response mode was carried out by setting a simulation time, the initial number of cells, and an intermediate dose. Two sets of Operation data were then generated or uploaded and virtual Operation data corresponding to the intermediate dose were created using those Operation data. For example, if the intermediate dose was 5 μM MNNG, virtual Operation data were generated from Operation data for 3 μM and 7 μM MNNG. Mixed culture mode performed a simulation for a virtual cell population composed of two cell types. A simulation time and initial numbers of the first and second cell populations were set. Two sets of Operation data were used to simulate the proliferation of the first and second cell populations individually. Two sets of simulation arrays were also created. This mode could be used to simulate the proliferation of a cell population surrounded by another cell population. Switch mode was used to perform virtual drug treatment. The simulation times before and after treatment, and the initial number of cells were set. Two sets of Operation data for the first and second doses were used, and simulation arrays corresponding to each dose were created. Mixed culture-Switch mode was a combined version of Mixed culture and Switch. This mode used four sets of Operation data and generated four sets of simulation arrays. Initial processing was then performed by creating a cell data information array (CDI) that included the Ltime and event type assigned to each cell (, *Figure 7—figure supplement 1*: 4, and *Figure 7—figure supplement 4a*). Based on the information in the CDI, a virtual cell-lineage database was created (*Figure 7—figure supplement 1*, *Figure 7—figure supplement 1*: 5, and *Figure 7—figure supplement 4b*). After these initial processes, a repeated cycle of assigning Ltime and event type (*Figure 7—figure supplement 5*), reassigning Ltime and event type to adjust for the effect of cell fusion on Ltime (*Figure 7—figure supplement 6a*), and creating a virtual cell-lineage database (*Figure 7—figure supplement 6b*) were carried out until the simulation time was reached (*Figure 7—figure supplement 1*: 6). If Switch or Mixed culture-Switch mode was used, the cell-lineage database end was trimmed (*Figure 7—figure supplement 6c*), Operation data was switched to corresponding Operation data, and the process proceeded to the second round of the repeated cycle (*Figure 7—figure supplement 1*: 7). When processing reached the simulation time, the simulation was terminated and the results were displayed.

## Cell-fate simulation algorithm: Operation data

Processing related to Operation data, which comprised three types of data, is outlined in *Figure 7—figure supplement 2*. Operation data were either uploaded or generated from a cell-lineage database. When the Operation data were generated, information held in the cell-lineage database was converted into an Operation data-Time (*Figure 7—figure supplement 2*: 1), Operation data-Event (*Figure 7—figure supplement 2*: 2), and Recovery data (*Figure 7—figure supplement 2*: 3). The Operation data-Time held categorized Ltime by event type; for example, Ltime between bipolar cell division and cell death (BD-CD data type) and Ltime from time 0 min to the time that an event occurred in a progenitor (First event data type) (*Figure 7—figure supplement 2*: 1). The Operation data-Event contained the frequency of events occurred following, for example, bipolar cell division (BD data type), or multipolar cell division (MD data type) (*Figure 7—figure supplement 2*: 2). Recovery data were used to simulate the rate of recovery of a cell population following treatment with MNNG. Most cells adopted a static status following treatment with MNNG; however, some cells regained their reproductive ability, as shown at the end of the imaging procedure. Typically, such cells underwent bipolar cell division to produce two offspring. However, live-cell imaging would be terminated before the offspring underwent cell division, as a culture of reference, nontreated cells become confluent. In this case, it was not possible to determine the cell-doubling time of the offspring. Recovery thus determined the percentage of cells that underwent bipolar cell division over a certain period from the end of the tracking time. For example, if the tracking time was 4000 min, the percentage of cells was calculated by the number of bipolar cell divisions that occurred within; for example, 3200–4000 min, and the number was converted into a percentage by dividing the total number of bipolar cell divisions that occurred during the tracking time and then multiplying by 100 (*Figure 7—figure supplement 2*; 3). Thus, if 10 bipolar cell divisions were found between 3200 and 4000 min out of a total of 300 bipolar cell divisions (entire time), the result was 3.3%. The simulation algorithm assigned bipolar cell division to 3.3% of nondividing cells at around 4000 min. Ltime, which was >80% of tracking time, was then assigned to cells produced by bipolar cell division. Using Dose–response mode, virtual Operation data were created from two sets of Operation data using the formula: virtual Operation data =

Operation data 1 + [(Operation data 2 − Operation data 1)/(Dose 2 −Dose 1)] × (Intermediate dose − Dose 1) (*Figure 7—figure supplement 2*: 4).

### Cell-fate simulation algorithm: Simulation arrays

The dataset stored in Operation data was then converted into Simulation arrays that were used to assign Ltime and event types (*Figure 7—figure supplement 3*). In the Operation data, each categorized Operation data-Time was stored in the format, that is, Ltime, and the number of Ltime that was found. For example, 20 min of Ltime found four times was written as 20:4. In the Simulation arrays corresponding to the Operation data-Time, this was converted to 20, 20, 20, 20, for example. If no corresponding type of event occurred, the array for the event held no data. To generate a simulation array from the Operation data-Event, the percentage of each event relative to the total number of cell divisions (total of bipolar cell division and multipolar cell division) was calculated. The maximum number of entries per array was 100. The total percentage of events could thus exceed 100, for example, first event list (bipolar cell division, cell death, multipolar cell division, and the number of cells with no cell division [nonDiv]). In this case, the percentage of each event was adjusted to make a total of 100. If the percentage of the event was <100, 0 was filled to make the total entry for the array 100. If no corresponding type of event occurred, an array for the event held no data. The Recovery percentage was calculated as described above.

### Cell-fate simulation algorithm: Initial Ltime and event-type assignment

In the first event assignment (*Figure 7—figure supplement 4a*), the CDI that held the status of each cell throughout a simulation was created. In this step, the cell-lineage number was generated following the initial number of cells set previously, and the cell number 0 was assigned to each progenitor. Using the FirstEvent array, Ltime was then assigned to each cell. If the array was empty, the average time that bipolar cell division occurred was calculated and −25% to +25% of the time selected by random number (C++ rand ()) was assigned as Ltime, and bipolar cell division was then assigned to the cell. If the FirstEvent array held data, Ltime was assigned by creating a random number, but if Ltime was equal to the length of single-cell tracking, −30% to +30% of the length of the time of single-cell tracking was assigned as Ltime and the event type was selected using the FirstEvent list array. If there was no entry in the array, NonDiv was assigned; otherwise, bipolar cell division, multipolar cell division, or cell death was assigned. If cell death was assigned, Ltime was reassigned using the NonDivCD time data array.

### Cell-fate simulation algorithm: Initial cell-lineage database creation

The cell-lineage database was created using information held by the CDI (*Figure 7—figure supplement 4b*). This array contained information regarding Ltime and event type as a blueprint of the virtual cells (*Figure 7—figure supplement 4b*). If Ltime exceeded the simulation time set previously, Ltime was adjusted to a time equal to the simulation time; otherwise, a cell-lineage database was generated composed of an X position, Y position, time point, event type, cell-lineage number, cell number, parent cell information, cell number, and the cell-lineage number of the cell that was fused in the event of cell fusion. If the event type was bipolar cell division or multipolar cell division, information related to the progeny created by bipolar cell division or multipolar cell division was entered into the CDI following the generation of corresponding cell numbers for each progeny.

### Cell-fate simulation algorithm: Ltime and event-type assignment

In the repeated assignment cycle (*Figure 7—figure supplement 1*: 6), the next Ltime and event type was determined based on the current event type. If the current event in a cell was NonDiv (*Figure 7—figure supplement 5*), a randomized number between 0 and 100 was generated, and if the number was lower than the Recovery %, bipolar cell division was assigned to the cell. In the next cycle, this cell was entered into a growing cycle. If the Ltime was 80–100% of the length of the single-cell-tracking time, the Ltime was determined by generating random numbers between 80 and 100. Assignment of bipolar cell division in this manner only occurred once in the first cycle of the assignment. If the random value was higher than the Recovery %, either NonDiv or cell death was assigned, in which case the % of cell death of total cell division was calculated and, if the random number between 0 and 100 was lower than the cell death %, a cell death was assigned. In this case, Ltime was assigned using

the NonDivCD time array. If the random number between 0 and 100 was higher than the cell death %, NonDiv was assigned, and the length of single-cell-tracking time was assigned as Ltime.

Next, if the current event was bipolar cell division, information related to its sibling was searched for. If a sibling was found, either bipolar cell division, multipolar cell division, or cell death was assigned using the Bipolar cell division list array; otherwise, bipolar cell division, multipolar cell division, cell fusion, or cell death was assigned. If the event assigned by the Bipolar cell division list array was cell fusion, then its sibling was again searched. If no sibling was found, Ltime was assigned by the BDCF time array. If the array was empty, the event type was changed to cell death and the average time that cell death occurred was set as Ltime. If a sibling was found, Ltime was assigned using the BDCF time array. If the array was empty, cell fusion was changed to cell death and the average cell death time was set as Ltime. If Ltime was set, but it was longer than the Ltime of its sibling, Ltime was made shorter than the sibling's Ltime. If the event assigned using the Bipolar cell division list array was bipolar cell division, Ltime was assigned by the BDBD time array. If the array was empty, the average bipolar cell division time was set as Ltime. If Ltime was set by the BDBD time array but the Ltime was not within −10% to +10% of Ltime of its parent cell, Ltime was reselected until the Ltime fell within this range. If Ltime assigned by the Bipolar cell division list array was multipolar division, Ltime was assigned by the BDBD time array. If the array was empty, the average bipolar cell division time was set as Ltime. If the event assigned by the Bipolar cell division list array was cell death, Ltime was assigned the BDCD time array. If the array was empty, the average cell death time was set as Ltime.

If the current event was multipolar cell division, information related to its siblings was searched for. If siblings were found and cell fusion was assigned to two siblings, either bipolar cell division, multipolar cell division, or cell death was assigned by the Multipolar cell division list array, otherwise bipolar cell division, multipolar cell division, cell fusion, or cell death was assigned. If the event assigned by the Multipolar cell division list array was cell fusion, then, event type of its sibling was again searched. If no sibling was found, Ltime was assigned by the MDCF time array. If the array was empty, the event type was changed to cell death, and the average time that cell death occurred was set as Ltime. If only one sibling was found, Ltime was assigned by the MDCF time array. If the array was empty, cell fusion was changed to cell death and the average cell death time was set as Ltime. If two siblings were found and cell fusion was assigned to one of the siblings, Ltime was assigned by the MDCF time array. If the array was empty, cell fusion was changed to cell death and the average cell death time was set as Ltime. If Ltime was longer than the Ltime of its siblings, Ltime was made shorter than its siblings. If cell fusion was not assigned to any of its siblings, Ltime was assigned by the MDCF time array. If the array was empty, cell fusion was changed to cell death and the average cell death time was set as Ltime. If the event assigned by the Multipolar cell division list array was bipolar cell division, Ltime was assigned by the MDMD time array. If the array was empty, the average bipolar cell division time was set as Ltime. If Ltime assigned by the Multipolar cell division list array was multipolar cell division, Ltime was assigned by the MDMD time array. If the array was empty, the average bipolar cell division time was set as Ltime. If the event assigned by the Multipolar cell division list array was cell death, Ltime was assigned by the MDCD time array. If the array was empty, the average cell death time was set as Ltime.

## Cell-fate simulation algorithm: Readjustment of Ltime in cells that underwent cell fusion

If cell fusion occurred in a cell, the Ltime of the cell being fused was often affected and the Ltime was readjusted (*Figure 7—figure supplement 6a*). Fusion between non-siblings was not simulated by this algorithm. If the current event was bipolar cell division, the next event was assigned by the BDCF lists. If the next event was bipolar cell division or multipolar cell division, Ltime was assigned by the DivCFDiv time array, and if the next event was cell death, Ltime was assigned by the BDCFCD array. If the current event was multipolar cell division, the next event was assigned by the MDCF lists. If the next event was bipolar cell division or multipolar cell division, Ltime was assigned by the DivCFDiv time array, and if the next event was cell death, Ltime was assigned by the MDCFCD array.

## Cell-fate simulation algorithm: Cell-lineage database creation

The cell-lineage database was created using information held by the CDI (*Figure 7—figure supplement 6b*). If Ltime exceeded the simulation time set previously, Ltime was adjusted to a time equal to

the simulation time. If the event type was bipolar cell division or multipolar cell division, information related to the progeny created by bipolar cell division or multipolar cell division was entered into the CDI following the generation of corresponding cell numbers for each progeny. If the event type was cell fusion, cells to be fused were searched and relevant information regarding cell-lineage number and cell number was recorded for the cell.

### Cell-fate simulation algorithm: Trimming the end of the cell-lineage database

In Switch and Mixed culture-Switch modes, the end of the cell-lineage data was trimmed to the nearest cell division event (*Figure 7—figure supplement 6c*) before entering the second repeated cycle of assignment.

### Statistical analysis

Statistical analyses were performed using Prism 8.

### Naming rule for cells and Operation data

A549 cells treated with scrambled siRNA and p53 siRNA are referred to as Control and p53 RNAi cells, respectively. Those cells treated with MNNG are named by adding, for example, Control-MNNG1 (cells treated with 1 μM of MNNG). If a cell population that is generated by cell fate simulation is referred, -Sim was added, for example, Control-Sim and Control-MNNG1-Sim. In the case that cells generated in silico to remove cell death are referred, cells are named, for example, Control-Silico(-)cell death. If an Operation data-Time or Operation data-Events are required to be specified, the name of cells was added following Operation data, for example, Operation data-Control.

## Acknowledgements

We thank Dr. Amélie Fradet-Turcotte for critical reading of the manuscript. We like to acknowledge the Bioimaging Platform of Research Centre for Infectious Diseases, CHU de Québec Research Centre for the technical support of microscopes. This work was supported by the Canadian Foundation for Innovation and Canadian Institutes for Health Research.

## Additional information

### Funding

| Funder | Grant reference number | Author |
|---|---|---|
| Canadian Institutes of Health Research | Operating grant | Masahiko S Satoh Sachiko Sato |
| Canada Foundation for Innovation | Equipment Grant | Sachiko Sato |

The funders had no role in study design, data collection and interpretation, or the decision to submit the work for publication.

### Author contributions

Ann Rancourt, Data curation, Formal analysis, Validation, Investigation, Visualization, Methodology, Writing – original draft; Sachiko Sato, Conceptualization, Supervision, Funding acquisition, Validation, Investigation, Visualization, Methodology, Writing – original draft, Project administration, Writing – review and editing; Masahiko S Satoh, Conceptualization, Resources, Data curation, Software, Formal analysis, Supervision, Funding acquisition, Validation, Investigation, Visualization, Methodology, Writing – original draft, Project administration, Writing – review and editing

### Author ORCIDs

Sachiko Sato http://orcid.org/0000-0002-5960-1703
Masahiko S Satoh http://orcid.org/0000-0002-0461-2296

Decision letter and Author response
Decision letter https://doi.org/10.7554/eLife.72498.sa1
Author response https://doi.org/10.7554/eLife.72498.sa2

## Additional files

### Supplementary files
• Transparent reporting form

### Data availability
All data generated or analyzed during this study are included in the paper and supporting file; Source Data files have been provided for Figure 1-figure supplements 2—4, Figures 2, Figures 3, Figures 4, Figure 4—figure supplement 1, Figures 5, Figures 6, Figure 7—figure supplements 1–7, Figure 8— figure supplement 1, Figure 9, Figure 9—figure supplement 1 and 2, and Figures 10–13. Source code has been provided for Figure 7. Figure 1—videos (cellular events), Figure 2—figure supplements (cell-lineage maps), Figure 2—videos (single-cell tracking), Figure 2—source data (cell-lineage database), and Figure 7—figure supplement 7 (cell-lineage maps) have been deposited in Dryad (https://doi.org/10.5061/dryad.pk0p2ngp5).

The following dataset was generated:

| Author(s) | Year | Dataset title | Dataset URL | Database and Identifier |
| --- | --- | --- | --- | --- |
| Rancourt A | 2022 | Empirical single-cell tracking and cell-fate simulation reveal dual roles of p53 in tumor suppression | https://doi.org/10.5061/dryad.pk0p2ngp5 | Dryad Digital Repository, 10.5061/dryad.pk0p2ngp5 |

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
