## [Editor Report]

This article examines the role of p53 in cell division by using a combination of live-cell imaging, cell tracking, and simulations. Overall, the results are extensively and transparently documented and are of interest to cell biologists studying cell division, cell death, and p53.

---

## [Decision Letter]

**Decision letter after peer review:**

Thank you for submitting your article "Empirical single-cell tracking and cell-fate simulation reveal dual roles of p53 in tumor suppression" for consideration by *eLife*. Your article has been reviewed by 3 peer reviewers, and the evaluation has been overseen by a Reviewing Editor and Aleksandra Walczak as the Senior Editor. The following individuals involved in review of your submission have agreed to reveal their identity: Jeroen S van Zon (Reviewer #1); Brian E Chen (Reviewer #2); Colarusso Pina (Reviewer #3).

Essential revisions:

1) The manuscript needs major rewriting to make it more comprehensible to a general audience. Please rewrite the manuscript according to the specific recommendations from the three peer reviewers.

2) Please provide better explanation of the simulation algorithm and the assumptions behind it in plain language.

3) Please revise the figures according to the specific recommendations from the reviewers.

4) Please better discuss the general impact of this study in the Discussion section.

*Reviewer #1 (Recommendations for the authors):*

1) Limited lineage analysis.

In general, the combination of cell tracking and manual annotation of events in general is powerful and I think the analysis of cell fusion in Figure 6 is a nice example of this power. However, I think it is under utilised in the paper. Figures 2 and 4 could have been constructed without cell tracking and lineage analysis, but only by analysing total cell number (Figure 2) or by counting events without lineage information (Figure 4). In Figure 3, lineage structures are analysed (but in a way I don't understand, see further below) but apart from changes with increasing MNNG it doesn't provide much insight: it seems that cell proliferation/death is decreased/increased with stress (as expected) leading to fewer large lineages, with no compelling difference between p53 siRNA or control. There is a sentence ('For example, suppression … generate 10-12 progeny (Figure 3b)’) that suggests there is a difference, but I can't see it in the figure.

It also seems that for a single condition, not all lineages have the same number of progeny (e.g 1-3 vs 13-15). Why is it that for the same imaging time lineages have such different offspring? Is it because some cells die? Is it impacted by limits to cell tracking, e.g. if a cell moves out of the field of view it gives rise to a short lineage because the offspring cannot be followed? In the latter case, that would rather represent a technical limitation rather than biological insight.

2) Insight in simulation lacking.

The section 'Cell-fate simulation algorithm' gives information on highest-level procedure of simulating, but it lacks any discussion of assumptions: does each cell choose 'fate' (MD, CD, CF, etc.) independently, or are correlation between sisters, cousins, etc. taken into account? How is experimental data used to constrain parameters of the simulation? The explanation in the supplementary information was not understandable for me. The presentation of the simulation algorithm in Figure S7a-h seems too detailed (e.g. dealing with the choice to load a file or not at the top of Figure S7b), while some of the diagrams (e.g. Figure S7F) are so complex that it appears to me as if a coherent underlying model is lacking. At the very least, it made it impossible for me to check the validity of the underlying assumptions.

3) In silico data generation.

in Figure 5, in silico data is generated based on experimental data, to asses the role of cell death. However, the explanation raises questions. It says that for any cell that dies its lineage was replaced by that of its sibling. But this only works if lineages without any cell death are always symmetric between siblings. Is this indeed the case? Or can you find cases where one sister generates 4 daughter but the other only 2? In that case, it is not clear what the in silico data represent. This is not discussed anywhere

4) Space in simulations.

In the simulations in Figure 9-11, space is taken into account, but I could not find how this is incorporated in the simulations. Also, it is not clear why space is relevant to begin with. It seems the cells are not directly interacting, e.g. by signalling, so could these results not have been obtained by simulations without space? Or is there competition for space? And if cells fuse, do they fuse with a neighbor? This should be explicitly discussed in the paper.

5) Writing and terms unclear.

In many places, explanations are not clear (see below for examples). The authors use a lot of acronyms, which I partly understand, but some sentences become very hard to understand (e.g. p.33 'we asked how progeny of MD that undergo BD survive by performing a simulation in Std mode'). Captions are strangely formatted and seem to sometimes miss panels (e.g. (b) lacking in caption Figure 3, (c) + (d) lacking in Figure 8).

*Reviewer #2 (Recommendations for the authors):*

Overall the study is very well done. There are several areas that can be better clarified.

The sentence, "However, the function of maintaining low levels of p53 in unstressed cells remains unclear." Is not a great introduction to the rationale of the paper; in that maintaining low levels is not what is being investigated, nor the function of the low levels, nor really the differences between low and high levels.

It is not clear throughout the text what "cell-lineage datasets" means. This could mean many things to many different readers. A sentence as to how the authors will define it is very important. Datasets can mean anything.

Similarly, the quantification of lineage is also ambiguous. How is this defined/quantified? This is never stated in the Methods, and again can mean different things.

The entire section on "Minimum number of cell-lineage datasets required to build a cell-lineage database" is unclear. Is this a random subsampling analysis that is being performed? Are these from experimental triplicates in Supplementary Figure 3, or just a randomly subsampling, as is implied in the statement, "The analysis was repeated three times (green, orange, and red lines) to show variations at a tracking time," so only the analysis was repeated three times, but not the experiments? What is the relationship between all of the datasets starting at 100 cells in the y-axis, but having different numbers of cell lineages?

Quantification of the cell lineage categories was unclear. What does "7-9 progeny" mean? The "number of cells/lineage" (text within the figure) vs "number of progeny/lineage" (main text) doesn't make sense. This is further confused by the Figure 3 legend stating "Values shown in (a) are number of lineages". So if the bar graph is read, then 136 cell lineages had around 350 cells, and 127 cell lineages had 900 cells. This is extremely confusing.

Why are cell densities in control and p53 RNAi in Figure 2 always the same, if MD, CD, and CF are all highly induced after RNAi knockdown?

The normalization factor, "/300 lineages" is not clear. This makes it difficult to compare the rates of the MD, CD, and CF between experiments across the paper. For the numbers "0.0475, 0.125, and 0.0775 events per 10 min/300 lineages" to mean anything substantial to any reader, it should be in a metric that is meaningful.

It is not clear what the numbers in Figure 6a mean. Here, again it is difficult to compare what a low frequency of occurrence of "3.5 to 3.8% of progeny", or "0.7 progeny of MD/100 cell lineage of p53si cells vs. 0.05 progeny of MD/100 cell lineages of Control cells" means when they are normalized to /100 cell lineage.

In the extensive simulations sections, it is not clear that many of them are necessary as main figures (e.g., Figure 8-11). For example, is a simulation necessary to show that extremely high levels in vitro of 7uM MNNG severely sickens the cells? "we estimated that 63.1% and 36.9% of cell growth inhibition … were due to the direct effect of MNNG and damage response, respectively" is very precise for a simulation, but is confounded by biology: off target effects of RNAi, imprecise knockdown of p53 (as the western blot shows), etc.

Overexpression of p53 protein causes senescence, and the paper shows very nicely that p53 is involved in both cell proliferation and cell death. However, it is not clear what the implications are from this and why this is important.

On pg. 35 the use of the word suppressed/suppression and in the rest of the paper implies an active cellular mechanism. Here, it is simply that they are outgrowing or reproducing faster to represent a larger proportion of the population. In the methods on the algorithm, there is nothing that has the two populations interacting, and the two arrays can be "physically" separate in a visualized image and still see the same results. Thus, to state that "p53-silenced cells started to expand their population over that of the p53-expressing cells, " and "allowed the expansion of the p53-silenced cells," and "the cell population was replaced with p53-silenced cells," are all implying physical interactions that are all not really occurring and just two population rates that are simulated and then normalized to 100%. Other statements like, "However, the suppression became less prominent … and p53-silenced cells started to expand their population over that of the p53-expressing cells" are problematic and imply an active biological simulation of physically and chemically interacting cells that is not the case.

There is substantial use of jargon and unnecessary abbreviations in the paper. This is not a minor point and severely affects comprehension and readability. In silico experiments are all simulated by computer. Why the need to keep track of p53Si-Sil(-CD)-Sim? Of BD, CD, MD, and CF; the terms BD, CD, and MD are not related, CD means cell death, CF is also not related to CD, but rather the opposite of BD. On pg. 22, the abbreviations are completely unnecessary, and the information of the abbreviations to be memorized by the reader do not come up until much later in the paper. Event is abbreviated Evt, why is this necessary to get rid of 2 letters, "en"? The abbreviation Evt is never again used until the Methods, thus it is only a methodological importance. I understand as a programmer Event and event type are coded as separate variables containing different data, but for the reader this is irrelevant and superfluous. Similarly, "Among the five simulation modes, Standard mode (Std) …" all of these abbreviations Std, Ind, MX, SS, MCS, in a search function of the PDF, are used only 2-3 times in the main text of the paper.

*Reviewer #3 (Recommendations for the authors):*

Recommendations for improvement

The paper illustrates the advantages and pitfalls of interdisciplinary work, here cell biology and computational approaches. A more pedagogical approach would have made the paper's rationale clearer because some of the approaches, including the DIC image tracking and the cell simulation modelling are not part of the standard biologist's toolkit.

The paper would be greatly strengthened if the authors could summarize the tracking and simulations through pseudocode or a plain language description of the algorithm used for simulation. That way, the average biologist with or without computational expertise would have a sense of the underlying algorithmic thinking, which is separate from the technical details of the computational implementation. Taking the time to explain the algorithm accessibly yet rigorously, allows a reader new to cell lineage tracking and/or simulations to come away with a better sense of the power of these approaches in cell biology.

The paper also needs more transitions to guide the reader. Unless a reader has a background in this area, it can be hard to keep track of the different testing conditions and visualization of results as they represent a formidable number of combinations. At times reading the paper felt like trying to parse a logical puzzle rather than reading experimental results. In addition, the data sets are rich and varied, additional guidance is required in explaining results as in providing transition sentences that explicitly state the reasoning with a bit more detail.

In terms of the biological question, a statement about the novelty and impact of this work would be useful as it is not clear that the model is indeed testing the effect of homeostatic p53 levels. Tumour cells without p53 are able to avoid cell death and proliferate. It is unclear how having less of a wild-type protein in a cell represents a homeostatic model, distinct from having mutated forms within a cell. Also presenting a general result about homeostatic p53 using one cell line, also derived from a tumour, is questionable.

Additional comments for authors:

1. How is the DIC segmentation and object tracking implemented? Is the code developed by the group and is it available on Github or another public repository? Or is it available commercially? It would be good to document the version of the code. Also Cell Profiler apparently can segment DIC cells, so it would be good to know if the approach used by the authors is novel and/or available through open source, collaboration, purchase etc.

On page 69, this is hard to decipher:

Cell density map created by assigning value 1 to a pixel within the 20-pixel diameter area from the position of a cell (blue); if an area overlapped with other areas, the pixel was assigned the sum of the number of overlapped areas (light green).

Can the authors clarify here as this may also help explain communicate the DIC segmentation/tracking approach better?

2. Although the Figures, including Supplemental Figure 7, describe the simulation steps, a high-level description, such as pseudo-code, is missing. That is, a guide that helps the reader understand the logic and implementation of the simulation and help with the interpretation of the detailed steps given in Supplemental Figure 7.

For example, my interpretation of the algorithm is below. It is highly likely my interpretation in incorrect, yet this gives an indication of how and why pseudo-code would be useful and may point out the parts of the explanation that need more detail.

– All these simulations are based on the empirical tracking data. [Note I am not sure how the authors extrapolated past the time length of the experiment, which means I am missing something important]

– Data sets inputted into the simulation include the location, cell division (BP or MP), cell fusion, or cell death, as well as technical notes that affected the tracking.

– Algorithm starts by loading in cell lineages obtained from experiments randomly and automated tracking/manual correction data.

– Simulations progress by filling in the cell areas based on the cell lineage diagrams, and cells and events are encoded for ready visualization when needed.

– Cells were considered to be the same size throughout[?].

– I am not sure how the lineages are extrapolated past the 4000 min experimental time.

– Also I would explain the reason the simulation is so useful. Here is an example where it may seem so obvious to the authors it needs no mention. Yet sometimes it is better to be explicit. It can seem you are talking down but it is helpful!

With this simulation approach, we test whether the simulation matches the experimental results, and also uncouple the combined effect of cell division, cell death and cell fusion on cell proliferation, which would be difficult to do in an experiment with live cells.

3. Tumour cells without p53 can avoid cell death and proliferate. It is unclear how having less of a wild-type protein in a cell is different from having mutated forms within the cells.

– That is, how does the function differ here compared to a mutation in p53?

– How does reduction in p53 followed by inducing cell damage test the need for basal p53 as a unique pathway in cancer progression?

– In addition, more justification of the choice of A549 cells chosen as the model is required, in light of the statement in the Introduction: Here, we investigated the effects of low levels of p53 on the behavior of cells using empirical single-cell tracking.

What is their endogenous expression of p53 in A549 compared to other p53 proficient cells? How do these levels compare to normal cells, such as those derived from primary culture?

– How does inducing DNA damage test the function of low levels of p53 in unstressed cells? Wouldn't a better model be to compare different cell lines with different endogenous levels of p53 and see how they respond? Or this could be used as a second method to back up the conclusions when you damage silenced cells, it seems that the model is not delineated precisely for studying low levels of p53 on cells. Rather the test is studying the effects of DNA damage on p53 null cells which has been done many times? It would be good to hear more about the rationale here, as it is not clear from reading the paper's arguments.

Detailed questions:

4. Why were the cells monitored at such a high density? Was this choice governed by the biological question and/image analysis algorithm limitations, or some other reason(s)? (to follow the fate of individual cells by monitoring live cells, because the culture eventually became over-confluent).

5. Why were the concentrations of CO2 different in the incubator and experiment? Is it due to maintenance of pH, and if so, it would be good to mention this as not everyone monitors pH when carrying out long-term imaging?

6. The authors mention maintaining the integrity of the cell population in multiple sections, including the summary. How is integrity defined? Is it metabolic, genomic, structural?

[Editors’ note: further revisions were suggested prior to acceptance, as described below.]

Thank you for resubmitting your work entitled "Empirical single-cell tracking and cell-fate simulation reveal dual roles of p53 in tumor suppression" for further consideration by *eLife*. Your revised article has been evaluated by Aleksandra Walczak (Senior Editor) and a Reviewing Editor.

The manuscript has been improved but there are some remaining issues that need to be addressed, as outlined below:

Main issues:

– l. 316-318. The authors suggest that P53 RNAi cells proliferate faster, but have more cell death. Can this not be quantified directly from the data, rather than in the indirect manner pursued here? For instance, one could measure the probability that a cell will divide again, if it doesn't undergo cell death. The prediction would be that this probability is higher for P53 RNAi cells compared to Control.

– l. 391-396: This section is still unclear to me. Why not follow the same approach as shown in Figure 7c for time to MD? Is it because there is a correlation in cell cycle time between generations, i.e. a rapidly-dividing mother has rapidly dividing offspring? This should be explained more clearly.

– l. 418-421. I now understand conceptually what the others do, but from the text and Figure 8 I don't understand how this works in practice. Do the authors linearly interpolate between two Operation data sets? If so, then write that down explicitly. Some things in the text here still make no sense to me. "A start event of 10 and 1", what does this refer to, what sort of event? "The chance of a cell population exposed" A chance of what? Is there some probabilistic process that underlies this?

– l. 489-494. I am not sure I properly understand the origin of shorter doubling time. Is the problem that the cell cycle time is not drawn from a measured distribution (as in Figure 7b) and therefore misses rare but very long cell cycle times?

– l. 523-526. This seems a rather counterintuitive effect: removing the P53 stress response leads to higher cell proliferation when higher stress is applied. I do think it should be emphasized more that this is counterintuitive and perhaps also a possible explanation for this effect should be given.

– l 532-535. It is not clear what the simulations can add here. Is it not possible to use experimental data to measure this directly? After all, the simulations are driven by experimental data, so in principle, there cannot really be anything new from the simulations in that regard.

– l 535. "we assumed" I don't understand what the assumption is based on. Measurements in this paper? Existing biological knowledge?

– l. 550-551. The low frequency of multipolar division is put into the simulations directly from the experimental measurements. So why is this simulation result, which makes exactly the same conclusion, surprising or interesting?

– l. 552. "growth of virtual cells" It is not explained in the main text how the spatial cell dynamics is implemented in the simulations. It is also not explained why space is even important and what novel insights it will bring to incorporate it.

– l. 555-559. It is not clear why showing spatial expansion of MD progeny is important. Couldn't the same insights be gained from purely looking at images? This just means that you get clones of cells that are MD cell progeny, why is it important where they sit in space?

– l. 578. What novel insight does this section bring? The stress-shift experiments show the same conclusion of P53 RNAi outcompeting Control cells as Figure 10. Also, what does the inclusion of space bring to Figure 13a? Would the results in Figure 13b-h be different if space was not included? If not, then I would remove the spatial analysis.

– l. 636. "cancer tissue mass". I don't understand why this result is compared to cancer: most cells are Control (=wild-type?) cells with few P53- cells interspersed, and most of the growth is due to proliferation of Control cells, not P53- cells.

*Reviewer #3:*

Comment 1:

The experimental data and single-cell lineage analysis are important contributions to the field. Although the authors have included more detailed explanations, the additional insights offered by single cell tracking need to be described more clearly and emphasized throughout the text.

For example, when the authors move from reporting on the population level counting to the single cell tracking results (lines 197 -198), the authors should guide the reader by providing a transition between both approaches. Currently, the authors launch right from the population counting results to validation of single cell tracking without building up the need for single cell tracking.

To quote: We then compared the results obtained by cell counting with those obtained by computer-assisted single-cell tracking analysis to verify whether this analysis indeed yields results consistent with the classical counting method.

Before comparing the results, a transition highlighting the need for single-cell analysis is missing as well as the need for comparing both methods is needed. Suggestion: "Thus far, we have described how the different treatments affect cell numbers at the population level. We then analyzed the data sets using single cell tracking. This powerful approach can reveal how the events occurring at the individual cellular level contribute to what is observed at the population level."

Only then move on to the comparative results such as by stating: To test the accuracy and self-consistency between our standard cell counting and single cell tracking, we compared the results obtained with each approach.

Similarly, when the results obtained from single cell tracking add new information, additional context should be provided for the reader. This paper is cognitively challenging to read and interpret, and more explanation is needed to help the reader navigate and appreciate the significance of the results.

Comment 2:

Similar clear emphasis is needed when describing the simulation approaches and their significance. To illustrate: In the section introducing the cell-fate simulation algorithm (lines 363-372), confluency is mentioned as are "limitations of empirical approaches." Rather than stating confluency and the vague term "limitations," the justification needs to be clear from the start. My suggestion is move the explanation found later in the text to this introductory section.

For example, lines 440-442 state: "cell-fate simulation using Operation data allows the creation of various simulation options and provides flexibility for designing virtual experiments that would be difficult to perform empirically."

Rather than letting the reader wait for this statement, I recommend leading in with a similar statement (and dropping the technical reference to Operation data). Suggestion:" Cell-fate simulations are powerful and flexible tools help us model conditions that are not readily accessible by direct imaging, such as mixed cultures and mixed dosages. "

Other comments:

3. Ensure that cell growth is used when referring to cell size not to cell division within the text. The term "grow" is used colloquially in the lab but here should be reserved for area/volume not cell division. For example, see how "growth" is used in Line 125. Here it is confusing as the passage is describing cell area decreasing and then grow is used to mean cell proliferation.

4. What random number generator was used?

5. Figure 1—figure supplement 3. Y axis in last graph has the label "Variation" but the units are missing.

6. When discussing Figure 5, explain how the results

highlight the need for single-cell tracking studies and how this approach complements and enriches the population level studies. Suggestion: "The cell numbers over time are roughly equivalent for both the scr siRNA and the p53 siRNA treated cells, as shown in Figures 2c and 2i. The single cell tracking data, however, reveal differences that are not directly accessible through standard cell counting. Figure 4 reveals that although the overall cell death is higher in the p53 siRNA cells, it is compensated by increased multipolar cell division. Further, the simulations can help unravel the relative rates of multi-polar cell division compared to cell death. As shown in Figure 5, we see that the relative proliferation curves can be simulated in the presence or absence of cell death. By analyzing the individual cellular fates under well-defined conditions, we can then simulate scenarios that are not accessible through our direct imaging, such as analyzing how a population of mixed cells and/or heterogenous treatments will evolve over time. "

7. Line 226. Quote: To this end, we sorted each cell lineage into groups (Figure 3a).

The division into groups introduces a layer of complexity and seems arbitrary considering the small number of possible bins. Why is reporting the binned data through groups better than reporting the numerical value?

8. Lines 282-286. Quote: …the number of cell death events in Control cells determined by counting was higher than that determined by single-cell tracking analysis. This was because the area of single-cell tracking was slightly outside the area where cell death frequently occurred (Figure 4—figure supplement 1c). However, given that such events do not occur with the same probability throughout the field of view, some variation may occur with both the counting and single-cell tracking approaches.

Was this because cells that die move more than cells that do not or is it something specific about the cell imaging chamber with death occurring near the edges more than the centre?

9. I am confused by the reference to the "accuracy of a detecting a low frequency event." Quote: In general, the accuracy of detecting a low-frequency event is lower than that for frequently occurring events, e.g. bipolar cell division…

Are the authors referring to the probability of the event being captured by the imaging system rather than the detection of the event by their software and/or visual scorer? Or do they have evidence that the detection by the software and/or visual scorer is lower for these events compared to bipolar cell division?

10. The authors state in line 260 "We therefore determined if single-cell tracking could detect multipolar cell division, cell death, and cell fusion with adequate accuracy for statistical analysis. To this end, we determined the number of multipolar cell division and cell death events visually by manual counting in videos, and compared the results with those obtained using single-cell tracking analysis (Figure 4). Notably, cell fusion was not included in the counting analysis because it was difficult to detect without single-cell tracking."

Yet when review Figure 4, I am having trouble deciphering how the authors concluded that the single-cell tracking was accurate as the results are not directly compared to visual scoring. Please clarify.

11. Lines 316-318: These results suggest that p53 silencing promoted the reproductive ability of p53 RNAi cells, but this was counteracted by the induction of cell death, resulting in the formation of p53 RNAi cell populations that were smaller than Control cell populations (Figure 2i).

I see a small dip in the p53 RNAi curve compared to the Control curve in Figure 2i, but otherwise they look similar. Please clarify.

12. Line 318 Analysis also shows that the effect of silencing the low levels of p53 cannot be detected without access to the spatiotemporal information on individual cells provided by single-cell tracking.

Although this statement is likely, the counter-example can not be ruled out. Just to list possible alternative approaches that could reveal similarly nuanced effects of silencing p53. One can imagine a scenario where the number of live and dead cells are measured over time, without any single cell tracking. Or it may be possible to model the number of cells over time using an analytical expression that embeds cell division (bipolar, multipolar) as an exponential and cell death (say as a linear function of time). Suggestion-qualify statement: "This analysis also demonstrates single cell tracking provides more nuanced and detailed information about the effects of silencing p53. Here the single cell analysis revealed clear differences in the rates of cell proliferation and cell death among the scr siRNA and p53 siRNA conditions, differences would be obscured and/or missed if limiting the analysis to cell counts at the population level."

13. Line 331: Following cell fusion, Control and p53 RNAi cells demonstrated 79.4% and 93.2% multipolar cell divisions.

Please define the % values-how are they calculated? Also the numbers are listed on the bar graphs in Figure 6- the relative percentages of listed over the bar graphs is confusing in light of the numbers being relative to another 100, the 100 cell lineages. Also it would be helpful to see the spread of the data points illustrated by the bar graph.

14. Staring on line 385, the authors state: The algorithm then reflected this distribution to choose the End event.

– How is the weighted random assignment implemented? Specify the method or methods and justify why the choice is appropriate.

– Similarly, for LT and other random assignments, is the implementation always the same as for the Event assignment?

– Is the additional +/- 10% in ET imposed for BP events based on experimental data obtained here and/or reported elsewhere or there some other reason for constraining the ET for BP events in this way? Further, could this constraint have led to the observation summarized in lines 489-494: On the other hand, the simulation tended to yield an average cell doubling time about 2 h shorter than that determined by single-cell tracking analysis. Given that the algorithm assigned a cell doubling time to each cell by generating a random number with Operation data-Time (Figure 7), a long cell doubling time, e.g. 3,000 min, which occurred less frequently, may be less likely to be assigned, resulting in the generation of a simulated cell population

– When CF occurs, do these events also reflect the frequency of sibling vs non-sibling under the different conditions studied?

15. I have trouble understanding why calculating Recovery is important to the simulation and why it is listed before the other steps. Line 894 states: Recovery data was used to simulate the rate of recovery a cell population from the treatment of MNNG. When cells were treated with MNNG, the majority of cells may be killed. However, small number of cells gained reproductive ability, which could be found at the end of imaging.

I am confused here as I don't see evidence of the majority of cells being killed in the population expansion data and don't understand why they need to be tracked.

Lines 903-907: For example, if any of the progeny derived from a progenitor underwent bipolar cell division at last within 20% of the single-cell tracking period, e.g. 320 to 400 min when tracking was performed for 400 min, this cell lineage was counted as one upon the calculation of the percentage. Thus, if 10 such cell lineages were found out of a total of 300, the recovery percent was 3.3%.

I don't understand this statement-what does percentage refer to? Why is only BP considered and not MP, for example? Any why is calculating this value important to the simulations as illustrated in Figure 7, supplement 1c, Step 3?

16. Line 918-920: To generate a simulation array for the Operation data-Event, the percentage of each event relative to the total number of cell divisions (total of bipolar cell division and multipolar cell division) was calculated.

As shown in Figure In Figure 7, supplement 1 c, in the "Event" sequence I don't understand why calculating the "% event of total cell division" is possible before the simulation arrays are created?

17. Lines 931-933: the array was empty, the average time that bipolar cell division occurred was calculated and −25% to +25% of the time selected by random 933 number was assigned as LT, and bipolar cell division was then assigned to the cell.

Is this based on experimental data and/or another reason?

18. I appreciate the care and effort that went into detailing the algorithmic approach used for the simulations. The logical flow now is much clearer and moves away from being a "black box" to now being reproducible.

Recommendations and questions:

– Figure 7 supplement 2. Recommend placing this before supplement 1 as it presents the overall logic.

– In Figure 8, supplement 1a, (page 94). Change "Uze" to "Use" in Step 6.

– In Figure 7, supplement 1b (page 95), Step 3 defines the nBD but the description is confusing. Clarify the "last time point of cell tracking" – is this simply the LT associated with the event? Also what is the offset and what is its purpose?

– In Figure 7, supplement 2, there is a typo in "Select an e-vv-ent type"

19. Consider revising sub-headings to guide the reader. Revise to express purpose, result, significance etc. These could serve as signposts throughout the paper would improve the paper's accessibility.

20. In future work, consider graphical representations that are not limited to colour as in "burgundy" plot as 8-11% of the male sex is R/G colour blind. Consider using different plots styles as well as colour. I am not asking the figures to be revised for this publication, but ask that the authors consider visually accessible graphics / figure in future publications.

21. Similarly, consider box plots rather than bar graphs for future work.

---

## [Author Response]

Essential revisions:1) The manuscript needs major rewriting to make it more comprehensible to a general audience. Please rewrite the manuscript according to the specific recommendations from the three peer reviewers.

We have revised the entire manuscript to make it more understandable to a wider audience. We have revised the last part of the Introduction, added explanations between the paragraphs to smooth the transitions, and have rewritten the Discussion. We have also included methods related to image segmentation and single-cell tracking. We greatly appreciate the reviewers’ suggestions, which have allowed us to improve the manuscript.

2) Please provide better explanation of the simulation algorithm and the assumptions behind it in plain language.

We have created two sections related to the algorithms and added three figures. We hope that this revision will help the audience to follow the concept behind the development of the algorithm. We have retained details of the algorithms and code from the previous version in the current version.

3) Please revise the figures according to the specific recommendations from the reviewers.

We have revised the figures, following the reviewers’ recommendations.

4) Please better discuss the general impact of this study in the Discussion section.

We have added a general discussion regarding p53, single-cell tracking, and cell-fate simulation. We have also added a discussion regarding the prospective use of cell-fate simulations.

We have revised the manuscript extensively and have therefore assigned consecutive numbers to each comment, and indicated the numbers in the relevant parts of the revised manuscript, in addition to including line (L) and page (P) numbers.

Reviewer #1 (Recommendations for the authors):1) Limited lineage analysis.In general, the combination of cell tracking and manual annotation of events in general is powerful and I think the analysis of cell fusion in Figure 6 is a nice example of this power. However, I think it is under utilised in the paper.

We characterized the events that occurred before multipolar cell division using single-cell tracking data, and consider that the results were important for our conclusion. We have revised the related section “Events leading to multipolar cell division” in the Results (P14, L322).

Figures 2 and 4 could have been constructed without cell tracking and lineage analysis, but only by analysing total cell number (Figure 2) or by counting events without lineage information (Figure 4).

We made these figures to compare the results of counting and single-cell tracking data, given that no such verification experiments have been performed previously. We therefore think that both methods should be included in the same figure, to demonstrate the accuracy of single-cell tracking. We have rewritten the following sections in the Results to clarify the objective of the analyses:

“Effect of silencing the low levels of p53 on cell population expansion, P8, L169” and “Impact of p53-silencing and MNNG treatment on cell death, multipolar cell division, and cell fusion, P12, L255”.

In Figure 3, lineage structures are analysed (but in a way I don't understand, see further below) but apart from changes with increasing MNNG it doesn't provide much insight: it seems that cell proliferation/death is decreased/increased with stress (as expected) leading to fewer large lineages, with no compelling difference between p53 siRNA or control. There is a sentence ('For example, suppression … generate 10-12 progeny (Figure 3b)’) that suggests there is a difference, but I can't see it in the figure.

We agree that the section related to Figure 3 was difficult to follow. We have therefore rewritten this section and reorganized the figure. In this section, we intended to show that a cell population is composed of cells with different reproductive abilities, by analyzing the number of progeny produced from a progenitor. We also wanted to explain that cell expansion or growth curves may not represent the actual situation of the cell population. For example, if a cell population treated with a drug shows a similar expansion curve to a non-treated cell population, it may be concluded that the treatment had no effect. However, the drug may promote cell growth but also induce cell death, resulting in an overall similar cell population expansion curve to a non-treated population. Without analyzing the fates of individual cells, it is difficult to reveal the processes involved in building up a cell population. We believe that the relationship between Control and p53 RNAi cells provides an interesting example highlighting this matter: i.e. both Control and p53 RNAi cells showed similar cell population size, but p53 RNAi cells underwent more frequent cell death. We have rewritten the Results section “Analysis of a cell population at the level of cell-lineage” to clarify our intention (P10, L219).

It also seems that for a single condition, not all lineages have the same number of progeny (e.g 1-3 vs 13-15). Why is it that for the same imaging time lineages have such different offspring? Is it because some cells die? Is it impacted by limits to cell tracking, e.g. if a cell moves out of the field of view it gives rise to a short lineage because the offspring cannot be followed? In the latter case, that would rather represent a technical limitation rather than biological insight.

Regarding the first question: “*It also seems that for a single condition, not all lineages have the same number of progeny (e.g 1-3 vs 13-15). Why is it that for the same imaging time lineages have such different offspring?”,* cell populations are generally composed of cells with different reproductive abilities. Such differences will become evident during growth of the population, and these differences will be reflected in the number of progeny produced by a progenitor. We have reminded this in the first paragraph of the “Analysis of a cell population at the level of celllineage” in the Results (P10, L219).

Regarding the second question, *“Is it because some cells die?”,* the answer is yes, in some cases. This issue has been addressed in the section related to Figure 5 (P13, L296 and P55, L1255).

The third question was “*Is it impacted by limits to cell tracking, e.g. if a cell moves out of the field of view it gives rise to a short lineage because the offspring cannot be followed? In the latter case, that would rather represent a technical limitation rather than biological insight”*. This is not the case. We tracked all progeny until the end of tracking time, and the single-cell tracking data thus contained information on the progenitors and all their progeny. We have mentioned this in “System to investigate the functional implications of maintaining the low levels of p53 in unstressed cells, P5, L101” in the Results, and the technical matter related to single-cell tracking is mentioned in “Single-cell tracking” in the Methods (P38, L832).

2) Insight in simulation lacking.The section 'Cell-fate simulation algorithm' gives information on highest-level procedure of simulating, but it lacks any discussion of assumptions: does each cell choose 'fate' (MD, CD, CF, etc.) independently, or are correlation between sisters, cousins, etc. taken into account?

As noted in our response to Comment 1, we have revised the sections and prepared three additional figures. “Cell-fate simulation algorithm, P16, L363” and “Cell-fate simulation options with Operation data, P18, L413”, as well as Figure 7 (P57, L1289), Figure 8 (P58, L1306), and Figure 7—figure supplement 2 (P65, L1471).

How is experimental data used to constrain parameters of the simulation? The explanation in the supplementary information was not understandable for me. The presentation of the simulation algorithm in Figure S7a-h seems too detailed (e.g. dealing with the choice to load a file or not at the top of Figure S7b), while some of the diagrams (e.g. Figure S7F) are so complex that it appears to me as if a coherent underlying model is lacking. At the very least, it made it impossible for me to check the validity of the underlying assumptions.

This comment is also related to Comment 1, and Dr. Colarusso also raised similar comments related to the algorithms. Figure 7 summarizes how our simulation works and Figure 7—figure supplement 2 includes an outline of the simulation processes. We hope those revisions will make it easier for the audience to follow the processes of the cell-fate simulation. “Cell-fate simulation algorithm, P16, L363” and “Cell-fate simulation options with Operation data, P18, L413”, as well as Figure 7 (P57, L1289), Figure 8 (P58, L1306), and Figure 7—figure supplement 2 (P65, L1471).

3) In silico data generation.in Figure 5, in silico data is generated based on experimental data, to asses the role of cell death. However, the explanation raises questions. It says that for any cell that dies its lineage was replaced by that of its sibling. But this only works if lineages without any cell death are always symmetric between siblings. Is this indeed the case? Or can you find cases where one sister generates 4 daughter but the other only 2? In that case, it is not clear what the in silico data represent. This is not discussed anywhere

Regarding the growth patterns of sibling cells, it is rare for both to show exactly symmetric growth patterns. In response to the question “*Or can you find cases where one sister generates 4 daughters but the other only 2?*”, this could happen. However, in many cases, a cell shows the closest growth pattern to its sibling cell, as both inherit similar characteristics from their parent cell. Thus, if a cell only produces two siblings, its sibling also produces a similar number of offspring. Of course, there are many variations in growth patterns, but we believe that this assumption largely reflects the actual growth pattern of sibling cells, and the assumption was also confirmed using cell-fate simulation (Figure 12). We have emphasized some of these points in “*In silico* generation of a cell-lineage database of p53 RNAip53si cells” in the Results (P13, L296).

4) Space in simulations.In the simulations in Figure 9-11, space is taken into account, but I could not find how this is incorporated in the simulations. Also, it is not clear why space is relevant to begin with. It seems the cells are not directly interacting, e.g. by signalling, so could these results not have been obtained by simulations without space? Or is there competition for space? And if cells fuse, do they fuse with a neighbor? This should be explicitly discussed in the paper.

Dr. Chen made similar comments. The primary purpose of the animation was to analyze a cell of interest. As shown in Figure 7—figure supplement 3, the cell-lineage map created by the simulation was very crowded, and we were therefore unable to detect any unique patterns by simply examining the maps (we usually examined cell-lineage maps to find unique growth patterns). We therefore used software to display the cell of interest. We have explained this in “Survival of the progeny of multipolar cell division” in the Results (P24, L550).

As Dr. van Zon has pointed out, we are also interested in simulating cell-to-cell interactions in the future. Using empirical data for e.g. activation of neutrophils by primarily activated neutrophils, the chain reaction of the activation could be animated by adding a code to the software. This animation could provide a useful research tool (P30, L711).

5) Writing and terms unclear.In many places, explanations are not clear (see below for examples). The authors use a lot of acronyms, which I partly understand, but some sentences become very hard to understand (e.g. p.33 'we asked how progeny of MD that undergo BD survive by performing a simulation in Std mode').

Dr. Chen made similar comments regarding abbreviations. We have reduced the use of abbreviations in the revised version, e.g. BD, MD, CF and CD were spelled out (e.g. P5, L94).

Captions are strangely formatted and seem to sometimes miss panels (e.g. (b) lacking in caption Figure 3, (c) + (d) lacking in Figure 8).

We have revised the Legends for Figures 3 (P55, L1237) and 8 (now Figure 10, P59, L1344).

Reviewer #2 (Recommendations for the authors):Overall the study is very well done. There are several areas that can be better clarified.The sentence, "However, the function of maintaining low levels of p53 in unstressed cells remains unclear." Is not a great introduction to the rationale of the paper; in that maintaining low levels is not what is being investigated, nor the function of the low levels, nor really the differences between low and high levels.

We appreciate this comment and have revised the sentence accordingly (P3, L55).

It is not clear throughout the text what "cell-lineage datasets" means. This could mean many things to many different readers. A sentence as to how the authors will define it is very important. Datasets can mean anything.

We used the term “dataset” to distinguish between the cell-lineage database and a group of data for each cell lineage. However, these terms may be confusing. In the case of the DNA database, each gene is categorized by e.g. gene ID. We have therefore rephrased “dataset” as “cell lineage No”, and have included an explanation in “System to investigate the functional implications of maintaining low levels of p53 in unstressed cells” in the Results (P5, L103).

Similarly, the quantification of lineage is also ambiguous. How is this defined/quantified? This is never stated in the Methods, and again can mean different things.

We appreciate this comment. We agree that the phrase “quantitation of cell lineage” could be difficult to understand. In empirical studies e.g. with western blotting or enzyme assays, the analysis is designed to obtain information on a focus of interest, e.g. expression level of proteins. On the other hand, imaging data contains multidimensional information, e.g. cell shape and position, and events occurring in a cell, and cell-lineage data thus also contains some of this information. We therefore aimed to quantitate specific aspects of the cell lineage, e.g. the number of progeny produced from a progenitor, or time between events. We have added a paragraph, “System to investigate the functional implications of maintaining low levels of p53 in unstressed cells” to the Results to clarify this (P6, L108).

The entire section on "Minimum number of cell-lineage datasets required to build a cell-lineage database" is unclear. Is this a random subsampling analysis that is being performed? Are these from experimental triplicates in Supplementary Figure 3, or just a randomly subsampling, as is implied in the statement, "The analysis was repeated three times (green, orange, and red lines) to show variations at a tracking time," so only the analysis was repeated three times, but not the experiments?

We generated 485 cell lineage data, and selected 50-240 cells lineages randomly from the 485 cell lineages. Each repeated selection was performed such that the same lineage was not selected, except for the selection of 240 lineages. We have mentioned this in the related paragraphs (P7, L134) and in the legend for Supplementary Figure 3 (now Figure 1—figure supplement 3) (P63, L1428).

This analysis aimed to determine the number of cell lineages that needed to be used for singlecell tracking analysis. We therefore repeated the process three times using the cell-lineage database created for the purpose. Regarding variations between experiments or imaging, we have addressed this in the section related to Figure 2.

What is the relationship between all of the datasets starting at 100 cells in the y-axis, but having different numbers of cell lineages?

Because the initial number of cell lineages differed, we normalized the value to 100. We have mentioned this in Supplementary Figure 3 (now Figure 1—figure supplement 3) (P63, L1428).

Quantification of the cell lineage categories was unclear. What does "7-9 progeny" mean? The "number of cells/lineage" (text within the figure) vs "number of progeny/lineage" (main text) doesn't make sense. This is further confused by the Figure 3 legend stating "Values shown in (a) are number of lineages". So if the bar graph is read, then 136 cell lineages had around 350 cells, and 127 cell lineages had 900 cells. This is extremely confusing.

Dr. van Zon also commented on this section. We have therefore rewritten this section and reorganized the figures (P10, L219 and P55, L1237).

Why are cell densities in control and p53 RNAi in Figure 2 always the same, if MD, CD, and CF are all highly induced after RNAi knockdown?

We would like to emphasize this point in this manuscript. Cell population expansion curves or growth curves are often used in cell biological studies to assess the effects e.g. of drugs. However, such curves do not provide any information on the composition of the cell population. For example, if a drug promotes cell growth but also induces cell death, the size of the drugtreated population may appear to be similar to that of a non-treated population. However, singlecell tracking allows the growth profiles of each cell lineage to be analyzed, and can thus reveal the composition of the cell population. We consider that Control and p53 RNAi cells provide an interesting example to explain this matter. We have considered this in the “Analysis of a cell population at the level of cell-lineage, P10, L219” and “*In silico* generation of a cell-lineage database of p53 RNAi cells, P13, L296” sections in the Results.

The normalization factor, "/300 lineages" is not clear. This makes it difficult to compare the rates of the MD, CD, and CF between experiments across the paper. For the numbers "0.0475, 0.125, and 0.0775 events per 10 min/300 lineages" to mean anything substantial to any reader, it should be in a metric that is meaningful.

We realized that we normalized the number of cell-lineages to 100 or 300. We now consistently used 100 cell lineages for normalization (P13, L291).

It is not clear what the numbers in Figure 6a mean. Here, again it is difficult to compare what a low frequency of occurrence of "3.5 to 3.8% of progeny", or "0.7 progeny of MD/100 cell lineage of p53si cells vs. 0.05 progeny of MD/100 cell lineages of Control cells" means when they are normalized to /100 cell lineage.

We have reworked Figure 6 and the related sections. We previously used the terms e.g. CF > MD, etc., but have changed these to Pattern 1 and Pattern 2 in the revised version. Regarding the number in 100 cell lineages, we have included additional explanations in the figure legend. Because the number of multipolar cell divisions in Control cells was low, we searched all the videos (entire image of triplicated videos) to find multipolar cell divisions and then tracked the cells back to their progenitors to determine the events preceding multipolar cell division. To normalize the values, we therefore assumed that multipolar cell division occurred at a similar frequency to that recorded in the cell-lineage database. We have verified the numbers and revised the related sections. We hope that these changes make the manuscript easier to follow. Concerning 0.7 vs 0.05 progeny, this indicates the number of progeny produced by multipolar cell division that can undergo bipolar cell division, normalized to 100 cell lineages as described. The percentage is the percentage of such cells in the total number of progeny produced by multipolar cell division. We have revised “Events leading to multipolar cell division, P14, L322” in the Results, and have also revised the legend for Figure 6 (P56, L1267).

In the extensive simulations sections, it is not clear that many of them are necessary as main figures (e.g., Figure 8-11). For example, is a simulation necessary to show that extremely high levels in vitro of 7uM MNNG severely sickens the cells? "we estimated that 63.1% and 36.9% of cell growth inhibition … were due to the direct effect of MNNG and damage response, respectively" is very precise for a simulation, but is confounded by biology: off target effects of RNAi, imprecise knockdown of p53 (as the western blot shows), etc.

Regarding the comment*, “In the extensive simulations sections, it is not clear that many of them are necessary as main figures (e.g., Figure 8-11, Now 10-12)”,* we have changed the display items. We have removed the still images of the animations from Figures 12 and 13 (but kept the animations).

Regarding Figure 10e, we agree that the p53-mediated and non-mediated responses (e.g. by DNA-break formation) may not be distinguished. However, if a response is induced by exposure of cells to a drug following the accumulation of p53, it is often assumed that such a response is only related to p53, while it is likely to be due to the combined effect of p53 with other processes.

We also understand the point that Dr. Chen raised, as there would be a variation in the degree of silencing between cells, and p53 silencing may induce an effect that cannot be detected by single-cell tracking. However, conclusions that are drawn based on e.g. western blotting, and enzymatic assay (average-based analyses) may also contains ambiguities. As Dr. van Zon also raised similar comments, we responded to this matter in Comment 5.

We would therefore like to keep Figure 10e to make this point. Because the values may include an off-target effect, as mentioned by Dr. Chen, we made a statement about this matter in the text (P23, L539).

Overexpression of p53 protein causes senescence, and the paper shows very nicely that p53 is involved in both cell proliferation and cell death. However, it is not clear what the implications are from this and why this is important.

We have revised the entire Discussion to emphasize the implications of our work. We have included three sections to discuss the function of low levels of p53, the induction of multipolar cell division, cell death, and cell fusion, and the responses of Control and p53 RNAi cells to MNNG, respectively. We hope that these revisions have addressed Dr. Chen’s point (P26, L595).

On pg. 35 the use of the word suppressed/suppression and in the rest of the paper implies an active cellular mechanism. Here, it is simply that they are outgrowing or reproducing faster to represent a larger proportion of the population.

We agree that the mechanism was not “suppression”, because Control cells did not directly suppress the expansion of p53 RNAi cells. The current analysis was based on the relative growth abilities of Control and p53 RNAi cells. We have therefore rephrased this in “Limiting expansion of p53 RNAi p53-silenced cells by Control p53-expressing cells” in the Results (P24, L561).

In the methods on the algorithm, there is nothing that has the two populations interacting, and the two arrays can be "physically" separate in a visualized image and still see the same results. Thus, to state that "p53-silenced cells started to expand their population over that of the p53-expressing cells, " and "allowed the expansion of the p53-silenced cells," and "the cell population was replaced with p53-silenced cells," are all implying physical interactions that are all not really occurring and just two population rates that are simulated and then normalized to 100%. Other statements like, "However, the suppression became less prominent … and p53-silenced cells started to expand their population over that of the p53-expressing cells" are problematic and imply an active biological simulation of physically and chemically interacting cells that is not the case.

As noted in our response to Dr. van Zon (Comment 13), the primary purpose of the animation was to analyze the cell of interest. Because the simulation generates a large number of virtual cells, it is impossible to evaluate the events that occurred in the simulated cells. We therefore made animations to visually show the cells of interest. In the simulation mode with Control and p53 RNAi cells, the results reflected the relative growth abilities of both cell types. We therefore agree that our statement suggesting that the cell types interacted with each other was misleading, and we have rephrased such statements throughout the manuscript.

However, we would like to mention that cell-to-cell interactions could be simulated by adding some code and visualized by animations. However, this will be considered for future research (P30, L711).

There is substantial use of jargon and unnecessary abbreviations in the paper. This is not a minor point and severely affects comprehension and readability. In silico experiments are all simulated by computer. Why the need to keep track of p53Si-Sil(-CD)-Sim? Of BD, CD, MD, and CF; the terms BD, CD, and MD are not related, CD means cell death, CF is also not related to CD, but rather the opposite of BD. On pg. 22, the abbreviations are completely unnecessary, and the information of the abbreviations to be memorized by the reader do not come up until much later in the paper. Event is abbreviated Evt, why is this necessary to get rid of 2 letters, "en"? The abbreviation Evt is never again used until the Methods, thus it is only a methodological importance. I understand as a programmer Event and event type are coded as separate variables containing different data, but for the reader this is irrelevant and superfluous. Similarly, "Among the five simulation modes, Standard mode (Std) …" all of these abbreviations Std, Ind, MX, SS, MCS, in a search function of the PDF, are used only 2-3 times in the main text of the paper.

We have removed many of the abbreviations and jargon in the manuscript (e.g. BD, MD, CD, CF, Std, Ind, MX, SS, and MCS). For example, BD, MD, CD and CF were spelled out (P5, L94). Regarding p53 Si-sil(-CD)-Sim, etc., we have changed p53si to p53 RNAi, following the suggestion of Dr. Chen (Comment 40). Regarding “Sil”, we have changed the name of this cell population to p53 RNAi-Silico(-)cell death. We have also added the suffix “-Sim” where it was necessary to refer to cell populations generated by the simulation. We have revised the “Number of progenitors used for simulation (P19, L445)” section in the Results. We have removed the abbreviation “Evt” from the main text, and changed it to length of time (LT) in the Methods (P39, L862).

Reviewer #3 (Recommendations for the authors):Recommendations for improvementThe paper illustrates the advantages and pitfalls of interdisciplinary work, here cell biology and computational approaches. A more pedagogical approach would have made the paper's rationale clearer because some of the approaches, including the DIC image tracking and the cell simulation modelling are not part of the standard biologist's toolkit.The paper would be greatly strengthened if the authors could summarize the tracking and simulations through pseudocode or a plain language description of the algorithm used for simulation. That way, the average biologist with or without computational expertise would have a sense of the underlying algorithmic thinking, which is separate from the technical details of the computational implementation. Taking the time to explain the algorithm accessibly yet rigorously, allows a reader new to cell lineage tracking and/or simulations to come away with a better sense of the power of these approaches in cell biology.

We appreciate the encouraging comments. We have revised the Results section to improve the description of the algorithm. We have therefore included two sections, “Cell-fate simulation algorithm, P16, L363” and “Cell-fate simulation options with Operation data, P18, L413”, as well as Figure 7 (P57, L1289), Figure 8 (P58, L1306), and Figure 7—figure supplement 2 (P65, L1471).

Regarding the software for DIC segmentation and single-cell tracking, we described some details in our preprint (BioRxiv 508705; doi: https://doi.org/10.1101/508705 (2018)), and are planning to update and submit this for peer-review. We are currently cleaning up codes written during the past 10 years to allow third parties to perform code maintenance, and will include the pseudocode suggested by Dr. Colarusso in the preprint. We would like to make all the codes and software available to the public as soon as possible. In the current manuscript, we have included two sections regarding DIC segmentation (P35, L772) and single-cell tracking (P36, L798) in the Methods.

The paper also needs more transitions to guide the reader. Unless a reader has a background in this area, it can be hard to keep track of the different testing conditions and visualization of results as they represent a formidable number of combinations. At times reading the paper felt like trying to parse a logical puzzle rather than reading experimental results. In addition, the data sets are rich and varied, additional guidance is required in explaining results as in providing transition sentences that explicitly state the reasoning with a bit more detail.

As noted above, we have revised the section related to the algorithms and included a pseudocode (outlines, Figure 7—figure supplement 2). We have also revised the entire manuscript to try to make it more comprehensible to the readers.

In our previous version of the manuscript, we named the data used for the simulation as “Source data”. However, we realize that this journal uses this term for other purposes. We have therefore changed “Source data” to “Operation data” to avoid confusion.

In terms of the biological question, a statement about the novelty and impact of this work would be useful as it is not clear that the model is indeed testing the effect of homeostatic p53 levels. Tumour cells without p53 are able to avoid cell death and proliferate. It is unclear how having less of a wild-type protein in a cell represents a homeostatic model, distinct from having mutated forms within a cell. Also presenting a general result about homeostatic p53 using one cell line, also derived from a tumour, is questionable.

Please see our response to the use of one cell line and mutant p53 (Comment 44). With regard to mutant p53, we have included this in the Discussion section (P29, L686).

Additional comments for authors:1. How is the DIC segmentation and object tracking implemented?

Regarding the first question, *“How is the DIC segmentation and object tracking implemented?”,* please see our response to Comment 47.

Is the code developed by the group and is it available on Github or another public repository?

As noted in our response to Comment 47, we aim to make all the codes available upon submission of BioRxiv 508705; doi: https://doi.org/10.1101/508705 (2018).

"Is the code developed by the group and is it available on Github or another public repository? "

As noted in our response to Comment 47, we aim to make all the codes available upon submission of BioRxiv 508705; doi: https://doi.org/10.1101/508705 (2018).

Or is it available commercially?

Because code maintenance and technical writing (manual) requires a large amount of work, we have been looking for a partner, but have so far been unable to find a commercial partner that appreciates the need for a single-cell tracking system in the field of cell biology research.

It would be good to document the version of the code.

Our system includes 20 software items (BioRxiv 508705; doi: https://doi.org/10.1101/508705 (2018)), which were written by the last author of the manuscript. We are planning to release all these codes upon the submission of the preprint.

Also Cell Profiler apparently can segment DIC cells, so it would be good to know if the approach used by the authors is novel and/or available through open source, collaboration, purchase etc.

We did not mention commercially available or freeware resources because these did not meet our needs. In our early stage of developing the segmentation software, we tested thresholding, clustering, histograms, edge extraction, and region-growing methods by writing the code ourselves. These could work for certain types of images using certain parameters, but not for other images. Because we carried out long-term live-cell imaging for more than 7 days, the image quality changed over time due to cell growth and the formation of cell debris, etc. A total of ~200,000 image files (515 × 515 pixels) recording the different cell statuses were thus created, and we had to develop our own strategy to segment those DIC images, because existing approaches did not have the ability to process images with diverse qualities automatically, within a reasonable processing time. Most of our software was thus written using low-level computer languages, such as C and C++, which allowed memory allocation to be controlled manually. Cell Profiler etc. are designed for general purposes, but did not meet our specific needs. We have therefore not compared our segmentation method with others. From the point of view of system development, the entire process needed to be designed in a coordinating manner, e.g. file name assignment, data archiving, video viewing, segmentation, single-cell tracking, and data analysis. We have therefore focused on developing our own system rather than using existing systems.

However, we appreciate the point raised by Dr. Colarusso and have mentioned the main characteristics of our segmentation approach in the Methods (P35, L772).

With regard to availability, we are happy to make the process available to allow other research groups to carry out spatiotemporal analysis of individual cells.

On page 69, this is hard to decipher:Cell density map created by assigning value 1 to a pixel within the 20-pixel diameter area from the position of a cell (blue); if an area overlapped with other areas, the pixel was assigned the sum of the number of overlapped areas (light green).Can the authors clarify here as this may also help explain communicate the DIC segmentation/tracking approach better?

We generated a cell-density map using the cell positions determined by single-cell tracking. We then drew a circle from the position. If the circle overlapped with other circles, a value of 1 was added. Values were increased in areas of high cell density, according to the number of overlaps. We have added an explanation in the legend of Supplementary Figure 2 (now Figure 1—figure supplement 2) (P63, L1418).

2. Although the Figures, including Supplemental Figure 7, describe the simulation steps, a high-level description, such as pseudo-code, is missing. That is, a guide that helps the reader understand the logic and implementation of the simulation and help with the interpretation of the detailed steps given in Supplemental Figure 7.For example, my interpretation of the algorithm is below. It is highly likely my interpretation in incorrect, yet this gives an indication of how and why pseudo-code would be useful and may point out the parts of the explanation that need more detail.

We appreciate these comments. We have outlined the entire processing scheme of single-cell tracking (P17, L397) and cell-fate simulation in Figure 7—figure supplement 2 (P65, L1471).

– All these simulations are based on the empirical tracking data. [Note I am not sure how the authors extrapolated past the time length of the experiment, which means I am missing something important]

Simulations can be performed using both Operation data generated from empirically created single-cell tracking data and virtually created Operation data. We have described these in the sections related to the algorithms (P18, L413).

– Data sets inputted into the simulation include the location, cell division (BP or MP), cell fusion, or cell death, as well as technical notes that affected the tracking.

The current work did not take account of the location, and focused on other parameters, e.g. event patterns, for the simulation.

– Algorithm starts by loading in cell lineages obtained from experiments randomly and automated tracking/manual correction data.

We are unsure what you mean by “randomly”, but we hope that this is addressed in our revised explanations of the cell simulation in the sections related to the algorithm. In Figure 7—figure supplement 1, we wrote “randomize”, but it should be “generate random value”. This may be the cause of confusion. Thus, we revised Figure 7—figure supplement 1.

– Simulations progress by filling in the cell areas based on the cell lineage diagrams, and cells and events are encoded for ready visualization when needed.

Simulation *per se* is carried out based on event patterns (Figure 7), and the area is not a factor for the simulation. Regarding the cell area, this can be affected by the height of the cells (volume). We therefore considered that this did not provide useful information for the simulation. The animation was created to visualize the results of the simulation, given that the cell-lineage map created from the simulation data was too crowded to evaluate. We have mentioned this in the sections “Survival of the progeny of multipolar cell division“ in the Results, P24, L550.

– Cells were considered to be the same size throughout[?].

As for cell area, cell size was not taken into account for the simulation.

– I am not sure how the lineages are extrapolated past the 4000 min experimental time.

We have described how the simulation was performed in the section related to the algorithm (P16, L363).

– Also I would explain the reason the simulation is so useful. Here is an example where it may seem so obvious to the authors it needs no mention. Yet sometimes it is better to be explicit. It can seem you are talking down but it is helpful!

Thank you for your suggestion. We have added a section in the Discussion to emphasize how the single-cell tracking and simulation might be used (P30, L693).

With this simulation approach, we test whether the simulation matches the experimental results, and also uncouple the combined effect of cell division, cell death and cell fusion on cell proliferation, which would be difficult to do in an experiment with live cells.

We developed the simulation algorithm to follow the fates of cells beyond the time possible in empirical studies. We subsequently realized that this could be used to study the fates of cells in a population with other types of cells. In the typical setting of cell biological studies, comparisons are often made between e.g. cancer and control cells. Although such studies can provide information on mechanistic differences, it is difficult to evaluate how such differences affect the fate of e.g. cancer cells in the mixed cell population. Simulation can thus be used to investigate the fates of cells in a mixed population with other types of cells. We made a note in “Limiting expansion of p53 RNAi cells in the presence of Control cells” in the Results (P24, L561).

3. Tumour cells without p53 can avoid cell death and proliferate. It is unclear how having less of a wild-type protein in a cell is different from having mutated forms within the cells.

With regard to cells carrying mutant p53, we are interested in investigating if such cells have a p53 null+phenotype conferred by mutated p53, or if mutated p53 overrides the null phenotype. We have mentioned this in the Discussion section (P29, L686), and would like to investigate it further using our techniques.

– That is, how does the function differ here compared to a mutation in p53?

As noted above, this is an interesting question to pursue, given that most cancer cells carrying p53 mutations involve missense mutations.

– How does reduction in p53 followed by inducing cell damage test the need for basal p53 as a unique pathway in cancer progression?

This is an interesting question. In order to study the effect of lowering p53 levels on the cellular response to damage induced by physiological levels of a damaging agent, the damage is often undetectable using existing assays. However, a higher dose of the damaging agent could cause the cells to adopt a more homogeneous status, e.g. by inducing cell death, which may be not be relevant to the process of cancer progression.

If we use physiological levels of the damaging agent, we also need a technique able to deal with the heterogeneity of the cells. On the other hand, if the status of the cells is homogenized by exposing cells to e.g. a lethal dose of damaging agent, an average-based assay, e.g. cell death assay, can be used, although the status would not reflect the status of the cells under physiological conditions.

A similar paradox also exists for the studies using p53-proficient cells.

– In addition, more justification of the choice of A549 cells chosen as the model is required, in light of the statement in the Introduction: Here, we investigated the effects of low levels of p53 on the behavior of cells using empirical single-cell tracking.

In addition to the fact that A549 cells are p53-proficient cells, we also used these cells because their growth pattern is relatively linear. Some cell types, e.g. MCF10a (non-transformed breast epithelial) cells, are highly motile, but such motility is reduced by attaching cells together (a kind of contact inhibition, but not like primary fibroblasts). This reduced motility initially occurs locally and then spreads to the entire population as the cell density increases. These would be interesting cells for modeling, but it would be necessary to characterize their cell behaviors. We therefore considered that A549 cells were easier to handle. We have mentioned this in “Culture of A549 cells” in the Results (P6, L118).

What is their endogenous expression of p53 in A549 compared to other p53 proficient cells?

In the early stages of this study, we searched for studies regarding endogenous levels of p53 in unstressed cells, but were unable to find any specific papers dealing with the low levels of p53 itself. Some studies showed used cells with low levels as controls compared with accumulated levels, but the focus of these studies was on the function of the accumulated p53. In order to investigate the relationship between endogenous levels of p53 and induced cellular events, it is necessary to establish a technique to quantify the levels of p53 (or other proteins or genes) in individual tracked cells, and, as noted above, no such technique is currently available. However, it is possible that a technique could be developed by combining single-cell tracking analysis with e.g. spatial transcriptomics. We have mentioned this in the Discussion (P30, L693).

How do these levels compare to normal cells, such as those derived from primary culture?

Please see the above response. In addition, in order to make meaningful comparisons between cells with different levels of p53, it will be necessary to develop a method combining single-cell tracking analysis with e.g. spatial transcriptomics. Concerning primary cells, Guo et al. (*Cell* 156, 649-662 (2014)) found that the efficiency of iPS cell formation was increased by silencing the low levels of p53 (ref 19), and we are certainly interested in extending our study to other cell types.

– How does inducing DNA damage test the function of low levels of p53 in unstressed cells?

This is related to Comment 53, and as noted above, it involves the paradox of detection of damage vs. the status of the cells. If unstressed cells are exposed to a DNA-damaging agent that can induce detectable levels of damage, it could cause the accumulation of p53. On the other hand, if unstressed cells are exposed to physiological levels of a DNA-damaging agent, it may be difficult to detect the damage. We therefore consider that it is difficult to address this question without resolving this paradox.

Wouldn't a better model be to compare different cell lines with different endogenous levels of p53 and see how they respond?

The inclusion of more different cell lines would introduce new contexts. We agree that this type of study would be interesting, but at least six cell lines may be needed to allow a meaningful statistical analysis, and if a new context was found, more cell lines would need to be analyzed. If the behaviors of cells in a cell population differ (e.g. MCF10a vs A549), it may be necessary to develop a new algorithm to take account of the differences. There are thus many interesting possibilities, but these would involve too much work to be tackled by our group alone. We would therefore by happy to share our methods with other groups interested in performing similar studies.

Or this could be used as a second method to back up the conclusions when you damage silenced cells, it seems that the model is not delineated precisely for studying low levels of p53 on cells. Rather the test is studying the effects of DNA damage on p53 null cells which has been done many times? It would be good to hear more about the rationale here, as it is not clear from reading the paper's arguments.

With regard to the silencing of low levels of p53, we believe that our study provides a conclusion. Although it would be possible to prepare a manuscript focusing on this aspect, we chose to include the responses of cells to DNA-damaging agents, given that p53 is often referred to in the context of DNA damage.

As mentioned by Dr. Colarusso, previous studies have compared the responses of p53-null and Control cells exposed to DNA-damaging agents. However, the characteristics of the cells change over time, and there is no guarantee that the observed difference reflects the presence or absence of p53. In addition, cell status can be affected by the dose of DNA-damaging agent, with high doses potentially killing any growing cells, while intercellular heterogeneity becomes an issue at lower doses. Many factors may thus affect comparisons among cell types, or among cells treated with different doses. We therefore limited our study to a single cell type and transient silencing to minimize the influence of such factors on the interpretation of the results. We have emphasized this point in “System to investigate the functional implications of maintaining low levels of p53 in unstressed cells” in the Results (P5, L88).

Detailed questions:4. Why were the cells monitored at such a high density?

Although A549 cells are less motile than e.g. MCF10a cells, they still move around at low densities, when the cells are isolated from each other. In contrast, the cells are less motile at higher densities. Because we wanted to minimize motility, we carried out imaging at a density at which most cells were attached. We have mentioned this in the Results (P6, L118).

Was this choice governed by the biological question and/image analysis algorithm limitations, or some other reason(s)? (to follow the fate of individual cells by monitoring live cells, because the culture eventually became over-confluent).

From a technical point of view in relation to image segmentation and single-cell tracking, lowdensity cultured cells are easier to track than confluent ones. As noted above, we chose the conditions to reduce the motility of A549 cells.

5. Why were the concentrations of CO2 different in the incubator and experiment? Is it due to maintenance of pH, and if so, it would be good to mention this as not everyone monitors pH when carrying out long-term imaging?

Some textbooks recommend using a colorless medium for long-term live-cell imaging. However, this does not allow the pH of the medium to be monitored, and several groups have failed to maintain cell cultures in an environmental chamber for long periods, particularly using multiwell chambers. We therefore used 7.5% CO_2_ to maintain the pH of the medium. We have mentioned this in the Methods (P34, L755).

6. The authors mention maintaining the integrity of the cell population in multiple sections, including the summary. How is integrity defined? Is it metabolic, genomic, structural?

We have removed the term “integrity”, because this over-interprets our model.

[Editors’ note: further revisions were suggested prior to acceptance, as described below.]Main issues:– l. 316-318. The authors suggest that P53 RNAi cells proliferate faster, but have more cell death. Can this not be quantified directly from the data, rather than in the indirect manner pursued here? For instance, one could measure the probability that a cell will divide again, if it doesn't undergo cell death. The prediction would be that this probability is higher for P53 RNAi cells compared to Control.

This comment is related to the *in silico* generation of cell lineages. In this analysis, we evaluated the impact of the occurrence of cell death on the cell population expansion rate of p53 RNAi cells.

Concerning the quantitation of the number of cell deaths in the Control and p53 RNAi cells, this can indeed be carried out by analyzing the cell-lineage database or from live-cell imaging videos, as shown in Figure 4 and Figure 4—figure supplement 1, and the results accordingly show more cell death in p53 RNAi cells compared with Control cells. We had mentioned this in the previous version of the manuscript.

The main issue we wished to address was the impact of cell death on the cell population expansion rate, given that frequent cell death would counteract the expansion of the cell population. Both Control and p53 RNAi cells showed similar rates of cell population expansion (Figure 2c and i), but there was more cell death in p53 RNAi cells (Figure 4). These data thus suggest that some other factor(s) is/are altered in p53 RNAi cells, resulting in similar cell expansion rates in p53 RNAi and Control cells. The important question is what other factor(s) was altered?

We think that considering the “*probability*” per se does not address this issue, while analysis of cell-lineage data may provide a clue. By examining cell-lineage maps, we noted that some of the p53 RNAi cell lineages had higher reproductive abilities than Control cells. Thus, the promotion of reproductive ability may counteract the increased cell death in p53 RNAi cells, resulting in a similar cell population curve to Control cells. However, examining the cell-lineage map cannot prove this, and we therefore developed the *in silico* approach to examine this hypothesis. To clarify this matter, we have revised the first paragraph of this section (L335-371).

“*The authors suggest that P53 RNAi cells proliferate faster, but have more cell death.”:*

Notably, we did not specifically say that p53 cells proliferated faster, but rather that their reproductive ability (the number of progeny produced from a progenitor) was increased by the silencing of p53.

– l. 391-396: This section is still unclear to me. Why not follow the same approach as shown in Figure 7c for time to MD? Is it because there is a correlation in cell cycle time between generations, i.e. a rapidly-dividing mother has rapidly dividing offspring? This should be explained more clearly.

Compared with bipolar cell division, multipolar cell division contributes less to determine the population size, given that most progeny produced by multipolar cell division undergo cell death. We have added an explanation in the related section (L467–470).

– l. 418-421. I now understand conceptually what the others do, but from the text and Figure 8 I don't understand how this works in practice. Do the authors linearly interpolate between two Operation data sets? If so, then write that down explicitly. Some things in the text here still make no sense to me. "A start event of 10 and 1", what does this refer to, what sort of event? "The chance of a cell population exposed" A chance of what? Is there some probabilistic process that underlies this?

This comment is related to the Dose simulation mode. We have added an example to explain how the calculation was made and have revised the related section to clarify our meaning (L488-498).

– l. 489-494. I am not sure I properly understand the origin of shorter doubling time. Is the problem that the cell cycle time is not drawn from a measured distribution (as in Figure 7b) and therefore misses rare but very long cell cycle times?

In the related section, we intended to explain the possible reasons for some differences between cell-doubling time data obtained by single-cell tracking and simulation.

First, the difference may be related to the method used to assign the length of time between events; i.e., selecting a time from histogram data. In this case, less frequently occurring time lengths may have less chance of being assigned.

Second, as Dr. Colarusso pointed out, if the restraining range is > ±10%, the simulation of average cell-doubling time would change.

Third, the simulation data represent the actual distribution of cell-doubling time, while the distribution determined by single-cell tracking is biased because of fewer data compared with the simulation (e.g., ~2,000 v.s. ~9,000,000).

However, given that the difference was constant among cells, we did not make any adjustment, though this could be done by adding a certain bias upon assigning a doubling time. We have included the above explanations in the revised manuscript (L567-573).

Typically, cell-doubling time distribution tails towards longer doubling times, implying that cells with longer doubling times are generated from those with a shorter doubling time. Notably, a balance between the proliferation of cells with shorter cell-doubling times and the generation of longer ones may be critical to maintaining cell populations. The simulation of this process warrants further studies to clarify how the cell population is maintained.

– l. 523-526. This seems a rather counterintuitive effect: removing the P53 stress response leads to higher cell proliferation when higher stress is applied. I do think it should be emphasized more that this is counterintuitive and perhaps also a possible explanation for this effect should be given.

We have revised the related sections (L606-608). We have tried to clarify the relationship indicating that Control cells show a stronger response to MNNG than p53 RNAi because of the p53 response.

With regard to the statement, *“removing the P53 stress response leads to higher cell proliferation when higher stress”*, we would like to make a clarification, as the statement contains some ambiguity.

p53-induced stress responses, in general, lead to suppression of proliferation of cells by the induction of damage (stress) response (Introduction); not cause higher proliferation. This occurs through the elevation of the level of p53 in cells.

When the low level of p53 found in the non-stressed cells is removed, the reproductive ability of cells is increased, but cell death is also induced (our finding).

– l 532-535. It is not clear what the simulations can add here. Is it not possible to use experimental data to measure this directly?

Experimental data can be used directly but is limited by the scale of the practical analysis. For example, in the case of dose-response curve determination using classical colony-formation assays for two cell types, 3 dishes/dose and 10 doses can be done with 60 dishes; however, it would become more difficult to carry out these assays for more doses or cell lines. This limit can be significantly expanded by employing the simulation approach.

After all, the simulations are driven by experimental data, so in principle, there cannot really be anything new from the simulations in that regard.

As noted above, there are limitations to using an experimental approach. Considering the experiment shown in Figure 10 (a, b, and d): using a conventional approach, the cell culture has to be maintained for ~25 days, the number of doses is 15, and the cells are counted every day for 25 days, thus requiring at least 2250 dishes (15 doses × 25 counting × 3 (triplicate) × 2 cell types). Based on our experience, this is not realistic. We therefore consider that the results shown in Figure 10a, b, and d would be difficult to produce without using simulation.

With regard to “*anything new”,* we would like to make a clarification*.*

As single-cell tracking has never been used to investigate the function of p53, we think that data generated by single-cell tracking is “*new*”.

Thus, simulation data *per se* is also “new” relative to other data published by others. Relative to single-cell tracking data, simulation data shown in Figure 10c could provide a more detailed dose-response curve. However, the data shown in Figure 10d cannot be generated by single-cell tracking because of the limitations of real experiments.

In the manuscript, we add a statement that “to determine a more detailed dose-response curve” (L592–594).

– l 535. "we assumed" I don't understand what the assumption is based on. Measurements in this paper? Existing biological knowledge?

We used the word “*assume*” (as explained below), but agree that this word may be misleading, and have changed it to “hypothesized” (L616) in the revised text.

Many studies of p53-mediated damage response use, for example, > 10 Gy of ionizing radiation, and the induced responses tend to be interpreted as related to p53. Although this dosage has been used because of a lack of sensitivity in many classical approaches, it is an extremely high dose that could kill most cells within a day. Indeed, ionizing radiation induces various types of damage, and cellular responses induced by radiation are also caused by non-p53-related mechanisms. However, such distinctions are often not made in many reports. Although this seemed obvious (hence the word “*assume*”), we appreciate that it cannot be considered as “*Existing biological knowledge*”.

– l. 550-551. The low frequency of multipolar division is put into the simulations directly from the experimental measurements. So why is this simulation result, which makes exactly the same conclusion, surprising or interesting?

We would consider that, in a sense, the result to be confirmatory, rather than surprising or interesting. The hypothesis that the growth of progeny of multipolar cell division causes cancer (aneuploidy) was proposed in 1914 by T. Boveri (English translated version; ref 37). Gene mutations (chemical carcinogenesis) have subsequently become a central dogma of cancer development while it has been common knowledge that cancer tissues contain a high level of aneuploidy cells. Although our current work is not related to the cause of cancer, our results confirm Boveri’s hypothesis, as mentioned in our previous version (L748-750). We think it is often important to “confirm” a hypothesis that has been put aside for so many years.

Our single-cell tracking analyses allowed us to detect the progeny of multipolar cell division that undergo bipolar cell division, but the data did not demonstrate that the progeny could form a population, in generic words, a cell colony (it is not possible to extend single-cell tracking beyond cell confluence). However, the simulation results suggest that the progeny could indeed form a cell population. *We think it is important to confirm that such progeny could form a cell population.*

We have emphasized this in the revised version (L638-643).– l. 552. "growth of virtual cells" It is not explained in the main text how the spatial cell dynamics is implemented in the simulations. It is also not explained why space is even important and what novel insights it will bring to incorporate it.

In our previous responses, we mentioned that the animation was created to visually show cells of interest, rather than to analyze the spatial dynamics. Notably, we did not specifically say that we analyze spatial dynamics in our original and previous revised manuscript.

We evaluated the results generated by single-cell tracking by examining cell-lineage maps. However, the simulation generated cell-lineage data with many cells, which were difficult to evaluate by examining the cell-lineage maps. We therefore created the animation as a tool to allow us to find a cell of interest in the simulation data. This placed the progenitor within a 2D space and showed the proliferation of progeny by coloring the cells of interest. We did not use the animation to analyze cell dynamics in 2D, and we apologize for the expression we used if the reviewer misunderstood this. Similar comments were raised in the review of our original manuscript, and, in our previous version of manuscript, we have already responded accordingly.

Notably, however, our simulation algorithm has the potential to include simulating 2D dynamics, and we have discussed this in our previous revision (L801-803).

We have emphasized the purpose of the animations in the revised text (*L638-643*).

– l. 555-559. It is not clear why showing spatial expansion of MD progeny is important.

When distinct cells start making a population, they expand themselves in a space (e.g. generation of iPS cell colonies); thus, if a cell is expanding spatially, it is generally considered that a new cell population is being created. Similarly, expansion of the progeny produced by multipolar cell division implies the formation of a new cell population. We considered that it was important to show, using simulation, that a progeny derived from multipolar division could form a cell population, given that these progenies are generally considered to be fragile and may not be able to form a population. We have emphasized this in the revised text (*L638-643*).

Couldn't the same insights be gained from purely looking at images?

Of course, the same insights could be gained from images. However, the imaging data contain various types of information, and the aim of the study was to convert the imaging data into numeric values to characterize specific aspects of the cells, using image segmentation and cell-tracking algorithms. Thus although some ideas can be obtained by looking at images, publishable and analyzable data cannot be produced by simply “looking at images”.

This just means that you get clones of cells that are MD cell progeny, why is it important where they sit in space?

As noted above, the animation was created to visually show cells of interest, rather than to analyze the spatial dynamics.

– l. 578. What novel insight does this section bring?

We have revised the end of the paragraph to summarise the implications (L678-682). The implications of the analysis have been explained in the Discussion.

The stress-shift experiments show the same conclusion of P53 RNAi outcompeting Control cells as Figure 10.

We used a different simulation approach in this section from that used for Figure 10 (L665-666).

Also, what does the inclusion of space bring to Figure 13a? Would the results in Figure 13b-h be different if space was not included? If not, then I would remove the spatial analysis.

As noted above, the animation was created to visually show cells of interest, rather than to analyze the spatial dynamics.

– l. 636. "cancer tissue mass". I don't understand why this result is compared to cancer: most cells are Control (=wild-type?) cells with few P53- cells interspersed, and most of the growth is due to proliferation of Control cells, not P53- cells.

We have rephrased this sentence (L724-725). We have provided the number as a reference.

Reviewer #3:Comment 1:The experimental data and single-cell lineage analysis are important contributions to the field. Although the authors have included more detailed explanations, the additional insights offered by single cell tracking need to be described more clearly and emphasized throughout the text.For example, when the authors move from reporting on the population level counting to the single cell tracking results (lines 197 -198), the authors should guide the reader by providing a transition between both approaches. Currently, the authors launch right from the population counting results to validation of single cell tracking without building up the need for single cell tracking.To quote: We then compared the results obtained by cell counting with those obtained by computer-assisted single-cell tracking analysis to verify whether this analysis indeed yields results consistent with the classical counting method.Before comparing the results, a transition highlighting the need for single-cell analysis is missing as well as the need for comparing both methods is needed. Suggestion: "Thus far, we have described how the different treatments affect cell numbers at the population level. We then analyzed the data sets using single cell tracking. This powerful approach can reveal how the events occurring at the individual cellular level contribute to what is observed at the population level."Only then move on to the comparative results such as by stating: To test the accuracy and self-consistency between our standard cell counting and single cell tracking, we compared the results obtained with each approach.Similarly, when the results obtained from single cell tracking add new information, additional context should be provided for the reader. This paper is cognitively challenging to read and interpret, and more explanation is needed to help the reader navigate and appreciate the significance of the results.Comment 2:Similar clear emphasis is needed when describing the simulation approaches and their significance. To illustrate: In the section introducing the cell-fate simulation algorithm (lines 363-372), confluency is mentioned as are "limitations of empirical approaches." Rather than stating confluency and the vague term "limitations," the justification needs to be clear from the start. My suggestion is move the explanation found later in the text to this introductory section.For example, lines 440-442 state: "cell-fate simulation using Operation data allows the creation of various simulation options and provides flexibility for designing virtual experiments that would be difficult to perform empirically."Rather than letting the reader wait for this statement, I recommend leading in with a similar statement (and dropping the technical reference to Operation data). Suggestion:" Cell-fate simulations are powerful and flexible tools help us model conditions that are not readily accessible by direct imaging, such as mixed cultures and mixed dosages. "

We appreciate these suggestions. We have included more emphasis on the advantage of using the simulation (L417–429), and have also made revisions based on the other issues raised.

Other comments:3. Ensure that cell growth is used when referring to cell size not to cell division within the text. The term "grow" is used colloquially in the lab but here should be reserved for area/volume not cell division. For example, see how "growth" is used in Line 125. Here it is confusing as the passage is describing cell area decreasing and then grow is used to mean cell proliferation.

We have checked the manuscript and used the terms “proliferate” or “cell population expansion” accordingly.

4. What random number generator was used?

We generated the random numbers using the C++ rand () % value. We have mentioned this in the Materials and methods (L1032).

5. Figure 1—figure supplement 3. Y axis in last graph has the label "Variation" but the units are missing.

We have added the units (no. of cells).

6. When discussing Figure 5, explain how the resultshighlight the need for single-cell tracking studies and how this approach complements and enriches the population level studies. Suggestion: "The cell numbers over time are roughly equivalent for both the scr siRNA and the p53 siRNA treated cells, as shown in Figures 2c and 2i. The single cell tracking data, however, reveal differences that are not directly accessible through standard cell counting. Figure 4 reveals that although the overall cell death is higher in the p53 siRNA cells, it is compensated by increased multipolar cell division. Further, the simulations can help unravel the relative rates of multi-polar cell division compared to cell death. As shown in Figure 5, we see that the relative proliferation curves can be simulated in the presence or absence of cell death. By analyzing the individual cellular fates under well-defined conditions, we can then simulate scenarios that are not accessible through our direct imaging, such as analyzing how a population of mixed cells and/or heterogenous treatments will evolve over time. "

We appreciate the suggestions and have used some of the suggested phrases. We have added some sentences to the section related to the *in silico* analysis to emphasize the points raised (L366–371).

7. Line 226. Quote: To this end, we sorted each cell lineage into groups (Figure 3a).The division into groups introduces a layer of complexity and seems arbitrary considering the small number of possible bins. Why is reporting the binned data through groups better than reporting the numerical value?

This section is related to the analysis of the reproductive ability of cells comprising a cell population. Dr van Zon and Dr Chen pointed out (in their comments on the original version of our manuscript) that this section was difficult to follow. We extensively revised the section, and detected an error.

Regarding the error, we used two approaches to evaluate the reproductive ability of cells comprising a cell population; i.e., we determined the number of cells comprising a cell lineage, and the number of progeny found at a certain point in time. Both yielded similar results, but the former included cells that undergo; e.g., cell death. In our original version of the manuscript, we showed the data obtained using the former approach, and the same data was shown in the previous revision of our manuscript. However, when we added an explanation regarding how reproductive ability was analyzed, we used Figures for the latter approach (Figure 3a). We have corrected this error in the current version.

Regarding the “bin”, we combined, for example, cell lineages comprising 1 and 2 cells to generate the data, because the number of cells comprising a cell population tends to be an odd number (progenitor: 1, after the first division: 3, after the second division: 7 etc.), and odd numbers are thus more frequent than even numbers. A numerical graph would thus show ups and downs between consecutive numbers. We therefore “binned” the data. However, we agree that binning; e.g., 1–3, is subjective. We have therefore binned the odd and even numbers; e.g., cell lineages comprising 1 and 2, 3 and 4, etc., in the revised version. The results using this method were similar to the previous results, but some calculated numbers were changed (% of a cell in the total number of cells). We have revised the relevant paragraphs (L242–280), figure legend, and source data.

In the annotated pdf, the reviewer asked why we did not use an average. We considered that the average value would not provide useful information regarding the reproductive ability of cells comprising a cell population. The average would be obtained by first categorizing each cell lineage based on the number of cells comprising the lineage, and then calculating the total number of cells belonging to the category. For example, if 10 cell lineages were categorized as lineages comprising 5 cells, the numbers would be “Cell lineage: 10, total number of cells: 50”. In this case, the average (number of cells/category) would be 5 (same as the number of cells per lineage). However, we wanted to show how many cells were produced from a progenitor, and we therefore think that the total number of cells/category could better represent the reproductive ability of cells comprising a cell population.

8. Lines 282-286. Quote: …the number of cell death events in Control cells determined by counting was higher than that determined by single-cell tracking analysis. This was because the area of single-cell tracking was slightly outside the area where cell death frequently occurred (Figure 4—figure supplement 1c). However, given that such events do not occur with the same probability throughout the field of view, some variation may occur with both the counting and single-cell tracking approaches.Was this because cells that die move more than cells that do not or is it something specific about the cell imaging chamber with death occurring near the edges more than the centre?

We took an image of cells located close to the center of the cell population, given that the behaviors of cells located at the periphery were different from those at the center. We have emphasized this in the revised Materials and methods (L838–840).

In this sentence, we intended to indicate that cellular events do not occur with uniform probability throughout the field of view, and this variation may thus affect the results obtained by counting and single-cell tracking. Based on our experience, such variation occurs regardless of the type of dish or chamber. For example, in simple colony-formation analysis, larger colonies sometimes appeared more frequently in a certain area of a dish. Although using a larger number of dishes would normalize the results, the required number of dishes may become too large, making the experiment unrealistic to perform. In the case of single-cell tracking, increasing the area of imaging and the number of tracked cells would thus make the analysis *per se* unrealistic to perform. One aim of the current work was thus to find a realistic experimental scale for single-cell tracking. In the section related to Figure 1—figure supplement 3f, we mentioned that tracking ~240 cell lineages was sufficient to generate reproducible single-cell tracking data based on the rate of cell population expansion. We also evaluated the frequencies of multipolar cell division and cell death, and, after taking this into account, we selected 335 cell lineages. We have revised the manuscript to clarify the points raised by the reviewer (L322–326).

9. I am confused by the reference to the "accuracy of a detecting a low frequency event." Quote: In general, the accuracy of detecting a low-frequency event is lower than that for frequently occurring events, e.g. bipolar cell division…Are the authors referring to the probability of the event being captured by the imaging system rather than the detection of the event by their software and/or visual scorer? Or do they have evidence that the detection by the software and/or visual scorer is lower for these events compared to bipolar cell division?The authors state in line 260 "We therefore determined if single-cell tracking could detect multipolar cell division, cell death, and cell fusion with adequate accuracy for statistical analysis. To this end, we determined the number of multipolar cell division and cell death events visually by manual counting in videos, and compared the results with those obtained using single-cell tracking analysis (Figure 4). Notably, cell fusion was not included in the counting analysis because it was difficult to detect without single-cell tracking."Yet when review Figure 4, I am having trouble deciphering how the authors concluded that the single-cell tracking was accurate as the results are not directly compared to visual scoring. Please clarify.

We realize that “detecting” is not an appropriate word, because it gives the impression that bipolar and multipolar cell divisions were determined using a device. However, these divisions, as well as cell death, and cell fusion, were found by following each cell. We have revised this section accordingly (L287–297).

10. Lines 316-318: These results suggest that p53 silencing promoted the reproductive ability of p53 RNAi cells, but this was counteracted by the induction of cell death, resulting in the formation of p53 RNAi cell populations that were smaller than Control cell populations (Figure 2i).I see a small dip in the p53 RNAi curve compared to the Control curve in Figure 2i, but otherwise they look similar. Please clarify.

Single-cell tracking data for p53 RNAi cells showed that the rate of cell population expansion was reduced at ~2,000 min, and the population size then caught up with the Control. We have revised the text to clarify this (L356-359).

11. Line 318 Analysis also shows that the effect of silencing the low levels of p53 cannot be detected without access to the spatiotemporal information on individual cells provided by single-cell tracking.Although this statement is likely, the counter-example can not be ruled out. Just to list possible alternative approaches that could reveal similarly nuanced effects of silencing p53. One can imagine a scenario where the number of live and dead cells are measured over time, without any single cell tracking. Or it may be possible to model the number of cells over time using an analytical expression that embeds cell division (bipolar, multipolar) as an exponential and cell death (say as a linear function of time). Suggestion-qualify statement: "This analysis also demonstrates single cell tracking provides more nuanced and detailed information about the effects of silencing p53. Here the single cell analysis revealed clear differences in the rates of cell proliferation and cell death among the scr siRNA and p53 siRNA conditions, differences would be obscured and/or missed if limiting the analysis to cell counts at the population level."

We appreciate these suggestions and have revised the end of the section using some of the suggested phrases (L366–371).

12. Line 331: Following cell fusion, Control and p53 RNAi cells demonstrated 79.4% and 93.2% multipolar cell divisions.Please define the % values-how are they calculated? Also the numbers are listed on the bar graphs in Figure 6- the relative percentages of listed over the bar graphs is confusing in light of the numbers being relative to another 100, the 100 cell lineages. Also it would be helpful to see the spread of the data points illustrated by the bar graph.

Patterns 1 and 2 (now 1a and 1b) make 100%, and the value given refers to Pattern 1a. We have revised the text (L374-393) and mentioned 100% in the figure.

Regarding the 100 cell lineages and 100%, we have changed the number of cell lineages to 50 to avoid confusion and have revised the Figure and Source data accordingly.

Concerning the data points, we did not include these in the figure. To perform the analysis, we first searched for multipolar cell divisions and traced the cells that underwent this division back to determine the prior event. Totals of 34 and 59 multipolar cell divisions were found in the Control and p53 RNAi cells, respectively, and were used to produce data. Because different numbers of divisions were used, we showed the data as %. We have included the raw data related to the analysis in a Source data file in the current version of the manuscript.

13. Staring on line 385, the authors state: The algorithm then reflected this distribution to choose the End event.– How is the weighted random assignment implemented? Specify the method or methods and justify why the choice is appropriate.

We generated the histogram arrays (for bipolar cell division, multipolar cell division, and combined events; e.g., multipolar cell division followed by cell fusion) for the Start events. Each array contained information regarding the event that occurred following the Start event. The internal data format was e.g. 0:BD (bipolar division), 1:BD, 2:BD, 3:MD (multipolar division)…100:MD. A random value of 0–100 was created, and a value of e.g. 1 picks up BD (L442-452).

– Similarly, for LT and other random assignments, is the implementation always the same as for the Event assignment?

Yes. The selection method *per se* was the same as that for Event. The only difference was the array that was used for the assignment (L456–458).

– Is the additional +/- 10% in ET imposed for BP events based on experimental data obtained here and/or reported elsewhere or there some other reason for constraining the ET for BP events in this way?

We performed a repeated simulation to find the optimal restraining value. We have indicated this in the revised text (L463).

Further, could this constraint have led to the observation summarized in lines 489-494: On the other hand, the simulation tended to yield an average cell doubling time about 2 h shorter than that determined by single-cell tracking analysis. Given that the algorithm assigned a cell doubling time to each cell by generating a random number with Operation data-Time (Figure 7), a long cell doubling time, e.g. 3,000 min, which occurred less frequently, may be less likely to be assigned, resulting in the generation of a simulated cell population

Reviewer #2 also commented on this section. As suggested, the restraining range (±10%) could also affect the simulating average cell-doubling time.

In addition, the simulation data may also represent the actual distribution of cell-doubling time, while the distribution determined by single-cell tracking may be biased because of fewer data compared with the simulation.

We have noted these possibilities in the revised manuscript (L567-573).

– When CF occurs, do these events also reflect the frequency of sibling vs non-sibling under the different conditions studied?

No. This algorithm does not assign cell fusion between non-siblings. Assignment of cell fusion between non-siblings was not complicated, but could create a problem when displaying cell-lineage data. When cell fusion occurred between non-siblings of, for example, Lineages 1, 10, and 20, we displayed all cell lineages in one window. To do this, the display order of each lineage has to be determined, which requires complicated coding. Because a cell lineage generated by the simulation contained a large number of cells, arranging the display order of multiple cell lineages was extremely difficult. Furthermore, cell fusion between non-siblings occurred less frequently, and we therefore did not include the simulation of fusion between non-siblings. We have noted that the algorithm generates fusion between siblings, but not non-siblings, in the revised text (L1106-1107).

14. I have trouble understanding why calculating Recovery is important to the simulation and why it is listed before the other steps. Line 894 states: Recovery data was used to simulate the rate of recovery a cell population from the treatment of MNNG. When cells were treated with MNNG, the majority of cells may be killed. However, small number of cells gained reproductive ability, which could be found at the end of imaging.I am confused here as I don't see evidence of the majority of cells being killed in the population expansion data and don't understand why they need to be tracked.Lines 903-907: For example, if any of the progeny derived from a progenitor underwent bipolar cell division at last within 20% of the single-cell tracking period, e.g. 320 to 400 min when tracking was performed for 400 min, this cell lineage was counted as one upon the calculation of the percentage. Thus, if 10 such cell lineages were found out of a total of 300, the recovery percent was 3.3%.I don't understand this statement-what does percentage refer to? Why is only BP considered and not MP, for example? Any why is calculating this value important to the simulations as illustrated in Figure 7, supplement 1c, Step 3?

After pulse treatment of cells with a drug (e.g., high dose), most cells may become static or die, but some cells can regain reproductive ability. However, if a cell regains reproductive ability close to the end of the tracking time, no information regarding its doubling time can be found. Recovery thus calculates how many cells undergo bipolar division during a certain length of time (e.g., 3,600–4,000 min (end of tracking time)), and the algorithm then generates cells that are destined to gain reproductive ability. The algorithm assigned > 80% of the length of tracking time to the cell (3,200 min in the case of 4,000 min of tracking). We determined 3,600–4,000 min and > 80% as the value resulting in exponential cell population expansion for cells treated with MNNG7.

We have provided additional explanations in the revised version. Because A549 cells undergo non-dividing status following MNNG treatment, “killed” was not a correct term in relation to these cells.

We have revised the section related to Recovery (L991-1006).

15. Line 918-920: To generate a simulation array for the Operation data-Event, the percentage of each event relative to the total number of cell divisions (total of bipolar cell division and multipolar cell division) was calculated.As shown in Figure In Figure 7, supplement 1 c, in the "Event" sequence I don't understand why calculating the "% event of total cell division" is possible before the simulation arrays are created?

There was an error in this sentence. Simulation array was generated “from” rather than “for” Operation data. We have corrected this error (L1016).

16. Lines 931-933: the array was empty, the average time that bipolar cell division occurred was calculated and −25% to +25% of the time selected by random 933 number was assigned as LT, and bipolar cell division was then assigned to the cell.Is this based on experimental data and/or another reason?

If the FirstEvent array was empty, −25% to +25% was assigned. FirstEvent array holds the length of time to an event that occurred in the progenitors, implying that if the array was empty, it contained no cell-lineage data. Thus, if arrays containing data were used, processes related to the −25% to +25% would not be carried out. This step was included to prevent a software crash due to the processing of an empty data array, and −25% to +25% thus does not have any specific meaning.

17. I appreciate the care and effort that went into detailing the algorithmic approach used for the simulations. The logical flow now is much clearer and moves away from being a "black box" to now being reproducible.

We appreciate your help.

Recommendations and questions:– Figure 7 supplement 2. Recommend placing this before supplement 1 as it presents the overall logic.

We have changed the figure numbers following this suggestion (L430).

– In Figure 8, supplement 1a, (page 94). Change "Uze" to "Use" in Step 6.

We have corrected this typo.

– In Figure 7, supplement 1b (page 95), Step 3 defines the nBD but the description is confusing. Clarify the "last time point of cell tracking" – is this simply the LT associated with the event? Also what is the offset and what is its purpose?

We have revised Figure 7, supplement 2b (Recovery). nBD is the number of bipolar cell divisions that occurred within a certain length of time from the time end. The default value to calculate the length was 20%, but this could depend on the cell type. We have therefore removed the number.

– In Figure 7, supplement 2, there is a typo in "Select an e-vv-ent type"

We have corrected this typo.

18. Consider revising sub-headings to guide the reader. Revise to express purpose, result, significance etc. These could serve as signposts throughout the paper would improve the paper's accessibility.

We have revised some of the subheadings to summarize the main point of each section.

19. In future work, consider graphical representations that are not limited to colour as in "burgundy" plot as 8-11% of the male sex is R/G colour blind. Consider using different plots styles as well as colour. I am not asking the figures to be revised for this publication, but ask that the authors consider visually accessible graphics / figure in future publications.

We appreciate this suggestion.

20. Similarly, consider box plots rather than bar graphs for future work.

We greatly appreciate the provision of the annotated pdf. We have incorporated your suggestions/corrections in the revised manuscript.